# Decrypt Modality Gap in Multimodal Contrastive Learning: From Convergent Representation to Pair Alignment

## Abstract

Multimodal contrastive learning (MCL) aims to embed data from different modalities in a shared embedding space. However, empirical evidence shows that representations from different modalities occupy completely separate regions of embedding space, a phenomenon referred to as the modality gap. Moreover, experimental findings on how the size of the modality gap influences downstream performance are inconsistent. These observations raise two key questions: (1) What causes the modality gap? (2) How does it affect downstream tasks? To address these questions, this paper introduces the first theoretical framework for analyzing the convergent optimal representations of MCL and the modality alignment when training is optimized. Specifically, we prove that without any constraint or under the cone constraint, the modality gap converges to zero. Under the subspace constraint (i.e., representations of two modalities fall into two distinct hyperplanes due to dimension collapse), the modality gap converges to the smallest angle between the two hyperplanes. This result identifies *dimension collapse* as the fundamental origin of the modality gap. Furthermore, our theorems demonstrate that paired samples cannot be perfectly aligned under the subspace constraint. The modality gap influences downstream performance by affecting the alignment between sample pairs. We prove that, in this case, perfect alignment between two modalities can still be achieved via two ways: hyperplane rotation and shared space projection.

## 1 Introduction

Pre-trained vision–language models (VLMs) (Radford et al., 2021; Mu et al., 2022; Li et al., 2022) have achieved remarkable success across a wide range of tasks, including zero-shot image classification, zero-shot cross-modal retrieval, and visual question answering. These models are typically trained with multimodal contrastive learning on large-scale image–text pairs. Despite their strong empirical performance, our theoretical understanding of how VLMs learn representations and how these representations relate to downstream performance remains limited. In this work, we provide a theoretical study of these issues.

Our understanding of **unimodal** contrastive representation learning (Chen et al., 2020; Khosla et al., 2020) has advanced considerably. From a theoretical standpoint, when training is optimized (i.e., the training loss reaches its minimum), the learned representations converge to an optimal configuration. We refer to this process as *representational convergence* and to its limiting configuration as the *convergent optimal representation* (COR). Prior work has demonstrated that the COR of self-supervised learning (SSL) corresponds to a uniform distribution on the surface of an $h$-dimensional unit hypersphere ($\mathbb{S}^{h-1}$) (Wang & Isola, 2020). For supervised contrastive learning (SupCon), the COR forms a regular simplex inscribed in $\mathbb{S}^{h-1}$ (Graf et al., 2021), and a skewed simplex when the data is imbalanced (Yi et al., 2025b). (See additional related work in Sec. A.1.) These prior research on unimodal data demonstrate that examining the geometric and distributional properties of CORs yields critical insights into how pretraining with contrastive learning affects downstream performance.

This motivates us to investigate the COR of **multimodal** contrastive learning (MCL). Intuitively, MCL intends to align representations from different modalities in a shared embedding space. However, this is not supported by empirical evidence. Instead, representations of different modalities cluster

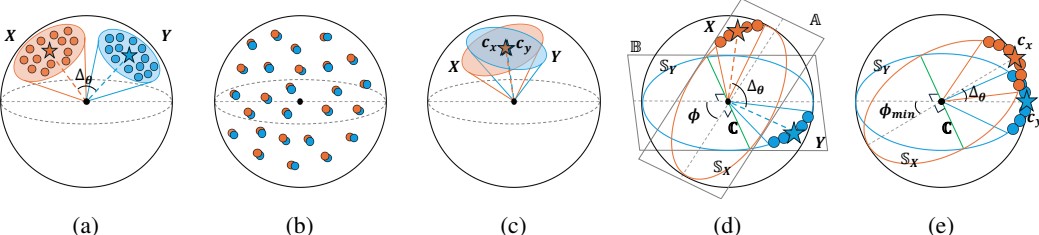

Figure 1: The COR of MCL. Orange and blue dots represent $X$ and $Y$. Starts are centers of $X$ and $Y$ (i.e., $c_x, c_y$). $\Delta_\theta$ denotes the size of modality gap. **(a)**: When a model is initialized, $(X, Y)$ are within two distinct cones. **(b)**: Without any constraint, $(X, Y)$ converge to a paired uniform distribution and $\Delta_\theta \to 0$. **(c)**: Under the cone constraint, $\Delta_\theta \to 0$. **(d)**: $(X, Y)$ collapse into two distinct subspaces $\mathbb{S}_X \in \mathbb{A}$ (orange circle) and $\mathbb{S}_Y \in \mathbb{B}$ (blue circle), respectively. $\phi$ is the angle between $\mathbb{A}$ and $\mathbb{B}$. Green line represent the shared space $\mathbb{C}$. See Definition 5 for details. **(e)**: Under the subspace constraint, when training is optimized, $c_x, c_y \perp \mathbb{C}$ and $\Delta_\theta \to \phi_{\min}$.

into disjoint cones in $\mathbb{S}^{h-1}$, forming a geometric phenomenon called the *modality gap* (Liang et al., 2022). To explain the origin of this gap, several hypotheses have been proposed, including the cone effect (Liang et al., 2022), the contrastive learning object (Fahim et al., 2024), insufficient training (Shi et al., 2023) and information bias (Schrodi et al., 2025).The impact of the modality gap on downstream performance also remains unclear. Some studies (Liang et al., 2022; Schrodi et al., 2025) show that narrowing the modality gap pos hoc may lead to degraded downstream performance, indicating that such reduction is not always beneficial. (See Sec. A.1 for more details). Prior work have mostly focused on numerical analysis. None of them has offered a satisfactory theoretical explanation of what causes the modality gap and how it affects downstream performance.

In this paper, we in turn focus on the theoretical explanation of the modality gap. We establish the first theoretical framework to systematically analyze the COR of MCL. In particular, we prove (Theorem 1) that, without any distributional constraints, representations of two modalities converge to a paired uniform distribution on $\mathbb{S}^{h-1}$ (Fig. 1b). As a result, the modality gap converges to zero. Meanwhile, the dispersion degree (i.e., how wild a distribution is spread) of the learned representation becomes infinite (Corollary 1). This shows that **the contrastive learning objective tends to close the modality gap**. However, we observe that dispersion degrees of the learned representation always remain finite in practice. Therefore, representations of each modality fall into a cone in $\mathbb{S}^{h-1}$ (Fig. 1a), a phenomenon known as *cone effect*. We prove (Theorem 2) that even under this cone constraint, the modality gap still converges to zero, regardless of the initial locations or sizes of the cones (Fig. 1c). This elucidates that **the cone effect is not the cause of the modality gap**.

The preceding analysis prompts us to ask whether there are any other geometric or distributional constraints on representations that ultimately give rise to the modality gap. Jing et al. (2022) show that the SSL learned representations collapse into a lower-dimensional subspace rather than spanning the entire embedding space, a phenomenon referred to as *dimension collapse*. Inspired by this insight, we observe that dimension collapse also arises in the MCL learned representations. We then prove (Theorem 3) that if representations of two modalities collapse into distinct hyperplanes (Fig. 1d), the modality gap converges to the smallest angle between these hyperplanes (Fig. 1e). This finding demonstrates that **the true origin of the modality gap is dimension collapse**.

That how modality gap influences downstream tasks still confuses researchers. We argue that downstream performance is determined by the alignment between all paired samples, i.e., *modality alignment*. First, we prove (Theorem 4 and Corollary 2) that when representations converge, **the mutual information between two modalities in the shared space is maximized** and in this case **paired samples cannot be perfectly aligned**. Next, we demonstrate that changes in the size of the modality gap alter the representation distribution, which in turn affects modality alignment. Then, we show that existing translation approaches, e.g., shifting image embeddings toward language embeddings by the average distance between image–language pairs, modify the representation distribution in arbitrary ways. This explains the worsen downstream performance observed when such methods are applied. Lastly, we prove derive two methods, hyperplane rotation (Corollary 3) and shared subspace projection (Corollary 4), that achieve perfect alignment and modality gap reduction without harming downstream performance. The major contributions of our work are listed below:

- We theoretically show that the contrastive learning objective tends to close the modality gap regardless of the existence of the cone effect.
- We reveal that the origin of the modality gap is dimension collapse. And under the subspace constraint, the modality gap converges to the smallest angle between two hyperplanes.
- We prove that paired samples cannot be perfectly aligned under the subspace constraint.
- We derive that perfect alignment can be achieved via hyperplane rotation or shared subspace projection.

## 2 PRELIMINARY

Suppose we have a dataset $D = \{(I_n, T_n)\}_{n=1}^N$ of $N$ image-text pairs, where $I = (i_1, \ldots, i_N) \in (\mathcal{I})^N$ and $T = (t_1, \ldots, t_N) \in (\mathcal{T})^N$. The unit hypersphere in $\mathbb{R}^h$ is defined as $\mathbb{S}^{h-1} = \{z \in \mathbb{R}^h : \|z\| = 1\}$. An image encoder $f_I(\cdot) : \mathcal{I} \to \mathbb{R}^h$ and a text encoder $f_T(\cdot) : \mathcal{T} \to \mathbb{R}^h$ map image and text data, respectively, into a shared embedding space. The resulting representations are denoted as $X = (f_I(i_1), \ldots f_I(i_N)) = (x_1, \ldots, x_N) \in (\mathbb{S}^{h-1})^N$ and $Y = (f_T(t_1), \ldots f_T(T_N)) = (y_1, \ldots, y_N) \in (\mathbb{S}^{h-1})^N$.

**Multimodal Contrastive Learning (MCL).** MCL aims to embed data from different modalities into a shared embedding space. This is achieved by minimizing the MCL loss, defined as:

**Definition 1** (Multimodal Contrastive Loss (MCL Loss)). *Let $(X, Y)$ be an $N$-pair configuration, where $X = (x_1, \ldots, x_N) \in (\mathbb{S}^{h-1})^N$ and $Y = (y_1, \ldots, y_N) \in (\mathbb{S}^{h-1})^N$. $\forall \tau > 0$, the multimodal contrastive loss $\mathcal{L}_{\mathrm{MCL}}(\cdot, \cdot) : (\mathbb{S}^{h-1})^N \times (\mathbb{S}^{h-1})^N \to \mathbb{R}$ is defined as:*

$$\mathcal{L}_{\mathrm{MCL}} = \frac{1}{N} \sum_{i=1}^N \mathcal{L}_{\mathrm{MCL}}^i, \quad where \ \mathcal{L}_{\mathrm{MCL}}^i = \mathcal{L}_{\mathcal{X} \to \mathcal{Y}}(x_i; Y) + \mathcal{L}_{\mathcal{Y} \to \mathcal{X}}(y_i; X). \tag{1}$$

*Here, $\mathcal{L}_{\mathcal{X} \to \mathcal{Y}}$ is the $\mathcal{X}$-to-$\mathcal{Y}$ alignment and $\mathcal{L}_{\mathcal{Y} \to \mathcal{X}}$ is the $\mathcal{Y}$-to-$\mathcal{X}$ alignment, defined respectively as:*

$$\mathcal{L}_{\mathcal{X} \to \mathcal{Y}}(x_i; Y) = -\log \frac{\exp(x_i \cdot y_i / \tau)}{\sum_{j=1}^N \exp(x_i \cdot y_j / \tau)}, \quad \mathcal{L}_{\mathcal{Y} \to \mathcal{X}}(y_i; X) = -\log \frac{\exp(x_i \cdot y_i / \tau)}{\sum_{j=1}^N \exp(x_j \cdot y_i / \tau)}. \tag{2}$$

In practice, contrastive learning is performed in a batch-wise manner due to memory limitations. For analytical simplicity, we assume unlimited memory to train on all samples in a single batch.

**Modality Gap.** Define $\mu_x = \frac{1}{N} \sum_{i=1}^N x_i$, $c_x = \frac{\mu_x}{\|\mu_x\|}$ as the mean and the center representation of $X$, $\mu_y = \frac{1}{N} \sum_{i=1}^N y_i$, $c_y = \frac{\mu_y}{\|\mu_y\|}$ as the mean and the center representation of $Y$.

**Definition 2** (Modality Gap). *Let $(X, Y)$ be an $N$-pair configuration, where $X = (x_1, \ldots, x_N) \in (\mathbb{S}^{h-1})^N$ and $Y = (y_1, \ldots, y_N) \in (\mathbb{S}^{h-1})^N$. The modality gap between $X$ and $Y$ can be defined as the difference between their mean representations:*

$$\Delta_\mu = \|\mu_x - \mu_y\|_2, \tag{3}$$

*or as the angle between their center representations:*

$$\Delta_\theta = \cos^{-1}(c_x \cdot c_y). \tag{4}$$

In this study, we use Eq. (4) to define the *modality gap*.

## 3 REPRESENTATIONAL CONVERGENCE AND MODALITY GAP

In this section, we study the relationship between MCL and the modality gap. To understand this, we establish a theoretical framework for analyzing the convergent optimal representations (COR) of $(X, Y)$. We prove that, with or without the cone constraint, as the MCL loss approaches its minimum, the modality gap converges to **zero**.

Both cases implicitly assume that $X$ and $Y$ are embedded in the same space $\mathbb{S}^{h-1}$. Empirical evidence, however, shows that $X$ and $Y$ tend to collapse into different subspaces. We further demonstrate that if $X$ and $Y$ lie in two distinct hyperplanes, then when the MCL loss is minimized, the modality gap converges to the **smallest angle between the two hyperplanes**.

## 3.1 VON MISES–FISHER (VMF) DISTRIBUTIONS

As shown in (Liang et al., 2022), when a model is initialized, the representations of each modality reside within a hypercone (Fig. 1a). During training, the representation distribution evolves as the size and shape of the hypercone change. The von Mises-Fisher (vMF) distribution (Mardia & Jupp, 2009),a generalization of the normal distribution on the surface of a hypersphere, also concentrates its samples within a hypercone. Hence, this distribution provides as an effective proxy for studying the geometric and distributional properties of representations learned by MCL.

**Definition 3** (vMF Distribution). $\forall c \in \mathbb{S}^{h-1}$ and $\kappa \geq 0$, the probability density of a random $h$-dimensional unit vector $z \sim \mathrm{vMF}(c, \kappa)$ is given by:

$$f_h(z; c, \kappa) = D_h(\kappa) e^{\kappa c^\top z}, \quad where \ D_h(\kappa) = \frac{\kappa^\nu}{(2\pi)^{\nu+1} I_\nu(\kappa)}. \tag{5}$$

Here, $\nu = h/2 - 1$, and $I_\nu(\cdot) : \mathbb{R} \to \mathbb{R}$ is the modified Bessel function of the first kind of order $\nu$, which is defined as:

$$I_\nu(x) = \sum_{k=0}^{\infty} \frac{1}{k! \Gamma(\nu + k + 1)} \left(\frac{x}{2}\right)^{2k+\nu}. \tag{6}$$

$c$ denotes the center vector and $\frac{1}{\kappa}$ denotes the dispersion degree. When $\frac{1}{\kappa} = \infty$, the samplesare maximally dispersed and uniformly distributed on $\mathbb{S}^{h-1}$. As $\frac{1}{\kappa}$ decreases, the samples become increasingly concentrated and cluster within a smaller hypercone. When $\frac{1}{\kappa} = 0$, the samples are fully concentrated and collapse to a single point. Throughout this work, we assume that $(X, Y)$ are $iid$ samples from two vMF distributions, that is, $x_i \sim \mathrm{vMF}(c_x, \kappa_x)$ and $y_i \sim \mathrm{vMF}(c_y, \kappa_y)$.

## 3.2 REPRESENTATIONAL CONVERGENCE WITHOUT DISTRIBUTIONAL CONSTRAINT

First, we assume that the encoders, $f_I$ and $f_T$, are sufficiently powerful, capable of realizing any representation distribution without any constraints. Theorem 1 reveals that when the limit of $\mathcal{L}_{\mathrm{MCL}}$ attains its minimum, the representations of each paired sample $(x_i, y_i)$ converge to the **same point**, while the representations of all pairs converge to the **uniform distribution** in $\mathbb{S}^{h-1}$ (Fig. 1b).

**Theorem 1.** Let $(X, Y)$ be an $N$-pair configuration, where $X = (x_1, \ldots, x_N) \in (\mathbb{S}^{h-1})^N$ are iid samples from $\mu_x$ and $Y = (y_1, \ldots, y_N) \in (\mathbb{S}^{h-1})^N$ are iid samples from $\mu_y$. Let $\nu = h/2 - 1$, it holds that:

$$\lim_{N \to \infty} \mathcal{L}_{\mathrm{MCL}} - 2\log(N) = \mathbb{E}_{x_i \sim \mu_x}\left[-\frac{x_i \cdot y_i}{\tau}\right] + \mathbb{E}_{x_i \sim \mu_x}\left[\log \mathbb{E}_{y_i \sim \mu_y}\left[\exp\left(\frac{x_i \cdot y_i}{\tau}\right)\right]\right]$$
$$+ \mathbb{E}_{y_i \sim \mu_y}\left[-\frac{x_i \cdot y_i}{\tau}\right] + \mathbb{E}_{y_i \sim \mu_y}\left[\log \mathbb{E}_{x_j \sim \mu_x}\left[\exp\left(\frac{x_i \cdot y_i}{\tau}\right)\right]\right] \tag{7}$$
$$\geq -2/\tau + 2\log\left(\Gamma(\nu + 1)(2\tau)^\nu I_\nu(1/\tau)\right),$$

where equality is attained if and only if there exists a configuration of $(X, Y)$ such that:

*(A1)* $\forall i \in [N], x_i = y_i$.

*(A2)* $\mu_x = \sigma_{h-1}$ and $\mu_y = \sigma_{h-1}$.

Here, $\sigma_{h-1}$ denotes the uniform probability measure on $\mathbb{S}^{h-1}$. The proof is provided in Sec. E.1. Under the assumption that $X$ and $Y$ are drawn from two vMF distributions, Corollary 1 implies that when the limit of $\mathcal{L}_{\mathrm{MCL}}$ attains its minimum, the modality gap converges to **zero** ($\Delta_\theta \to 0$), and both $\kappa_x$ and $\kappa_y$ converge to **zero**. This result follows directly from Theorem 1.

**Corollary 1.** Let $(X, Y)$ be an $N$-pair configuration, where $X = (x_1, \ldots, x_N) \in (\mathbb{S}^{h-1})^N$ are iid samples from $\mathrm{vMF}(c_x, \kappa_x)$, and $Y = (y_1, \ldots, y_N) \in (\mathbb{S}^{h-1})^N$ are iid samples from $\mathrm{vMF}(c_y, \kappa_y)$. $\lim_{N \to \infty} \mathcal{L}_{\mathrm{MCL}} - 2\log(N)$ attains its minimum if and only if the following conditions hold:

*(A3)* $\forall i \in [N], x_i = y_i \ (\Rightarrow \Delta_\theta = \cos^{-1}(c_x \cdot c_y) = 0)$.

*(A4)* $\kappa_x = \kappa_y = 0$.

> **Convergence 1:** Without any distributional constraints, $X$ and $Y$ converge to a paired uniform distribution on $\mathbb{S}^{h-1}$, and the modality gap converges to zero.

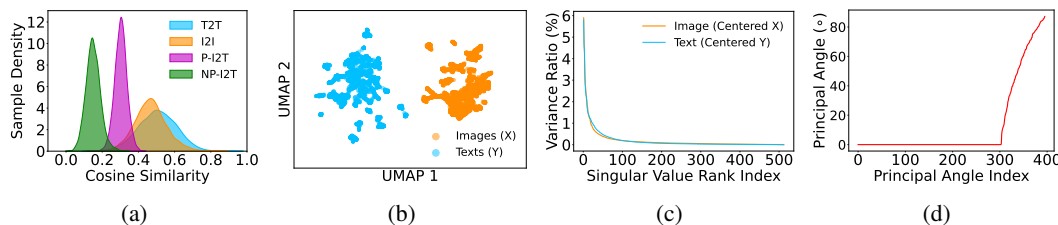

Figure 2: Distributional Constraints. CLIP ViT-B/32 embeddings of MSCOCO validation set. **(a)**: Density plot of cosine similarities between image and image (I2I), text and text (T2T), paired image and text (P I2T) and unpaired image and text (NP-I2T). **(b)**: UMAP plot. **(c)**: Explained variance ratio of singular values of $X - \mu_X$ and $Y - \mu_Y$. **(d)**: Valuse of principal angles.

### 3.3 REPRESENTATIONAL CONVERGENCE UNDER THE CONE CONSTRAINT

However, in practice, sufficiently powerful encoders are not available. Fig. 2b reveals that intra-modal similarities between two modalities are larger than inter-model similarities. Fig. 2a further shows that $(X, Y)$ separate into two clusters. Both indicate that $X$ and $Y$ lie in two hypercones on $\mathbb{S}^{h-1}$.

In this subsection, we assume that the encoders, $f_I$ and $f_T$, are powerful to the extent that $(X, Y)$ are embedded in two hypercones spanning all dimensions of $\mathbb{S}^{h-1}$, i.e., $(X, Y)$ are subject to the *cone constraint*. In this case, $\kappa_x > 0$ and $\kappa_y > 0$. Since the modality gap depends solely on the angle between the two center vectors, we focus on the configuration of $(c_x, c_y)$ and their corresponding loss terms: $\mathcal{L}_{MCL}^c = \mathcal{L}_{\mathcal{X} \rightarrow \mathcal{Y}}(c_x; Y) + \mathcal{L}_{\mathcal{Y} \rightarrow \mathcal{X}}(c_y; X)$. We first define a convergence function $\mathcal{J}$.

**Definition 4.** $\forall \kappa, \nu, \tau > 0$, a function $\mathcal{J}(\cdot; \kappa, \nu) : [-1, 1] \rightarrow \mathbb{R}$ is defined as:

$$\mathcal{J}(w; \kappa, \nu) = -\frac{w}{\tau} + \log\left(\frac{I_\nu(M_\kappa(w))}{M_\kappa(w)^\nu}\right) - \log\left(\frac{I_\nu(\kappa)}{\kappa^\nu}\right), \tag{8}$$

where the function $M_\kappa(\cdot) : [-1, 1] \rightarrow \mathbb{R}_0^+$ is defined as:

$$M_\kappa(w) = \sqrt{\kappa^2 + \frac{2\kappa w}{\tau} + \frac{1}{\tau^2}}. \tag{9}$$

Then, Theorem 2 shows that when the limit of $\mathcal{L}_{\text{MCL}}^c$ attains its minimum, the modality gap converges to **zero** ($\Delta_\theta \rightarrow 0$) (Fig. 1c).

**Theorem 2.** *Let $(X, Y)$ be an $N$-pair configuration, where $X = (x_1, \ldots, x_N) \in (\mathbb{S}^{h-1})^N$ are iid samples from $\mu_x = \text{vMF}(c_x, \kappa_x)$, and $Y = (y_1, \ldots, y_N) \in (\mathbb{S}^{h-1})^N$ are iid samples from $\mu_y = \text{vMF}(c_y, \kappa_y)$. Let $\nu = h/2 - 1$. Suppose there exists an index $i = c$ such that $x_c = c_x$, $y_c = c_y$. Denote $\Delta_\theta = \cos^{-1}(c_x \cdot c_y)$. For any fixed $\kappa_x, \kappa_y > 0$, it holds that:*

$$\lim_{N \rightarrow \infty} \mathcal{L}_{\text{MCL}}^c - 2\log(N) = \mathcal{J}(\cos(\Delta_\theta); \kappa_y, \nu) + \mathcal{J}(\cos(\Delta_\theta); \kappa_x, \nu)$$
$$\geq \mathcal{J}(1; \kappa_y, \nu) + \mathcal{J}(1; \kappa_x, \nu), \tag{10}$$

*where equality is attained if and only if there exists a configuration of $(X, Y)$ such that:*

*(A5)* $\Delta_\theta = \cos^{-1}(c_x \cdot c_y) = 0.$

The proof is provided in Sec. E.2. Since the distributions of $X$ and $Y$ are symmetric, non-center pairs $(x_i, y_i)_{i \neq c}$ do not affect the configuration of $(c_x, c_y)$, as confirmed by Theorem 4.

> **Convergence 2:** Under the cone constraint, the modality gap still converges to zero.

### 3.4 REPRESENTATIONAL CONVERGENCE UNDER THE SUBSPACES CONSTRAINT

To investigate whether $X$ and $Y$ collapse into subspaces of $\mathbb{S}^{h-1}$, we plot singular values $\sigma_i$ of the centered $X$ and the centered $Y$ in Fig. 2c. Zero $\sigma_i$s confirm dimension collapse. Fig. 2d shows the principal angles $\gamma_i$ between the subspaces where $X$ and $Y$ collapse. Zero $\gamma_i$s imply that the two subspaces share overlapped dimensions. Detailed explanations are provided in Sec. C.1 and Sec. C.2.

In this subsection, we assume that the encoders, $f_I$ and $f_T$, embed $(X, Y)$ into two partially overlapping subspaces of $\mathbb{S}^{h-1}$ (Fig. 1d), i.e., $(X, Y)$ are subject to the *subspace constraint*. To simplify the analysis, we require that the two subspaces are hyperplanes, as described below:

**Definition 5.** *Let $\mathbb{A}$ and $\mathbb{B}$ be two distinct $(h-1)$-dimensional linear subspaces (i.e., hyperplanes through the origin) with normal vectors $n_A$ and $n_B$, projection matrices $P_A$ and $P_B$. Denote $\mathbb{C} = \mathbb{A} \cap \mathbb{B}$, with $P_C$ as its projection matrix. Define $\phi = \cos^{-1}\left(\frac{n_A \cdot n_B}{\|n_A\| \cdot \|n_B\|}\right)$ as the angle between $\mathbb{A}$ and $\mathbb{B}$, restricted to $0 < \phi_{\min} \le \phi < \frac{\pi}{2}$. Then, $\mathbb{S}_X$ and $\mathbb{S}_Y$ can be represented as:*

$$
\begin{aligned}
\mathbb{S}_X &= \mathbb{S}^{h-1} \cap \mathbb{A} = \left\{ x \in \mathbb{R}^h : \|x\| = 1, n_A \cdot x = 0 \right\} \cong \mathbb{S}^{h-2} \in \mathbb{S}^{h-1}, \\
\mathbb{S}_Y &= \mathbb{S}^{h-1} \cap \mathbb{B} = \left\{ y \in \mathbb{R}^h : \|y\| = 1, n_B \cdot y = 0 \right\} \cong \mathbb{S}^{h-2} \in \mathbb{S}^{h-1}.
\end{aligned}
\tag{11}
$$

$\mathbb{C}$ is an $(h-2)$ dimensional linear subspace (Strang, 2022). We now define a convergence function $\tilde{\mathcal{J}}$. Note that function $\mathcal{J}$ in Definition 4 is a special case of $\tilde{\mathcal{J}}$ with $\mathcal{J}(w; \kappa, \nu) = \tilde{\mathcal{J}}(w, w, 1; \kappa, \nu)$.

**Definition 6.** $\forall \kappa, \nu, \tau > 0$, $\tilde{\mathcal{J}}(\cdot, \cdot, \cdot; \kappa, \nu) : [-1, 1] \times [-1, 1] \times [0, 1] \to \mathbb{R}$ *is defined as:*

$$
\tilde{\mathcal{J}}(w_1, w_2, t; \kappa, \nu) = -\frac{w_1}{\tau} + \log\left(\frac{I_\nu\left(\tilde{M}_\kappa(w_2, t)\right)}{\tilde{M}_\kappa(w_2, t)^\nu}\right) - \log\left(\frac{I_\nu(\kappa)}{\kappa^\nu}\right),
\tag{12}
$$

*where the function $\tilde{M}_\kappa(\cdot, \cdot) : [-1, 1] \times [0, 1] \to \mathbb{R}_0^+$ is defined as:*

$$
\tilde{M}_\kappa(w, t) = \sqrt{\kappa^2 + \frac{2\kappa w}{\tau} + \frac{t^2}{\tau^2}}.
\tag{13}
$$

Theorem 3 shows that when the limit of $\mathcal{L}_{\mathrm{MCL}}^c$ attains its minimum, $c_x, c_y$ are orthogonal to $\mathbb{C}$, and the modality gap converges to the **smallest angle between $\mathbb{A}$ and $\mathbb{B}$** ($\Delta_\theta \to \phi_{\min}$) (Fig. 1e).

**Theorem 3.** *Let $(X, Y)$ be an $N$-pair configuration, where $X = (x_1, \ldots, x_N) \in (\mathbb{S}_X \setminus \mathbb{C})^N$ are iid samples from $\mu_x = \mathrm{vMF}(c_x, \kappa_x)$, and $Y = (y_1, \ldots, y_N) \in (\mathbb{S}_Y \setminus \mathbb{C})^N$ are iid samples from $\mu_y = \mathrm{vMF}(c_y, \kappa_y)$. Let $\tilde{\nu} = (h-1)/2 - 1$. Suppose there exists an index $i = c$ such that $x_c = c_x$, $y_c = c_y$. Denote $\Delta_\theta = \cos^{-1}(c_x \cdot c_y)$ and assume that $c_x, c_y \notin \mathbb{C}$ with $c_x \cdot c_y > 0$. For any fixed $\kappa_x, \kappa_y > 0$, it holds that:*

$$
\lim_{N \to \infty} \mathcal{L}_{\mathrm{MCL}}^c - 2\log(N)
$$

$$
= \tilde{\mathcal{J}}(\cos(\Delta_\theta), \cos(\Delta_\theta), \|P_B c_x\|; \kappa_y, \tilde{\nu}) + \tilde{\mathcal{J}}(\cos(\Delta_\theta), \cos(\Delta_\theta), \|P_A c_y\|; \kappa_x, \tilde{\nu})
$$

$$
\ge \tilde{\mathcal{J}}(\cos(\phi_{\min}), \cos(\phi_{\min}), \cos(\phi_{\min}); \kappa_y, \tilde{\nu}) + \tilde{\mathcal{J}}(\cos(\phi_{\min}), \cos(\phi_{\min}), \cos(\phi_{\min}); \kappa_x, \tilde{\nu}),
\tag{14}
$$

*where equality is attained if and only if there exists a configuration of $(X, Y)$ such that:*

*(A6)* $c_x \perp \mathbb{C}$ *and* $c_y \perp \mathbb{C}$ *($\Rightarrow \Delta_\theta = \phi$).*

*(A7)* $\Delta_\theta = \cos^{-1}(c_x \cdot c_y) = \phi_{\min}$.

The proof is provided in Sec. E.3. Condition (A6) shows the optimal configuration of $(c_x, c_y)$ for any given $\phi$. Condition (A7) establishes that the loss decreases monotonically as $\phi$ decreases to $\phi_{\min}$. Since the distributions of $X$ and $Y$ are symmetric, non-center pairs $(x_i, y_i)_{i \ne c}$ do not affect Condition (A6). Moreover, optimizing of $\mathcal{L}_{\mathrm{MCL}}^{i \ne c}$ also yields Condition (A7), as shown in Theorem 4.

> **Convergence 3:** Under the subspace constraint, the modality gap converges to the smallest angle between the two hyperplanes.

## 4 REPRESENTATIONAL CONVERGENCE AND MODALITY ALIGNMENT

In Sec. 3.4, we identified the true origin of the modality gap by analyzing the configuration of the center pair. However, the relationship between the modality gap and downstream performance, which depends on the configuration of all pairs, remains unclear. In this section, we show that, under the subspace constraint, non-center pairs cannot be perfectly aligned when the MCL loss is minimized.

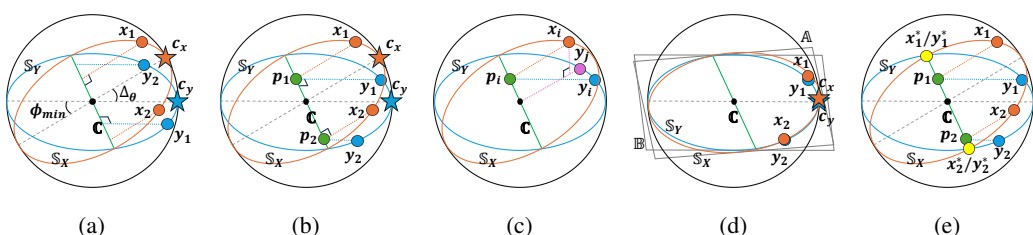

|  |  |  |  |  |
|---|---|---|---|---|
| (a) | (b) | (c) | (d) | (e) |

Figure 3: Modality Alignment. Notations follow Fig. 1. **(a)**: Condition (A6) $(c_x, c_y \perp \mathbb{C})$ and IMS $(x_i \cdot c_x = y_i \cdot c_y)$ hold. **(b)**: The projections of $(x_i, y_i)_{i \neq c}$ on $\mathbb{C}$ converge to $p_i$ (green point), i.e., $P_C x_i = P_C y_i = p_i$. **(c)**: When Condition (A6) and (A8) $(P_C x_i = P_C y_i)$ hold, $P_B x_i \nparallel y_i$, $P_A y_i \nparallel x_i$. Denote $y_j = \frac{P_B x_i}{\|P_B x_i\|}$ (purple dot), then $x_i \cdot y_j > x_i \cdot y_i$ and $(x_i, y_i)_{i \neq c}$ are not perfectly aligned. **(d)**: Rotating $X$ with the hyperplane $\mathbb{A}$ towards $\mathbb{B}$, $X$ and $Y$ can be aligned perfectly. **(c)**: Project $x_i$ and $y_i$ onto $\mathbb{C}$ and re-normalize, then $x_i^*$ and $y_i^*$ (yellow dots) are perfectly aligned.

### 4.1 Intra-Modal Isometry and Perfect Alignment

The Platonic Representation Hypothesis (Huh et al., 2024) suggests that contrastive learners are optimized by representations of $X$ and $Y$ whose intra-modal kernels (i.e., pairwise similarities) align. Building on this idea, we define the kernel alignment as *Intra-Modal Isometry*.

**Definition 7** (Intra-Modal Isometry (IMS)). *Let $(X, Y)$ be an $N$-pair configuration in $\mathbb{R}^h$, we say $(X, Y)$ achieves Intra-Modal Isometry if and only if $\forall i, j \in [N], i \neq j, x_i \cdot x_j = y_i \cdot y_j$.*

The Intra-Modal Isometry assumption implies that $\forall i \in [N], x_i \cdot c_x = y_i \cdot c_y$, and thus $\kappa_x = \kappa_y$ (Fig. 3a). However, knowledge of the intra-modal configuration alone is insufficient to determine how the modality gap affects downstream performance. In downstream tasks such as zero-shot image classification, given an input from one modality (e.g., $x_i$), CLIP retrieves data from the other modality (e.g., $y_j$) with the largest similarity to the input. Ideally, the output should be $y_j = y_i$. We therefore define an ideal inter-modal configuration as *Perfect Alignment*. And when Perfect Alignment is achieved, downstream performance is maximized.

**Definition 8** (Perfect Alignment). *Let $(X, Y)$ be an $N$-pair configuration in $\mathbb{R}^h$, we say $(x_i, y_i)$ is perfectly aligned if and only if $\forall j \neq i, x_i \cdot y_i > x_i \cdot y_j$ and $x_i \cdot y_i > x_j \cdot y_x$ If $\forall i \in [N], (x_i, y_i)$ is perfectly aligned, we say $(X, Y)$ achieves Perfect Alignment.*

### 4.2 Representational Convergence of Non-Center Pairs

To investigate the alignment between two modalities, we examine the optimal configuration of each data pair. Theorem 4 states that if Condition (A6) (in Theorem 3) is satisfied through the optimization of $\mathcal{L}_{\text{MCL}}^{i=c}$, and if $(X, Y)$ achieves Intra-Modal Isometry (Fig. 3a), then when the limit of $\mathcal{L}_{\text{MCL}}^{i \neq c}$ attains its minimum, the projections of any non-center pair $(x_i, y_i)_{i \neq c}$ onto $\mathbb{C}$ converge to the same vector (Fig. 3b).

**Theorem 4.** *Let $(X, Y)$ be an $N$-pair configuration, where $X = (x_1, \ldots, x_N) \in (\mathbb{S}_X \setminus \mathbb{C})^N$ are iid samples from $\mu_x = \text{vMF}(c_x, \kappa_x)$, and $Y = (y_1, \ldots, y_N) \in (\mathbb{S}_Y \setminus \mathbb{C})^N$ are iid samples from $\mu_y = \text{vMF}(c_y, \kappa_y)$. Let $\tilde{\nu} = (h-1)/2 - 1$. Denote $\Delta_\theta = \cos^{-1}(c_x \cdot c_y)$ and assume $c_x, c_y \perp \mathbb{C}$ with $c_x \cdot c_y > 0$. Suppose $(X, Y)$ achieves Intra-Modal Isometry. Then $\forall i \in [N]$, denote $\theta_i^c = \cos^{-1}(x_i \cdot c_x) = \cos^{-1}(y_i \cdot c_y)$, and $\kappa = \kappa_x = \kappa_y$. Let $\theta_i^c \in (0, \frac{\pi}{2})$ and $\kappa > 0$, it holds that:*

$$\lim_{N \to \infty} \mathcal{L}_{\text{MCL}}^{i \neq c} - 2\log(N)$$

$$= \tilde{\mathcal{J}}\left(\cos(\Delta_\theta), \cos(\theta_i^c), \|P_B x_i\|; \kappa, \tilde{\nu}\right) + \tilde{\mathcal{J}}\left(\cos(\Delta_\theta), \cos(\theta_i^c), \|P_A y_i\|; \kappa, \tilde{\nu}\right)$$

$$\geq 2\tilde{\mathcal{J}}\left(\cos^2(\theta_i^c)\cos(\phi_{\min}) + \sin^2(\theta_i^c), \cos(\theta_i^c), \sqrt{\cos^2(\theta_i^c)\cos^2(\phi_{\min}) + \sin^2(\theta_i^c)}; \kappa, \tilde{\nu}\right),$$

(15)

*where equality is attained if and only if there exists a configuration of $(X, Y)$ such that:*

*(A8) $P_C x_i = P_C y_i$.*

*(A9)* $\Delta_\theta = \cos^{-1}(c_x \cdot c_y) = \phi_{\min}$.

The proof of Theorem 4 is provided in Sec. E.4. Condition (A8) characterizes the optimal configuration of $(x_i, y_i)_{i \neq c}$ for any given $\phi$. Condition (A9) establishes that the loss decreases monotonically as $\phi$ decreases to $\phi_{\min}$, consistent with Condition (A7) of Theorem 3. Moreover, Theorem 4 implies that MCL aims to **maximize the mutual information between the two modalities in the shared space while preserving modality-specific information in the complementary space**.

### 4.3 Representational Convergence Dose Not Ensure Perfect Alignment

In Lemma 12, we show that $(x_i, y_i)_{i \neq c}$ are perfectly aligned if and only if the projections of $(x_i, y_i)_{i \neq c}$ onto $\mathbb{B}$ and $\mathbb{A}$ are collinear, i.e., $P_B x_i \parallel y_i$ and $P_A y_i \parallel x_i$. However, when training is optimized such that conditions (A6) and (A8) hold, $P_B x_i \nparallel y_i$ and $P_A y_i \nparallel x_i$. This implies that $(x_i, y_i)_{i \neq c}$ are **not perfectly aligned** (Fig. 3c).

**Corollary 2.** $\forall i \in [N], i \neq c$, if $c_x, c_y \perp \mathbb{C}$ and $P_C x_i = P_C y_i \neq \vec{0}$ and $\phi > 0$, then it holds:

*(A10)* $(x_i, y_i)_{i \neq c}$ *are not perfectly aligned.*

The proof of Corollary 2 is provided in Sec. E.4.3. Since the limit of $\mathcal{L}_{\mathrm{MCL}}$ attains its minimum when both $\mathcal{L}_{\mathrm{MCL}}^c$ and $\mathcal{L}_{\mathrm{MCL}}^{i \neq c}$ attain their minima, and since all paired samples are non-center pairs almost surely (the 'center' forms a zero measure set in $\mathbb{S}_X$ or $\mathbb{S}_Y$), then we conclude that:

> **Convergence 4:** Under the subspace constraint, paired samples cannot be perfectly aligned.

## 5 Shared Subspace Projection Improves Modality Alignment

In Sec. 4, we prove that the representations of paired samples are not perfectly aligned. Despite this undesirable configuration, in this section we derive potential methods to improve the alignment between the two modalities.

### 5.1 How to Achieve Perfect Alignment

In downstream tasks, when $(x_i, y_i)_{i \neq c}$ are not perfectly aligned, $x_i$ can be misaligned to some $y_{j \neq i}$ (Fig. 3c). A straightforward way to address this is to manually shift $(x_i, y_i)$ in $\mathbb{S}^{h-1}$. For example, Liang et al. (2022) translate $x_i$ toward $y_i$ as $x_i^{\mathrm{new}} = x_i + \Delta_u$, followed by renormalization. This operation clearly alters the distributions of $X$. Since downstream performance depends on the number of misaligned $y_j$ in the test set. A change in the distribution of $X$ leads to a change in the proportion of misaligned $y_j$, but in an unpredictable direction. Therefore, the impact of translating $X$ on downstream performance can be arbitrary. An illustrative example is provided in Sec. B.1.

As shown in Fig. 3d, if we rotate $\mathbb{A}$ to overlap with $\mathbb{B}$, then $\mathbb{A} = \mathbb{B} = \mathbb{C}$. In this case, Condition (A8) implies $x_i = y_i$, and thus $x_i$ and $y_j$ are **perfectly aligned**. Hence, modality alignment can be improved by rotating the hyperplanes $\mathbb{A}$ and $\mathbb{B}$ until $\mathbb{A} = \mathbb{B}$ ($\Delta_\theta = \phi = 0$).

**Corollary 3.** $\forall i \in [N], i \neq c$, if $c_x, c_y \perp \mathbb{C}$, $P_C x_i = P_C y_i$ and $(x_i, y_i)_{i \neq c} \in \mathbb{S}^{h-1} \setminus \mathbb{C}$, then $(x_i, y_i)_{i \neq c}$ are perfectly aligned if the following condition holds:

*(A11)* $\Delta_\theta = \phi = 0$.

The proof of Corollary 3 is provided in Sec. E.4.3. Despite this theoretical guarantee, rotating a high-dimensional hyperplane can be complicated in practice. As illustrated in Fig. 3e, if we project $x_i$ and $y_i$ onto $\mathbb{C}$ and then renormalize, we obtain $x_i^* = y_i^*$. And $(x_i^*, y_i^*)$ are **perfectly aligned**.

**Corollary 4.** $\forall i \in [N], i \neq c$, if $c_x, c_y \perp \mathbb{C}$ and $P_C x_i = P_C y_i$, then the following holds:

*(A12)* $\left( \frac{P_C x_i}{\| P_C x_i \|}, \frac{P_C y_i}{\| P_C y_i \|} \right)_{i \neq c}$ *are perfectly aligned*

The proof of Corollary 4 is provided in Sec. E.4.3. Note that in Fig. 3e, $\mathbb{C}$ is a 1D line, so all transformed paired samples overlap at $y_1^*$ and $y_2^*$. In practice, however, the dimension of $\mathbb{C}$ is typically greater than 1 (e.g., 212D for MS-COCO dataset). For instance, in the 4D example in Fig. 6 of Sec. B.2, $\mathbb{C}$ is a 2D plane, and the samples are distributed along a unit circle.

Table 1: Size of $\theta_\Delta$ and accuracies (%) of zero-shot image classification of ViT-B/32.

| Model | CIFAR-10 | | | CIFAR-100 | | | ImageNet-1K | | |
|---|---|---|---|---|---|---|---|---|---|
| | $\Delta_\theta$ | R1 | R5 | $\Delta_\theta$ | R1 | R5 | $\Delta_\theta$ | R1 | R5 |
| CLIP | 74.69° | 89.00 | 99.36 | 74.19° | 65.23 | 88.88 | 71.02° | 63.34 | 88.82 |
| CLIP + Translation | 7.02° | 80.97 | 96.09 | 30.50° | 54.46 | 77.25 | 51.68° | 60.37 | 86.93 |
| CLIP + Removal | 72.5° | 14.91 | 56.22 | 73.16° | 16.82 | 6.44 | 69.71° | 49.50 | 78.55 |
| CLIP + SSP | **5.37°** | **86.43** | **99.27** | **30.39°** | **64.51** | **88.79** | **50.40°** | **62.45** | **88.41** |

## 5.2 EXPERIMENT

Theorem 4, Corollary 3 and Corollary 4 suggest that if projections of $X$ and $Y$ are aligned in the shared space, modality alignment can be improved. This also indicates that the modality gap can be reduced pos hoc without harming downstream performance.

**Method.** Following Corollary 4, we apply the shared space projection (SSP) method pos hoc to improve the alignment of the modality. Detailed procedures are described in Sec. C.3.

**Modality Alignment.** To validate the effectiveness of our method, we start by visualizing $X$ and $Y$ after applying SSP. We first project $X$ and $Y$ onto an estimated shared space of 212 dimensions. Fig. 4a (vs. Fig. 2a) shows the cosine similarities of the projected $X$ and $Y$. It indicates that our method improves both intermodal alignment (larger P-I2T) and and intramodal uniformity (smaller T2T and I2I). Since the shared space is not estimated from the original training data, the estimation can be noisy. Hence, we select a 10 dimensional subspace of the estimated shared space to reduce the estimation error (details explained in Sec. C.3). We

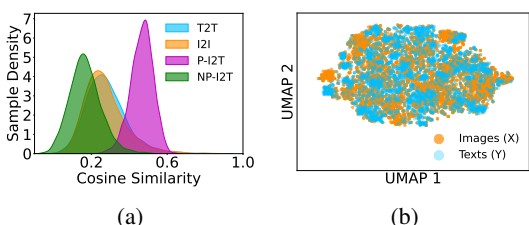

(a)        (b)

Figure 4: Results. CLIP ViT-B/32 embeddings of MSCOCO validation set after applying SSP are used. **(a)**: UMAP plot. **(b)**: Density plot of cosine similarities.

project $X$ and $Y$ onto this subspace. Fig. 4b (vs. Fig. 2b) shows that the projected $X$ and $Y$ are no longer in separate clusters.

**Zero-Shot Image Classification.** We also test our method in the zero-shot image classification task on various datasets. Details of this experiment are provided in Sec. D.1. Our goal is to reduce the size of the modality gap as much as possible without harming downstream performance. In Tab. 1, we list results of the size of the modality gap ($\Delta_\theta$), the top-1 accuracy (R1), and the top-5 accuracy (R5). We include two baseline methods: a translation-based approach (Liang et al., 2022) and a dimension-removal approach (Schrodi et al., 2025). Our results show that our method outperforms these baselines by achieving a greater reduction in the modality gap while maintaining comparable downstream performance prior to the post hoc operation. Despite its advantages, our method does not lead to improved downstream performance, as indicated in Corollary 4. We argue that this limitation arises because the intra-model isometry assumption does not hold in CLIP. Prior work has shown that CLIP's vision and text spaces exhibit different neighborhood structures (Udandarao, 2022; Schrodi et al., 2025). We provide additional experiments of **Zero-Shot Cross-Modal Retrieval** in Sec. D.2.

## 6 CONCLUSION

Our work comprehensively investigates two key questions: (1) What causes the modality gap? (2) How does it affect downstream tasks? Our theorems identify *dimension collapse* as the fundamental origin of the modality gap. Our theorems also demonstrate that paired samples cannot be perfectly aligned under the subspace constraint. We further prove that two approaches, hyperplane rotation and shared space projection, can achieve perfect alignment between two modalities. We apply the latter approach post-hoc and validate its effectiveness in downstream tasks. Besides the pos hoc application, our method has potential to be applied to pretraining. It can directly optimize modality alignment in the shared space to achieve the intra-modal isometry. We will explore it in the next step.

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

# A  APPENDIX A: MORE DISCUSSIONS

## A.1  RELATED WORK

Due to the page limit of the initial submission (9 pages), we include the related work here. In the final version (10 pages), this section will be moved into the main text.

### A.1.1  REPRESENTATION LEARNING AND REPRESENTATIONAL CONVERGENCE

Unimodal representations can be learned in an unsupervised manner using self-supervised contrastive learning (SSL) (Chen et al., 2020). When the InfoNCE loss (Wu et al., 2018) reaches its minimum, the representations of differently augmented views of an image converge to a single point, and the representation of all images converge to a uniform distribution on $\mathbb{S}^{h-1}$ (Wang & Isola, 2020). However, Jing et al. (2022) empirically shows that this theoretical optimum may not be realized in practice: the learned representations tend to collapse into a lower-dimensional subspace rather than spanning the entire embedding space.

In the supervised setting, representations can be learned through a neural classifier. When the cross-entropy loss is minimized, representations of samples from different balanced classes converge to the vertices of a regular simplex inscribed in $\mathbb{S}^{h-1}$, a phenomenon known as *neural collapse* (Papyan et al., 2020). Graf et al. (2021) provide a theoretical explanation of this phenomenon. Representations can also be learned with supervised contrastive learning (SupCon) (Khosla et al., 2020). Graf et al. (2021) prove that the COR of a balanced dataset of SupCon also forms a regular simplex. Yi et al. (2025b) provide a refined proof and further show that, for imbalanced datasets, representations converge to a skewed simplex or even collapse into two distinct points. Other works extend the concept of neural collapse to semi-supervised learning (Yi et al., 2025a) and OOD detection (Liu & Qin, 2025).

Multimodal representations are learned through multimodal contrastive learning (MCL). However, the COR of MCL remains poorly understood. In this work, we address this gap by characterizing the COR of MCL. Our theorems suggest that MCL seeks to maximize the mutual information between the two modalities in the shared space while preserving modality-specific information in the complementary space.

### A.1.2  MODALITY GAP

Liang et al. (2022) first identified the modality gap, a geometric phenomenon characterized by the complete separation of representations of different modalities in the embedding space. They hypothesize that the gap arises from the cone effect due to random model initialization and is preserved by the contrastive learning objective. Fahim et al. (2024) argues that the modality gap is inherent to contrastive loss. Yaras et al. (2024); Udandarao (2022) examine the role of mismatch pairs and the temperature parameter. Shi et al. (2023) attribute the cause of the modality gap to insufficient training. Schrodi et al. (2025) suggests that problematic training data, which contain information bias, create the gap. Most of these works validate their hypotheses through numerical examples on a small number of data pairs. By contrast, we provide an analysis based on the entire distribution.

In addition, several studies have proposed post-hoc methods to mitigate the modality gap. Liang et al. (2022) attempts to translate the representations of one modality toward those of another using a constant shift. Schrodi et al. (2025) explores removing the few dimensions that primarily drive the modality gap. However, experiments in both works reported that narrowing the modality gap pos hoc may lead to degraded downstream performance. Eslami & de Melo (2025) mitigates the modality gap by retraining CLIP from scratch. Our work focuses on training-free pos-hoc plug-and-play methods that can directly leverage existing pre-trained models.

## A.2  LIMITATIONS

While our work investigates the origin of the modality gap and attributes it to dimension collapse, we do not address the exact factors that lead to dimension collapse. (Jing et al., 2022) theoretically show that dimension collapse occurs whenever negative eigenvalues appear in the weight matrix of a neural network. (Schrodi et al., 2025) suggests that when training data with information bias are sufficiently

aligned, 'more dimensions' are required to focus on objects and and 'less dimensions' to focus on attributes, ultimately resulting in dimension collapse. (Chun, 2025) provides a more comprehensive study of the inherent challenges within MCL, including intra-modal variability, asymmetries in information, and task-dependent alignment. We suspect that all these factors contribute to dimension collapse in the learned representations. Identifying the causes of dimension collapse thus constitutes a major open problem, parallel to understanding the origin of the modality gap, and represents an important direction for future research.

### A.3 CONNECTION BETWEEN OUR THEOREMS AND PREVIOUS HYPOTHESES

In this subsection, we examine the connection between empirical observations from prior studies and our theoretical conclusions.

**Cone Effect:** The cone effect hypothesis (Liang et al., 2022) posits that the representations of $X$ and $Y$ fall into different cones on the hypersphere, thereby causing the modality gap. In our theoretical framework, as described in Sec. 3.1, the cone size of the representations is modeled by the parameter $\kappa$. However, in contrast to this hypothesis, Theorem 2 shows that the cone size has no effect on the convergence of the modality gap, even when the representations follow a uniform distribution (i.e., $\kappa \to 0$).

**Temperature:** It is hypothesized that the choice of temperature contributes to the emergence of the modality gap (Yaras et al., 2024; Udandarao, 2022). However, Theorem 2 suggests that the temperature parameter, $\tau$, has no effect on the convergence of the modality gap. We suspect that if temperature has any impact, it operates indirectly by influencing dimension collapse.

**Information Bias:** (Schrodi et al., 2025) argue that information bias, i.e., images containing more information than the corresponding text, leads to the modality gap. The unequal amount of information across modalities prevents Intra-Modal Isometry of the representations (see Definition 7), making it difficult for the model to align representations from the two modalities. This results in sub-optimal inter-modal alignment, which in turn imposes a lower bound on the alignment terms and ensures $\Delta_\theta > 0$. We posit that there is a strong connection between information bias and dimension collapse: information bias induces dimension collapse in the learned representations, thereby causing the modality gap.

### A.4 DISCLOSURE OF LLM USAGE

In the preparation of this paper, we used large language models (LLMs) as general-purpose assistive tools. Specifically, we used an LLM to help with grammar polishing, wording improvements, and proof-reading.

Any text or content generated by the LLM have been reviewed and edited by the authors. We take full responsibility for the content of the submission. The LLM was not used to produce novel research claims, data analysis, results formulation, or conclusions. The research ideation, theoretical contributions, experiments, and all core technical work are entirely the work of the authors.

## B APPENDIX B: MORE SUPPORTING EXAMPLES

In this subsection, we provide more examples and illustrations.

### B.1 ILLUSTRATIVE EXAMPLE OF TRANSLATION-BASED METHOD

In this subsection, we provide an illustrative example showing that the impact of translating $X$ pos hoc on downstream performance can be arbitrary. Fig. 5a depicts a set of $X$ and $Y$ where Condition (A6) and Condition (A8) hold. Fig. 5b illustrates how $X$s are going to be translated. Fig. 5c shows the positions of $X^*$s after translation. Fig. 5d illustrates how $X^*$s are going to be normalized. Fig. 5e shows the positions of $X^{**}$s after normalization. In Fig. 5f, we observe that the distribution of $X^{**}$s differs substantially from that of $X$: they no longer reside in the same shared space (the large circle in this example), and their projections onto the shared space diverge from those of $Y$s. The direction of these changes depends on the specific configuration of $X$ and is therefore unpredictable. Hence,

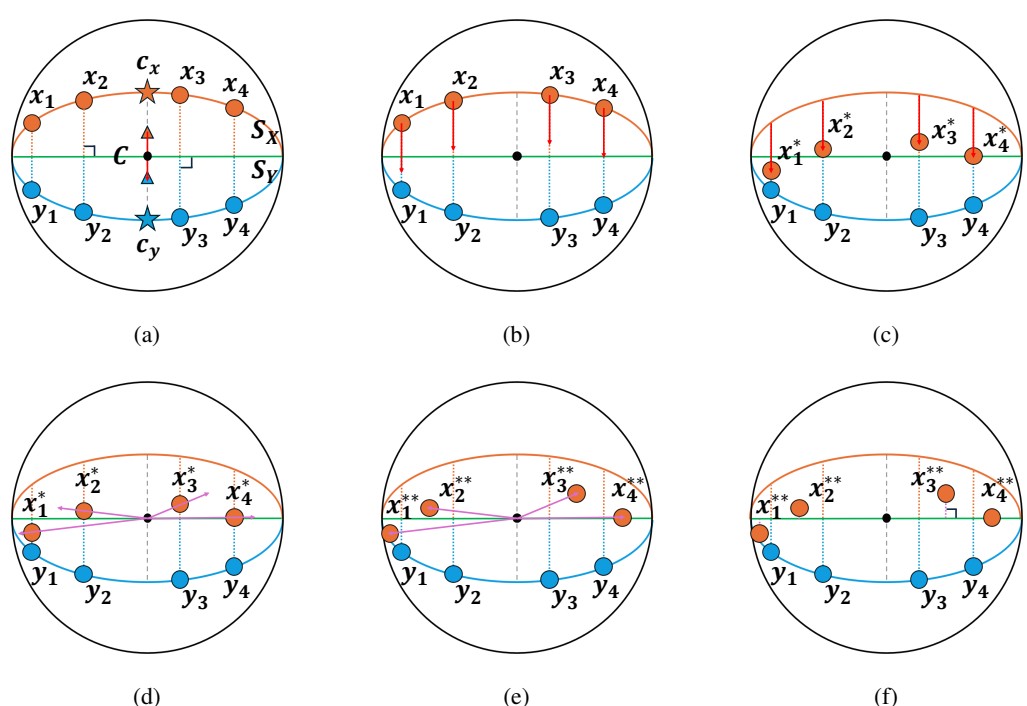

Figure 5: Translation-based method. Notations follow Fig. 1. **(a)**: Condition (A6) $(c_x, c_y \perp \mathbb{C})$ and (A8) $(P_C x_i = P_C y_i)$ hold. Orange/blue triangles represents $\mu_x$ and $\mu_y$. The red arrows denotes the direction and scale of the constant translation $(\mu_y - \mu_x)$. **(b)**: Translating $X$. **(c)**: $X$ are translated to $X^*$. **(d)**: $X^*$ are being re-normalized. Purple arrows are denotes the direction and scale of the normalization. **(e)**: $X^*$ are re-normalized to $X^{**}$. **(f)**: Distribution of $X$ altered after translation with $P_C x_i^{**} \neq P_C y_i$.

the impact of translating $X$ on downstream performance is unpredictable. In practice, the impact is often a negative one.

## B.2 ADDITIONAL EXAMPLE OF MODALITY ALIGNMENT

In Sec. 5, we discuss how the shared space projection approach can improve modality alignment. As an illustrative case, Fig. 3e presents an example in a $3D$ embedding space where $\mathbb{C}$ corresponds to a 1D line. However, this example may be misinterpreted as implying that all transformed paired samples $(x_i^*, y_i^*)$ perfectly overlap at $y_1*$ and $y_2*$. To clarify this point, in this subsection we examine a more intricate example in a 4D embedding space.

First, recall Definition 5 and set $h = 4$:

**Definition 5** [Restate with $h = 4$] Let $\mathbb{A}$ and $\mathbb{B}$ be two distinct $(h-1)$-dimensional linear subspaces (i.e., hyperplanes through the origin) with normal vectors $n_A$ and $n_B$, projection matrices $P_A$ and $P_B$. Denote $\mathbb{C} = \mathbb{A} \cap \mathbb{B}$, with $P_C$ as its projection matrix. Define $\phi = \cos^{-1}\left(\frac{n_A \cdot n_B}{\|n_A\| \cdot \|n_B\|}\right)$ as the angle between $\mathbb{A}$ and $\mathbb{B}$, restricted to $0 < \phi_{\min} \leq \phi < \frac{\pi}{2}$. Then, $\mathbb{S}_X$ and $\mathbb{S}_Y$ can be represented as:

$$\begin{aligned} \mathbb{S}_X = \mathbb{S}^3 \cap \mathbb{A} = \left\{x \in \mathbb{R}^4 : \|x\| = 1, n_A \cdot x = 0\right\} \cong \mathbb{S}^2 \in \mathbb{S}^3, \\ \mathbb{S}_Y = \mathbb{S}^3 \cap \mathbb{B} = \left\{y \in \mathbb{R}^4 : \|y\| = 1, n_B \cdot y = 0\right\} \cong \mathbb{S}^2 \in \mathbb{S}^3. \end{aligned} \tag{16}$$

Now, $\mathbb{S}^3$ denotes the 4D unit hypersphere. To analyze this case, we decompose the embedding space. Let $\{e_1, e_2, e_3, e_4\}$ be an orthonormal basis of $\mathbb{R}^4$. Suppose that the shared space $\mathbb{C}$ lies within the span of $e_1$ and $e_2$:

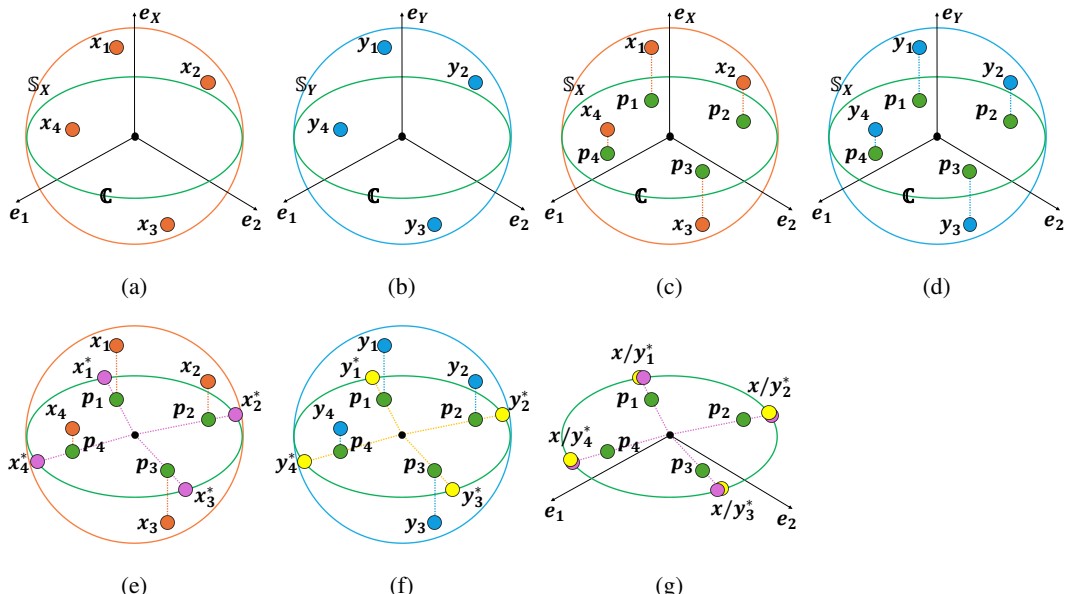

Figure 6: Modality Alignment in 4D Space. $\mathbb{S}_X$ (orange circle) and $\mathbb{S}_Y$ (blue circle) are two $3D$ unit sphere within $\mathbb{S}^3$ as described in Eq. (16) and Eq. (22). The shared space $\mathbb{C}$ is a 2D plane as described in Eq. (17). $\mathbb{S}_X \cap \mathbb{C} = \mathbb{S}_Y \cap \mathbb{C}$ is a 2D circle (green circle). **(a)** 4 samples from $X$ (orange dots). **(b)** 4 samples from $Y$ (blue dots). **(c), (d)**: The projections of $(x_i, y_i)_{i \neq c}$ on the shared space $\mathbb{C}$ converge to $p_i$ (green point), i.e., $P_C x_i = P_C y_i = p_i$. **(e), (f)**: Re-normalize $p_i$ to get $x_i^*$ (purple dots) and $y_i^*$ (yellow dots) as described in Eq. (23). **(g)**: $(x_i^*, y_i^*)$ are perfectly aligned.

$$\mathbb{C} = \operatorname{span}\{e_1\} \oplus \operatorname{span}\{e_2\}. \tag{17}$$

Therefore, $\mathbb{C}^\perp$ is a 2-dimensional orthogonal complement of $C$, and $\mathbb{C}^\perp$ satisfies:

$$\mathbb{C}^\perp = \operatorname{span}\{e_3\} \oplus \operatorname{span}\{e_4\},$$
$$\mathbb{R}^h = \mathbb{C} \oplus \mathbb{C}^\perp. \tag{18}$$

Define two unit vectors $e_X$ and $e_Y$ such that:

$$e_X \in \mathbb{S}_X, \text{ and } e_X \perp \mathbb{C},$$
$$e_Y \in \mathbb{S}_Y, \text{ and } e_Y \perp \mathbb{C}. \tag{19}$$

Since $n_A, n_B \in \mathbb{C}^\perp$, $n_A \perp e_X$ and $n_B \perp e_Y$, we have:

$$\langle e_X, e_Y \rangle = \pm \langle n_A, n_B \rangle, \tag{20}$$

and we choose a pair of $e_X$ and $e_Y$ such that:

$$\langle e_X, e_Y \rangle = \langle n_A, n_B \rangle = \cos(\phi) \in (0, 1). \tag{21}$$

Therefore, $\mathbb{S}_X$ and $\mathbb{S}_Y$ can be represented by two orthonormal bases:

$$\mathbb{S}_X \in \mathbb{A} = \operatorname{span}\{e_1\} \oplus \operatorname{span}\{e_2\} \oplus \operatorname{span}\{e_X\},$$
$$\mathbb{S}_Y \in \mathbb{B} = \operatorname{span}\{e_1\} \oplus \operatorname{span}\{e_2\} \oplus \operatorname{span}\{e_Y\}. \tag{22}$$

In Theorem 3, we show that when $\mathcal{L}_{\mathrm{MCL}}^c$ is minimized, $c_x, c_y \perp \mathbb{C}$ (Condition (A6)). Accordingly, we can set $c_x = e_X$ and $c_y = e_Y$. These settings are illustrated in Fig. 6a and Fig. 6b.

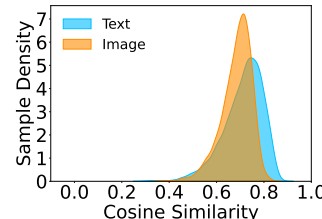

Figure 7: Density plot of $\theta_i^c$ of CLIP ViT-B/32 embeddings of MS-COCO validation set.

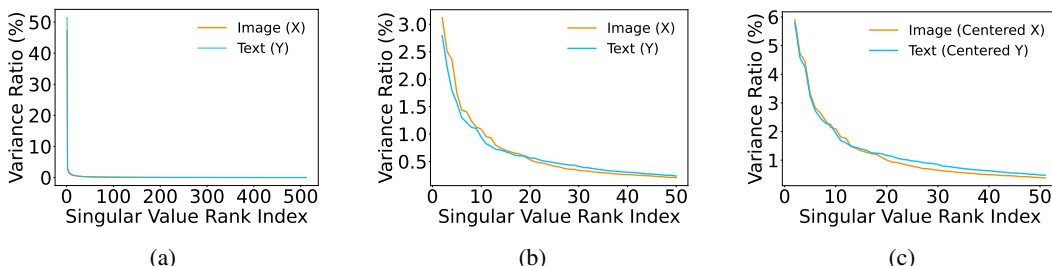

| (a) | (b) | (c) |

Figure 8: Singular values. CLIP ViT-B/32 embeddings of MS-COCO validation set are used. **(a)**: All singular values $\sigma_i$ of $X$ and $Y$. **(b)**: The $2^{\text{nd}}$ to the $50^{\text{th}}$ $\sigma_i$ of $X$ and $Y$. **(c)**: The $1^{\text{st}}$ to the $50^{\text{th}}$ $\sigma_i$ of the centered $X$ and the centered $Y$.

In Theorem 4, we show that when $\mathcal{L}_{\text{MCL}}^{i \neq c}$ is minimized, $P_C x_i = P_C y_i$ (Condition (A8)). This condition is illustrated in Fig. 6c and Fig. 6d.

Re-normalize the projections to obtain the transformed pairs:

$$
\begin{aligned}
x_i^* &= \frac{P_C x_i}{\|P_C x_i\|}, \\
y_i^* &= \frac{P_C y_i}{\|P_C y_i\|},
\end{aligned}
\tag{23}
$$

We illustrate $x_i^*$ and $y_i^*$ in Fig. 6e and Fig. 6f. In Corollary 4, we show that $(x_i^*, y_i^*)$ are perfectly aligned, as illustrated in Fig. 6g.

### B.3 JUSTIFICATION OF ASSUMPTION IN THEOREM 4

In Theorem 4, we assume that the angle between a modality input and its center, $\theta_i^c$, satisfies $\theta_i^c \in \left(0, \frac{\pi}{2}\right)$. In Lemma 15, we provide a theoretical justification for this assumption. Furthermore, the density plot of $\theta_i^c$ in Fig. 7 shows that almost all $\theta_i^c$ indeed lie within $\left(0, \frac{\pi}{2}\right)$.

## C APPENDIX C: DETAILS OF METHOD

In this subsection, we describe in details about how to detect dimension collapse, how to detect the shared space of two subspaces, and how to conduct projection onto the shared space.

### C.1 DETECT DIMENSION COLLAPSE

Suppose we have two point clouds, $X$ and $Y$, each consisting of $h$-dimensional normalized vectors: $X = (x_1, \ldots, x_N) \in (\mathbb{S}^{h-1})^N$ and $Y = (y_1, \ldots, y_N) \in (\mathbb{S}^{h-1})^N$. Then we have:

$$
\begin{aligned}
\mathbb{A} &= \operatorname{span}(X), \quad d_X = \dim(\mathbb{A}), \\
\mathbb{B} &= \operatorname{span}(Y), \quad d_Y = \dim(\mathbb{B}), \\
\mathbb{C} &= \mathbb{A} \cap \mathbb{B}, \quad d_{\text{overlap}} = \dim(\mathbb{C}).
\end{aligned}
\tag{24}
$$

Apply the Singular Value Decomposition (SVD) to $X$ and $Y$ and we get:

$$
\begin{aligned}
X &= U_X \Sigma_X V_X^\top, \\
Y &= U_Y \Sigma_Y V_Y^\top.
\end{aligned}
\tag{25}
$$

If $X$ and $Y$ collapse into subspaces of $\mathbb{S}^{h-1}$, then $\Sigma_X$ and $\Sigma_Y$ have $d_X < h$ and $d_Y < h$ significant singular values, respectively.

In the discussion in Sec. 3.4, $X$ and $Y$ represent the image and text embeddings of the MS-COCO dataset. Since $X$ and $Y$ are not centered at zero, the first singular values, $\sigma_1^x$ and $\sigma_1^y$, dominate when SVD is applied. Correspondingly, the first right singular vectors of $X$ and $Y$ are $c_x$ and $c_y$, respectively. As shown in Fig. 8a, these first right singular vectors account for approximately 50% of the explained variance. Therefore, in Fig. 2c, we plot the singular values of the centered $X$ and $Y$, which better capture the patterns of variation. In Fig. 8b, we present the $2^{\text{nd}}$ to the $50^{\text{th}}$ singular values of $X$ and $Y$, while in Fig. 8c, we show the $1^{\text{st}}$ to the $50^{\text{th}}$ singular values of the centered $X$ and the centered $Y$.

And dimension collapse in $X$ and $Y$ occurs when zero values appear on the diagonals of $\Sigma_X$ and $\Sigma_Y$.

## C.2 FIND THE SHARED SPACE

We then select the first $d_X$ columns from $V_X$ and the first $d_Y$ columns from $V_Y$ whose cumulative explained variance exceeds a predefined threshold $c$ (e.g., $c = 99\%$). We obtain:

$$
\begin{aligned}
B_X &= V_X[:, : d_X] \in \mathbb{R}^{h \times d_X} : \text{orthonormal basis for } \mathbb{A}, \\
B_Y &= V_Y[:, : d_Y] \in \mathbb{R}^{h \times d_Y} : \text{orthonormal basis for } \mathbb{B}.
\end{aligned}
\tag{26}
$$

To investigate whether $\mathbb{A}$ and $\mathbb{B}$ have overlap dimensions, we need to check the principal angles between $\mathbb{A}$ and $\mathbb{B}$, which are defined as:

**Definition 9.** *The principal angles $\gamma_1 \leq \gamma_2 \leq \cdots \leq \gamma_k$ between $\mathbb{A}$ and $\mathbb{B}$ are recursively defined as:*

$$
\cos(\gamma_i) = \max_{u \in \mathbb{A}, v \in \mathbb{B}} u^\top v, \quad \|u\| = \|v\| = 1, \quad u^\top u_j = v^\top v_j = 0 \; (j < i),
\tag{27}
$$

*where $k = \min(d_X, d_Y)$.*

The principal angles quantify the alignment between these subspaces:

- The smallest principal angle $\theta_1$ measures how close the two subspaces are: if $\gamma_1 = 0$, there is at least one common direction.

- If multiple principal angles are zero, then the intersection of the subspaces has a larger dimension.

The principal angles between subspaces $\mathbb{A}$ and $\mathbb{B}$ can be computed as follows:

1. Compute the singular values of the matrix $G = B_X^\top B_Y \in \mathbb{R}^{d_X \times d_Y}$.
2. The singular values $\sigma_i^p \in [0, 1]$
3. Then the principal angles are $\gamma_i = \arccos(\sigma_i^p)$

The number of principal angles equal to zero gives the dimension of the intersection:

$$
d_{\text{overlap}} = \#\{i : \gamma_i = 0\}.
\tag{28}
$$

In practice, due to noise or finite precision, we use a threshold: count how many $\sigma_i^p > 1 - \epsilon$ (e.g., $\epsilon = 10^{-3}$). Thus:

$$d_{\text{overlap}} = \# \left\{ i : \sigma_i^p > 1 - \epsilon \right\}. \tag{29}$$

The empirical result of MS-COCO dataset is provided in Fig. 2d.

### C.3 PROCEDURES OF SSP METHOD

In this subsection, we provide the details of the Shared Space Projection (SSP) algorithm.

**Step 1**: Apply the SVD decomposition to $X$ and $Y$ to get $V_X$ and $V_Y$ as Eq. (25).

**Step 2**: Select the first $d_X$ and $d_Y$ right singular vectors of $X$ and $Y$ whose cumulative explained variance are great than $99\%$. The resulting vectors, $B_X$ and $B_Y$, form the bases for $\mathbb{A}$ and $\mathbb{B}$, as indicated by Eq. (26).

**Step 3**: Apply the SVD decomposition $G = B_X^\top B_Y \in \mathbb{R}^{d_X \times d_Y}$.

$$G = U_G \Sigma_G V_G^\top \tag{30}$$

**Step 4**: Compute $d_{\text{overlap}}$ according to Eq. (29) while setting $\epsilon = 10^{-3}$. Compute the basis of the shared space $B_S$ by:

$$B_S = B_X U_G[:, : d_{\text{overlap}}] = B_Y V_G[:, : d_{\text{overlap}}]. \tag{31}$$

**Explain**: Since the shared space is estimated from the available data rather than the original training data (assumed inaccessible), the estimation may be noisy. To mitigate this, we can select $k < d_{\text{overlap}}$ columns from $B_S$ to form $B_S^k$. The columns of $B_S^k$ constitute an orthonormal basis for a $k$-dimensional subspace of the estimated shared space. By removing dimensions that carry minimal information, the estimation error can be reduced. The following optional step explains how to select these $k$ dimensions.

**Step 5 (Optional)**: Project $X$ and $Y$ onto each column of $B_S$:

$$
\begin{aligned}
P &= B_S^T X^T, \\
X' &= \text{einsum}(\text{'hk, kn->knh'})(B_S, P), \\
X'' &= \text{Normalize}(X') \text{ by the last dimension.}
\end{aligned}
\tag{32}
$$

Here, $\text{einsum}$ denotes Einstein summation notation. Compute the variance of $X''$ along the last two dimensions to obtain an array $S$ of length $d_{\text{overlap}}$. Each entry of $S$ is actually the singular value of projections onto the corresponding column of $B_S$. $S$ quantifies the amount of information contained in each column of $B_S$. By ranking $S$ in descending order, select the top $k$ columns from $B_S$ to form $B_S^K$.

**Step 6**: Project $X$ and $Y$ onto the column space of $B_S^k$ and get $X^*$ and $Y^*$.

$$
\begin{aligned}
X^* &= \left( B_S^k B_S^{k\,T} X^T \right)^T, \\
Y^* &= \left( B_S^k B_S^{k\,T} Y^T \right)^T.
\end{aligned}
\tag{33}
$$

**Step 7**: Normalize $X^*$ and $Y^*$ to get $X^{**}$ and $Y^{**}$. Use $X^{**}$ and $Y^{**}$ for downstream tasks.

Notably, Fig. 8b indicates that fewer than 10 dimensions account for more than $1\%$ of the explained variance, suggesting that the essential information of $X$ and $Y$ can be effectively captured using only 10 dimensions. Consequently, in Fig. 4b, we project $X$ and $Y$ onto a 10-dimensional subspace that preserves the most information.

Table 2: Size of $\theta_\Delta$ and accuracies (%) of zero-shot image classification of ViT-L/14 on various dataset.

| Model | CIFAR-10 | | | CIFAR-100 | | | ImageNet-1K | | |
|---|---|---|---|---|---|---|---|---|---|
| | $\Delta_\theta$ | R1 | R5 | $\Delta_\theta$ | R1 | R5 | $\Delta_\theta$ | R1 | R5 |
| CLIP | 77.63° | 95.12 | 99.46 | 74.19° | 65.23 | 88.88 | 77.29° | 75.56 | 94.58 |
| CLIP + Translation | 14.73° | 92.39 | 98.97 | 30.50° | 54.46 | 77.25 | 62.61° | 74.05 | 94.10 |
| CLIP + Removal | 79.36° | 12.23 | 62.33 | 73.16° | 16.82 | 6.44 | 76.84° | 67.04 | 89.76 |
| CLIP + SSP | **13.27°** | **95.12** | **99.46** | 30.39° | **64.51** | 88.79 | 62.40° | 75.26 | 94.51 |

Table 3: Size of $\theta_\Delta$ and accuracies (%) of zero-shot cross-modal retrieval of ViT-L/14 on MSCOCO.

| Model | MSCOCO | | | | | | |
|---|---|---|---|---|---|---|---|
| | $\Delta_\theta$ | $\mathbf{I \rightarrow T}$ | | | $\mathbf{T \rightarrow I}$ | | |
| | | R@1 | R@5 | R@10 | R@1 | R@5 | R@10 |
| CLIP | 78.16° | 56.06 | 79.56 | 86.84 | 35.33 | 59.96 | 70.21 |
| CLIP + Translation | 68.49° | 54.14 | 78.32 | 86.30 | 35.13 | 59.79 | 69.85 |
| CLIP + Removal | 76.03° | 49.56 | 73.42 | 82.18 | 31.23 | 54.29 | 65.00 |
| CLIP + SSP | **68.06°** | **55.54** | **78.94** | **86.64** | **35.22** | **59.86** | **70.22** |

# D APPENDIX D: DETAILS OF EXPERIMENTS

In this section, we describe in details about the set up of our experiments.

## D.1 ZERO-SHOT IMAGE CLASSIFICATION.

**Datasets.** We first evaluate our method on the zero-shot image classification task using three widely adopted datasets: two small-scale image dataset **CIFAR-10/100** Krizhevsky et al. (2009) and one large scale image dataset**ImageNet-1k** Deng et al. (2009). For CIFAR-10/100, we adopt the small set of prompts provided by OpenAI for CLIP Radford et al. (2021) (https://github.com/openai/CLIP.com). For ImageNet-1k, we adopt the large set of prompts provided by OpenAI for CLIP Radford et al. (2021) (https://colab.research.google.com/github/openai/CLIP/blob/main/notebooks/Prompt_Engineering_for_ImageNet.ipynb).

**Implementation Setup.** Our implementation refers to Eslami & de Melo (2025). For model backbone, we adopt CLIP's ViT-B/32 ViT-L/14 models. For the implementation of baseline models, we remove the same number of dimensions in the removal method Schrodi et al. (2025) with that of our SSP method. For translation Liang et al. (2022), the hyperparameter $\lambda$ controls the scale of translation. We choose the smallest value of $\lambda$, rounded to two decimal places, that yields an angle reduction larger than SSP.

**Additional Results.** We report the results using the CLIP ViT-L/14 model as the backbone in Tab. 2. Similar patterns to those in Tab. 1 can be observed, indicating that our conclusions hold across different model backbones.

As shown in both Tab. 1 and Tab. 2, reducing the modality gap becomes more challenging as the number of classes in the test set increases. This is because a larger number of classes introduces a more complex data distribution, thereby enlarging the discrepancy between the test and training distributions. Consequently, our shared space estimation incurs greater estimation error, which limits the capacity of our method to further reduce the modality gap.

## D.2 ZERO-SHOT CROSS-MODAL RETRIEVAL.

**Datasets.** In addition to zero-shot image classification, we evaluate our method on zero-shot image-to-text and text-to-image retrieval using the MSCOCO (Lin et al., 2014) and Flickr30K (Plummer

Table 4: Size of $\theta_\Delta$ and accuracies (%) of zero-shot cross-modal retrieval of ViT-L/14 on Flickr30K.

| Model | | Flickr30K | | | | | |
| | $\Delta_\theta$ | $\mathbf{I \rightarrow T}$ | | | $\mathbf{T \rightarrow I}$ | | |
| | | R@1 | R@5 | R@10 | R@1 | R@5 | R@10 |
| CLIP | 78.16° | 56.06 | 79.56 | 86.84 | 35.33 | 59.96 | 70.21 |
| CLIP + Translation | - | - | - | - | - | - | - |
| CLIP + Removal | - | - | - | - | - | - | - |
| CLIP + SSP | - | - | - | - | - | - | - |

et al., 2015) datasets. Unlike the common practice of appending a prompt such as 'a photo of the caption', we directly use the raw captions to generate text embeddings. This approach aims to align the text space more closely with its natural form rather than introducing distortion through artificial prompts.

**Implementation Setup.** This implementation setup follows Sec. D.1. The only difference is that we only use CLIP ViT-L/14 as the model backbone.

### D.3 RESULTS

The goal of this experiment is to reduce the size of the modality gap as much as possible without harming downstream performance. In Tab. 3 and Tab. 4, we list results of the size of the modality gap ($\Delta_\theta$), the top-1 accuracy (R@1), the top-5 accuracy (R@5), and the top-10 accuracy (R@10). Similar patterns to those in Tab. 1 can be observed, indicating that our conclusions hold across different downstream tasks.

# E APPENDIX E: PROOFS

## E.1 DETAILS OF THEOREM 1

In this section, we provide proofs of Theorem 1 that is proposed in Sec. 3.2. We also provide details of the auxiliary theorems (Theorem S1 and Theorem S2) and technical lemmas (Lemma 1, Lemma 2, Lemma 3, Lemma 4) that support the proof of Theorem 1. For convenience in reading, let us recall some related notions and definitions.

- $h, N \in \mathbb{N}$.
- $\mathbb{S}^{h-1} = \left\{ z \in \mathbb{R}^h : \|z\| = 1 \right\}$.
- $\sigma_{h-1}$: the uniform probability measure of $\mathbb{S}^{h-1}$.

**Definition** (Multimodal Contrastive Loss (MCL Loss)). Let $(X, Y)$ be an $N$-pair configuration, where $X = (x_1, \ldots, x_N) \in (\mathbb{S}^{h-1})^N$ and $Y = (y_1, \ldots, y_N) \in (\mathbb{S}^{h-1})^N$. $\forall \tau > 0$, the multimodal contrastive loss $\mathcal{L}_{\mathrm{MCL}}(\cdot, \cdot) : (\mathbb{S}^{h-1})^N \times (\mathbb{S}^{h-1})^N \to \mathbb{R}$ is defined as:

$$\mathcal{L}_{\mathrm{MCL}} = \frac{1}{N} \sum_{i=1}^{N} \mathcal{L}_{\mathrm{MCL}}^i, \quad \text{where } \mathcal{L}_{\mathrm{MCL}}^i = \mathcal{L}_{\mathcal{X} \to \mathcal{Y}}(x_i; Y) + \mathcal{L}_{\mathcal{Y} \to \mathcal{X}}(y_i; X).$$

Here, $\mathcal{L}_{\mathcal{X} \to \mathcal{Y}}$ is the $\mathcal{X}$-to-$\mathcal{Y}$ alignment and $\mathcal{L}_{\mathcal{Y} \to \mathcal{X}}$ is the $\mathcal{Y}$-to-$\mathcal{X}$ alignment, which are defined respectively as:

$$\mathcal{L}_{\mathcal{X} \to \mathcal{Y}}(x_i; Y) = -\log \frac{\exp\left(x_i \cdot y_i / \tau\right)}{\sum_{j=1}^{N} \exp\left(x_i \cdot y_j / \tau\right)}, \quad \mathcal{L}_{\mathcal{Y} \to \mathcal{X}}(y_i; X) = -\log \frac{\exp\left(x_i \cdot y_i / \tau\right)}{\sum_{j=1}^{N} \exp\left(x_j \cdot y_i / \tau\right)}.$$

### E.1.1 PROOF OF THEOREM 1

In this subsection, we provide the proof of Theorem 1. For convenience in reading, we first restate Theorem 1 here.

**Theorem 1.** [Restate] Let $(X, Y)$ be an $N$-pair configuration, where $X = (x_1, \ldots, x_N) \in (\mathbb{S}^{h-1})^N$ are $iid$ samples from $\mu_x$ and $Y = (y_1, \ldots, y_N) \in (\mathbb{S}^{h-1})^N$ are $iid$ samples from $\mu_y$. Let $\nu = h/2 - 1$, it holds that:

$$\lim_{N \to \infty} \mathcal{L}_{\mathrm{MCL}} - 2\log(N) = \mathbb{E}_{x_i \sim \mu_x}\left[-\frac{x_i \cdot y_i}{\tau}\right] + \mathbb{E}_{x_i \sim \mu_x}\left[\log \mathbb{E}_{y_i \sim \mu_y}\left[\exp\left(\frac{x_i \cdot y_i}{\tau}\right)\right]\right]$$

$$+ \mathbb{E}_{y_i \sim \mu_y}\left[-\frac{x_i \cdot y_i}{\tau}\right] + \mathbb{E}_{y_i \sim \mu_y}\left[\log \mathbb{E}_{x_j \sim \mu_x}\left[\exp\left(\frac{x_i \cdot y_i}{\tau}\right)\right]\right]$$

$$\geq -\frac{2}{\tau} + 2\log\left(\Gamma(\nu + 1)(2\tau)^\nu I_\nu\left(\frac{1}{\tau}\right)\right)$$

where equality is attained if and only if there exists a configuration of $(X, Y)$ such that:

(A1) $\forall i \in [N]$, $x_i = y_i$.

(A2) $\mu_x = \sigma_{h-1}$ and $\mu_y = \sigma_{h-1}$.

*Proof.* We first decompose $\lim_{N \to \infty} \mathcal{L}_{\mathrm{MCL}}^c - 2\log(N)$ into two parts:

$$\lim_{N \to \infty} (\mathcal{L}_{\mathrm{MCL}} - 2\log(N)) = \lim_{N \to \infty} \left(\frac{1}{N} \sum_{i=1}^{N} \mathcal{L}_{\mathcal{X} \to \mathcal{Y}}(x_i; Y) - \log(N)\right)$$

$$+ \lim_{N \to \infty} \left(\frac{1}{N} \sum_{i=1}^{N} \mathcal{L}_{\mathcal{Y} \to \mathcal{X}}(y_i; X) - \log(N)\right). \tag{34}$$

According to Theorem S2, the convergent function and its lower bound of $\mathcal{L}_{\mathcal{X}\to\mathcal{Y}}$ are:

$$
\lim_{N\to\infty} \frac{1}{N} \sum_{i=1}^{N} \mathcal{L}_{\mathcal{X}\to\mathcal{Y}}(x_i; Y) - \log(N)
$$

$$
= \mathbb{E}_{x_i\sim\mu_x}\left[-\frac{x_i\cdot y_i}{\tau}\right] + \mathbb{E}_{x_i\sim\mu_x}\left[\log\mathbb{E}_{y_i\sim\mu_y}\left[\exp\left(\frac{x_i\cdot y_j}{\tau}\right)\right]\right] \tag{35}
$$

$$
\geq -\frac{1}{\tau} + \log\left[\Gamma\left(\frac{h}{2}\right)(2\tau)^{\frac{h}{2}-1} I_{\frac{h}{2}-1}\left(\frac{1}{\tau}\right)\right]
$$

where equality is attained if and only if there exists a configuration of $(X, Y)$ such that:

(i) $\forall i \in [N]$, $x_i = y_i$.

(ii) $\mu_x = \sigma_{h-1}$ and $\mu_y = \sigma_{h-1}$.

This Theorem also holds for $\mathcal{L}_{\mathcal{Y}\to\mathcal{X}}$:

$$
\lim_{N\to\infty} \frac{1}{N} \sum_{i=1}^{N} \mathcal{L}_{\mathcal{Y}\to\mathcal{X}}(y_i; X) - \log(N)
$$

$$
= \mathbb{E}_{y_i\sim\mu_x}\left[-\frac{x_i\cdot y_i}{\tau}\right] + \mathbb{E}_{y_i\sim\mu_y}\left[\log\mathbb{E}_{x_i\sim\mu_x}\left[\exp\left(\frac{x_i\cdot y_j}{\tau}\right)\right]\right] \tag{36}
$$

$$
\geq -\frac{1}{\tau} + \log\left[\Gamma\left(\frac{h}{2}\right)(2\tau)^{\frac{h}{2}-1} I_{\frac{h}{2}-1}\left(\frac{1}{\tau}\right)\right]
$$

where equality is attained if and only if there exists a configuration of $(X, Y)$ such that:

(iii) $\forall i \in [N]$, $x_i = y_i$.

(iv) $\mu_x = \sigma_{h-1}$ and $\mu_y = \sigma_{h-1}$.

Combining Eq. (34), Eq. (35) and Eq. (36), we conclude that:

$$
\lim_{N\to\infty} \mathcal{L}_{\mathrm{MCL}} - 2\log(N) = \mathbb{E}_{x_i\sim\mu_x}\left[-\frac{x_i\cdot y_i}{\tau}\right] + \mathbb{E}_{x_i\sim\mu_x}\left[\log\mathbb{E}_{y_i\sim\mu_y}\left[\exp\left(\frac{x_i\cdot y_i}{\tau}\right)\right]\right]
$$

$$
+ \mathbb{E}_{y_i\sim\mu_y}\left[-\frac{x_i\cdot y_i}{\tau}\right] + \mathbb{E}_{y_i\sim\mu_y}\left[\log\mathbb{E}_{x_j\sim\mu_x}\left[\exp\left(\frac{x_i\cdot y_i}{\tau}\right)\right]\right] \tag{37}
$$

$$
\geq -\frac{2}{\tau} + 2\log\left[\Gamma\left(\frac{h}{2}\right)(2\tau)^{\frac{h}{2}-1} I_{\frac{h}{2}-1}\left(\frac{1}{\tau}\right)\right]
$$

where equality is attained if and only if the following conditions hold:

(A1) $\forall i \in [N]$, $x_i = y_i$.

(A2) $\mu_x = \sigma_{h-1}$ and $\mu_y = \sigma_{h-1}$.

$\square$

### E.1.2 AUXILIARY THEOREMS PART 1

In this subsection, we provide details and proofs of the auxiliary theorems (Theorem S1 and Theorem S2) that support the proof of Theorem 1.

**Theorem S1.** *Let $(X, Y)$ be an $N$-pair configuration, where $X = (x_1, \ldots, x_N) \in (\mathbb{S}^{h-1})^N$ are iid samples from $\mu_x$ and $Y = (y_1, \ldots, y_N) \in (\mathbb{S}^{h-1})^N$ are iid samples from $\mu_y$. It holds that:*

$$
\lim_{N \to \infty} \frac{1}{N} \sum_{i=1}^{N} \mathcal{L}_{\mathcal{X} \to \mathcal{Y}}(x_i; Y) - \log(N) = \lim_{N \to \infty} \frac{1}{N} \sum_{i=1}^{N} -\log \frac{\exp(x_i \cdot y_i / \tau)}{\sum_{j=1}^{N} \exp(x_i \cdot y_j / \tau)} - \log(N)
$$
$$
= \mathbb{E}_{x_i \cdot y_i} \left[ -\frac{x_i \cdot y_i}{\tau} \right] + \mathbb{E}_{x_i \sim \mu_x} \left[ \log \mathbb{E}_{y_i \sim \mu_y} \left[ \exp \left( \frac{x_i \cdot y_j}{\tau} \right) \right] \right]
$$
(38)

*Proof.* $\forall x_i \in X$, the $\mathcal{X}$-to-$\mathcal{Y}$ alignment of $x_i$ can be rewritten as:

$$
\mathcal{L}_{\mathcal{X} \to \mathcal{Y}}(x_i; Y) = -\log \frac{\exp(x_i \cdot y_i / \tau)}{\sum_j \exp(x_i \cdot y_j / \tau)}
$$
$$
= -\frac{x_i \cdot y_i}{\tau} + \log \left( N \frac{1}{N} \sum_{j=1}^{N} \exp \left( \frac{x_i \cdot y_j}{\tau} \right) \right)
$$
(39)
$$
= -\frac{x_i \cdot y_i}{\tau} + \log \left( \frac{1}{N} \sum_{j=1}^{N} \exp \left( \frac{x_i \cdot y_j}{\tau} \right) \right) + \log(N).
$$

Denote $h_N(x)$ and $h(x)$ as:

$$
h_N(x) = \log \left( \frac{1}{N} \sum_{j=1}^{N} \exp \left( \frac{x \cdot y_j}{\tau} \right) \right),
$$
$$
\text{and } h(x) = \log \left( \mathbb{E}_{y \sim \mu_y} \left[ \exp \left( \frac{x \cdot y}{\tau} \right) \right] \right).
$$
(40)

Lemma 2 reveals that $h_N(x)$ uniformly converges to $h(x)$ almost surely. Thus, we have:

$$
\sup_{x \in \mathbb{S}^{h-1}} |h_N(x) - h(x)| \xrightarrow[N \to \infty]{\text{a.s.}} 0.
$$
(41)

According to the Strong Law of Large Numbers (SLLN), we have:

$$
\frac{1}{N} \sum_{i=1}^{N} h(x_i) \xrightarrow[N \to \infty]{\text{a.s.}} \mathbb{E}_{x \sim \mu_x}[h(x)].
$$
(42)

Combining Eq. (41) and Eq. (42), we get:

$$
\frac{1}{N} \sum_{i=1}^{N} h_N(x_i) = \frac{1}{N} \sum_{i=1}^{N} h(x_i) + \frac{1}{N} \sum_{i=1}^{N} (h_N(x_i) - h(x_i))
$$
(43)
$$
\xrightarrow[N \to \infty]{\text{a.s.}} \mathbb{E}_{x \sim \mu_x}[h(x)].
$$

Similarly, by the Strong Law of Large Numbers (SLLN), we have:

$$
\frac{1}{N} \sum_{i=1}^{N} -\frac{x_i \cdot y_i}{\tau} \xrightarrow[N \to \infty]{\text{a.s.}} \mathbb{E}_{x_i \sim \mu_x} \left[ -\frac{x_i \cdot y_i}{\tau} \right].
$$
(44)

Putting Eq. (39), Eq. (43) and Eq. (44) together, the convergent function of $\frac{1}{N} \sum_{i=1}^{N} \mathcal{L}_{\mathcal{X} \to \mathcal{Y}}(x_i; Y)$ can be derived as:

$$\lim_{N\to\infty} \frac{1}{N}\sum_{i=1}^{N}\mathcal{L}_{\mathcal{X}\to\mathcal{Y}}(x_i;Y) - \log(N) = \lim_{N\to\infty} \frac{1}{N}\sum_{i=1}^{N}\left(-\frac{x_i\cdot y_i}{\tau} + h_N(x_i)\right)$$

$$= \mathbb{E}_{x_i\cdot y_i}\left[-\frac{x_i\cdot y_i}{\tau}\right] + \mathbb{E}_{x_i\sim\mu_x}\left[h(x_i)\right] \tag{45}$$

$$= \mathbb{E}_{x_i\cdot y_i}\left[-\frac{x_i\cdot y_i}{\tau}\right] + \mathbb{E}_{x_i\sim\mu_x}\left[\log\mathbb{E}_{y_j\sim\mu_y}\left[\exp\left(\frac{x_i\cdot y_j}{\tau}\right)\right]\right].$$

$$\square$$

**Theorem S2.** *Let $(X,Y)$ be an $N$-pair configuration, where $X = (x_1,\ldots,x_N) \in (\mathbb{S}^{h-1})^N$ are iid samples from $\mu_x$ and $Y = (y_1,\ldots,y_N) \in (\mathbb{S}^{h-1})^N$ are iid samples from $\mu_y$. Let $\nu = h/2 - 1$, it holds that:*

$$\lim_{N\to\infty} \frac{1}{N}\sum_{i=1}^{N}\mathcal{L}_{\mathcal{X}\to\mathcal{Y}}(x_i;Y) - \log(N)$$

$$= \mathbb{E}_{x_i\sim\mu_x}\left[-\frac{x_i\cdot y_i}{\tau}\right] + \mathbb{E}_{x_i\sim\mu_x}\left[\log\mathbb{E}_{y_i\sim\mu_y}\left[\exp\left(\frac{x_i\cdot y_j}{\tau}\right)\right]\right] \tag{46}$$

$$\geq \log\left(\Gamma\left(\nu+1\right)\left(2\tau\right)^{\nu}I_{\nu}\left(\frac{1}{\tau}\right)\right)$$

*where equality is attained if and only if the following conditions hold:*

*(B1)* $\forall i \in [N]$, $x_i = y_i$.

*(B2)* $\mu_x = \sigma_{h-1}$ and $\mu_y = \sigma_{h-1}$.

*Proof.* **Step 1**: We start the proof by find the convergent function of $\frac{1}{N}\sum_{i=1}^{N}\mathcal{L}_{\mathcal{X}\to\mathcal{Y}}(x_i;Y)$ as $N \to \infty$. $\forall x_i \in X$, as prove in Theorem S1:

$$\lim_{N\to\infty} \frac{1}{N}\sum_{i=1}^{N}\mathcal{L}_{\mathcal{X}\to\mathcal{Y}}(x_i;Y) - \log(N) = \lim_{N\to\infty} \frac{1}{N}\sum_{i=1}^{N} -\log\frac{\exp\left(x_i\cdot y_i/\tau\right)}{\sum_{j=1}^{N}\exp\left(x_i\cdot y_j/\tau\right)} - \log(N)$$

$$= \mathbb{E}_{x_i\cdot y_i}\left[-\frac{x_i\cdot y_i}{\tau}\right] + \mathbb{E}_{x_i\sim\mu_x}\left[\log\mathbb{E}_{y_i\sim\mu_y}\left[\exp\left(\frac{x_i\cdot y_j}{\tau}\right)\right]\right]. \tag{47}$$

**Step 2**: Next, we find the minimal value and the optimal condition of convergent function.

According to the Cauchy-Schwarz inequality, the first term in Eq. (47) can be bounded below:

$$\mathbb{E}_{x_i\cdot y_i}\left[-\frac{x_i\cdot y_i}{\tau}\right] \geq \mathbb{E}_{x_i\cdot y_i}\left[-\frac{\|x_i\|\,\|y_i\|}{\tau}\right] = -\frac{1}{\tau}. \tag{48}$$

where equality is attained if and only if there exists a configuration of $(X,Y)$ such that :

(B1) $\forall i \in [N]$, $x_i = y_i$.

Note that condition (B1) implies $\mu_x = \mu_y$. Applying this condition to the second term in Eq. (47), we can transform it as:

$$\mathbb{E}_{x\sim\mu_x}\left[\log\left(\mathbb{E}_{y\sim\mu_y}\left[\exp\left(\frac{x\cdot y}{\tau}\right)\right]\right)\right] = \mathbb{E}_{x\sim\mu}\left[\log\left(\mathbb{E}_{y\sim\mu}\left[\exp\left(\frac{x\cdot y}{\tau}\right)\right]\right)\right]. \tag{49}$$

Let $\mathbb{M}(\mathbb{S}^{h-1})$ be the set of Borel probability measures in $\mathbb{S}^{h-1}$. The RHS of Eq. (49) is then a functional $\mathcal{F}[\cdot] : \mathbb{M}(\mathbb{S}^{h-1}) \to \mathbb{R}$:

$$\mathcal{F}[\mu] = \mathbb{E}_{x \sim \mu} \left[ \log \left( \mathbb{E}_{y \sim \mu} \left[ \exp \left( \frac{x \cdot y}{\tau} \right) \right] \right) \right]. \tag{50}$$

According to Lemma 3, $\mathcal{F}[\mu]$ is minimized when $\mu = \sigma_{h-1}$ where $\sigma_{h-1}$ is the uniform measure of $\mathbb{S}^{h-1}$:

$$\sigma_{h-1} = \underset{\mu \in \mathbb{M}(\mathbb{S}^{h-1})}{\arg \min} \mathcal{F}[\mu]. \tag{51}$$

Therefore, we have:

$$\mathcal{F}[\mu] \geq \mathcal{F}[\sigma_{h-1}]. \tag{52}$$

where equality is attained if and only if there exists a configuration of $(X, Y)$ such that :

(B2) $\mu_x = \mu_y = \sigma_{h-1}$

Let $\Gamma \left( \cdot \right)$ be the Gamma function, Lemma 4 derives that:

$$\begin{aligned}
\mathcal{F}[\sigma_{h-1}] &= \mathbb{E}_{x \sim \sigma_{h-1}} \left[ \mathbb{E}_{y \sim \sigma_{h-1}} \left[ \exp \left( \frac{x \cdot y}{\tau} \right) \right] \right] \\
&= \log \left[ \Gamma \left( \frac{h}{2} \right) (2\tau)^{\frac{h}{2}-1} I_{\frac{h}{2}-1} \left( \frac{1}{\tau} \right) \right]
\end{aligned} \tag{53}$$

Combining Eq. (47), Eq. (48), Eq. (49), Eq. (53), we conclude that:

$$\begin{aligned}
&\lim_{N \to \infty} \frac{1}{N} \sum_{i=1}^{N} \mathcal{L}_{\mathcal{X} \to \mathcal{Y}}(x_i; Y) - \log(N) \\
&= \mathbb{E}_{x_i \sim \mu_x} \left[ -\frac{x_i \cdot y_i}{\tau} \right] + \mathbb{E}_{x_i \sim \mu_x} \left[ \log \mathbb{E}_{y_i \sim \mu_y} \left[ \exp \left( \frac{x_i \cdot y_j}{\tau} \right) \right] \right] \\
&\geq -\frac{1}{\tau} + \log \left[ \Gamma \left( \frac{h}{2} \right) (2\tau)^{\frac{h}{2}-1} I_{\frac{h}{2}-1} \left( \frac{1}{\tau} \right) \right]
\end{aligned} \tag{54}$$

where equality is attained if and only if the following conditions hold:

(B1) $\forall i \in [N], x_i = y_i$.

(B2) $\mu_x = \sigma_{d-1}$ and $\mu_y = \sigma_{d-1}$.

$\square$

### E.1.3 TECHNICAL LEMMAS PART 1

In this section, we provide details and proofs of the technical lemmas (technical lemmas (Lemma 1, Lemma 2, Lemma 3, Lemma 4) that support the proof of Theorem 1, Theorem S1 and Theorem S2.

**Lemma 1.** *Let $x \in \mathbb{S}^{h-1}$ and $Y$ be an $N$-point configuration, where $Y = (y_1, \ldots, y_N) \in (\mathbb{S}^{h-1})^N$ are iid samples from $\mu_y$. $\forall \tau > 0$, define a sequence of functions $\{g_N\} : \mathbb{S}^{h-1} \to \mathbb{R}$ as:*

$$g_N(x) = \frac{1}{N} \sum_{j=1}^{N} \exp\left(\frac{x \cdot y_j}{\tau}\right). \tag{55}$$

*Define a function $g : \mathbb{S}^{h-1} \to \mathbb{R}$ as:*

$$g(x) = \mathbb{E}_{y \sim \mu_y}\left[\exp\left(\frac{x \cdot y}{\tau}\right)\right]. \tag{56}$$

*It holds that $\{g_N\}$ converges uniformly to $g$:*

$$g_N(x) \xrightarrow[N \to \infty]{\text{unif.}} g(x). \tag{57}$$

*Proof.* **Step 1 Boundedness and Lipschitz Property:**

Consider a function class $\mathcal{F} = \left\{ f_x(y) = \exp\left(\frac{x \cdot y}{\tau}\right) : x, y \in \mathbb{S}^{h-1} \right\}$. Since $\|x\| = \|y\| = 1, x \cdot y \in [-1, 1]$, hence $\forall f_x \in F$, we have:

$$|f_x(y)| \leq e^{1/\tau}. \tag{58}$$

Therefore, $f_x(y)$ is uniformly bounded in $y$, so is its derivative:

$$\|\nabla_x f_x(y)\| = \left\|\frac{y}{\tau} f_x(y)\right\| \leq \frac{1}{\tau} e^{1/\tau}. \tag{59}$$

Then $\forall x_k \in \mathbb{S}^{h-1}$,

$$|f_x(y) - f_{x_k}(y)| \leq \frac{1}{\tau} e^{1/\tau} =: L. \tag{60}$$

Thus, $f_x(y)$ is Lipschitz in $x$ with constant $L = \frac{e^{1/\tau}}{\tau}$, uniformly in $y$.

**Step 2 $\eta$-Net:**

According to Lemma 5.2 in (Vershynin, 2010), $\forall \varepsilon > 0$ and $\eta = \frac{\varepsilon}{4L}$, there exists a finite $\eta$-net, $\mathcal{N}_\eta = \{x_1, x_2, \ldots, x_K\} \subset \mathbb{S}^{h-1}$, with cardinality:

$$K = |\mathcal{N}_\eta| \leq \left(1 + \frac{2}{\eta}\right)^h < \left(\frac{3}{\eta}\right)^h. \tag{61}$$

$\forall x \in \mathbb{S}^{h-1}, \exists x_k \in \mathcal{N}_\eta$ such that $\|x - x_k\| < \eta$. Because $f_x(y)$ is $L$-Lipschitz in $x$, we have:

$$|f_x(y) - f_{x_k}(y)| \leq L \|x - x_k\| = L\eta. \tag{62}$$

And we also have:

$$\begin{aligned} |g_N(x) - g_N(x_k)| &\leq L\eta, \\ |g(x) - g(x_k)| &\leq L\eta. \end{aligned} \tag{63}$$

**Step 3 Probability Bound:**

$\forall x_k \in \mathcal{N}_\eta$, the random variables $Z_j := f_{x_k}(y_j)$ are iid and lie in $\left[e^{-1/\tau}, e^{1/\tau}\right]$. According to the Hoeffding's inequality:

$$P\left(|g_N(x_k) - g(x_k)| > \frac{\varepsilon}{2}\right) \leq 2\exp\left(-\frac{2N(\varepsilon/2)^2}{(2e^{1/\tau})^2}\right) = 2e^{-cN\varepsilon^2}, \tag{64}$$

where $c = \frac{1}{8e^{2/\tau}} > 0$. Taking a union bound over the $\eta$-net:

$$P\left(\max_{x_k \in \mathcal{N}_\eta} |g_N(x_k) - g(x_k)| > \frac{\varepsilon}{2}\right) \leq 2Ke^{-cN\varepsilon^2}. \tag{65}$$

**Step 4 Uniform Convergence:**

Since $\forall x \in \mathbb{S}^{h-1}$, $|g_N(x) - g(x)|$ can be decomposed as:

$$
\begin{aligned}
|g_N(x) - g(x)| &\leq |g_N(x) - g_N(x_k)| + |g_N(x_k) - g(x_k)| + |g(x_k) - g(x)| \\
&\leq 2L\eta + \max_{x_k \in \mathcal{N}_\eta} |g_N(x_k) - g(x_k)| \\
&= \frac{\varepsilon}{2} + \max_{x_k \in \mathcal{N}_\eta} |g_N(x_k) - g(x_k)|.
\end{aligned}
\tag{66}
$$

Plugging Eq. (65) into Eq. (66), we have:

$$
\begin{aligned}
P\left(\sup_{x \in \mathbb{S}^{h-1}} |g_N(x) - g(x)| > \varepsilon\right) &\leq P\left(\max_{x_k \in \mathcal{N}_\eta} |g_N(x_k) - g(x_k)| > \frac{\varepsilon}{2}\right) \\
&\leq 2Ke^{-cN\varepsilon^2},
\end{aligned}
\tag{67}
$$

and therefore:

$$\sup_{x \in \mathbb{S}^{h-1}} |g_N(x) - g(x)| \xrightarrow[N\to\infty]{P} 0. \tag{68}$$

Eq. (67) justifies that:

$$\sum_{N=1}^\infty P\left(\sup_{x \in \mathbb{S}^{h-1}} |g_N(x) - g(x)| > \varepsilon\right) \leq 2K\sum_{N=1}^\infty e^{-cN\varepsilon^2} < \infty. \tag{69}$$

According to the Borel–Cantelli lemma:

$$P\left(\limsup_{N\to\infty} \sup_{x \in \mathbb{S}^{h-1}} |g_N(x) - g(x)| > \varepsilon\right) = 0. \tag{70}$$

Therefore:

$$\sup_{x \in \mathbb{S}^{h-1}} |g_N(x) - g(x)| \xrightarrow[N\to\infty]{\text{a.s.}} 0. \tag{71}$$

We conclude now the empirical averages $g_N(\cdot)$ converge uniformly in $\mathbb{S}^{h-1}$ to $g(\cdot)$:

$$g_N(x) \xrightarrow[N\to\infty]{\text{unif.}} g(x). \tag{72}$$

$\square$

**Lemma 2.** *Let $x \in \mathbb{S}^{h-1}$ and $Y$ be an $N$-point configuration, where $Y = (y_1, \ldots, y_N) \in (\mathbb{S}^{h-1})^N$ are iid samples from $\mu_y$. $\forall \tau > 0$, define a sequence of functions $\{h_N\} : \mathbb{S}^{h-1} \to \mathbb{R}$ as:*

$$h_N(x) = \log \left( \frac{1}{N} \sum_{j=1}^{N} \exp \left( \frac{x \cdot y_j}{\tau} \right) \right). \tag{73}$$

*Define a function $h : \mathbb{S}^{h-1} \to \mathbb{R}$ as:*

$$h(x) = \log \left( \mathbb{E}_{y \sim \mu_y} \left[ \exp \left( \frac{x \cdot y}{\tau} \right) \right] \right). \tag{74}$$

*It holds that $\{h_N\}$ converges uniformly to $h$:*

$$\lim_{N \to \infty} h_N(x) \xrightarrow[N \to \infty]{\text{unif.}} h(x). \tag{75}$$

*Proof.* According to Lemma 1:

$$\sum_{j=1}^{N} \exp \left( \frac{x \cdot y_j}{\tau} \right) = g_N(x) \xrightarrow[N \to \infty]{\text{unif.}} g(x) = \mathbb{E}_{y \sim \mu_y} \left[ \exp \left( \frac{x \cdot y}{\tau} \right) \right], \tag{76}$$

and

$$\sup_{x \in \mathbb{S}^{h-1}} |g_N(x) - g(x)| \xrightarrow[N \to \infty]{\text{a.s.}} 0. \tag{77}$$

Because $\langle x, y \rangle \in [-1, 1]$ for unit vectors, $\exp(x \cdot y / \tau)$ satisfies:

$$e^{-1/\tau} \leq \exp \left( \frac{x \cdot y}{\tau} \right) \leq e^{1/\tau}. \tag{78}$$

Hence $\forall x, g_N(x), g(x) \in [a, b]$ with $a = e^{-1/\tau} > 0, b = e^{1/\tau} > 0$. In the compact interval $[a, b]$, by the mean value theorem, $\forall u < v \in [a, b], \exists u < \xi < v$ such that:

$$|\log u - \log v| = \frac{|u - v|}{\xi} \leq \frac{1}{a} |u - v| = e^{1/\tau} |u - v|. \tag{79}$$

Thus, the function $\log(\cdot)$ is Lipschitz . Therefore:

$$\sup_{x \in \mathbb{S}^{h-1}} |h_N(x) - h(x)| = \sup_{x \in \mathbb{S}^{h-1}} |\log g_N(x) - \log g(x)| \leq \frac{1}{a} \sup_{x \in \mathbb{S}^{h-1}} |g_N(x) - g(x)| \xrightarrow[N \to \infty]{\text{a.s.}} 0 \tag{80}$$

We conclude now $h_N(\cdot)$ converge uniformly in $\mathbb{S}^{h-1}$ to $h(\cdot)$:

$$\lim_{N \to \infty} h_N(x) \xrightarrow{\text{unif.}} h(x) \tag{81}$$

$\square$

**Lemma 3.** *Let $M\left(\mathbb{S}^{h-1}\right)$ be the set of Borel probability measures in $\mathbb{S}^{h-1}$. Let $\sigma_{h-1} \in M\left(\mathbb{S}^{h-1}\right)$ be the uniform probability measure in $\mathbb{S}^{h-1}$. $\forall x, y \in \mathbb{S}^{h-1}$ and $\tau > 0$, a function $f : \mathbb{S}^{h-1} \times \mathbb{S}^{h-1} \to \mathbb{R}^+$ is defined as:*

$$f(x, y) = \exp\left(\frac{x \cdot y}{\tau}\right). \tag{82}$$

*$\forall \mu \in M\left(\mathbb{S}^{h-1}\right)$, a functional $\mathcal{G} : M\left(\mathbb{S}^{h-1}\right) \to \mathbb{R}^+$ is defined as:*

$$\mathcal{F}_f[\mu] = \int_{\mathbb{S}^{h-1}} \log\left(\int_{\mathbb{S}^{h-1}} f(x, y)\mathrm{d}\mu(y)\right) \mathrm{d}\mu(x). \tag{83}$$

*It holds that $\sigma_{h-1}$ is the unique minimizer of $\mathcal{F}$:*

$$\min_{\mu \in \mathcal{M}(\mathbb{S}^{h-1})} \mathcal{F}_f[\mu] = \min_{\mu \in \mathcal{M}(\mathbb{S}^{h-1})} \int_{\mathbb{S}^{h-1}} \log\left(\int_{\mathbb{S}^{h-1}} f(x, y)\mathrm{d}\mu(y)\right) \mathrm{d}\mu(x). \tag{84}$$

*Proof.* **Step 1: A change of probability measure.**

Let $\sigma := \sigma_{h-1}$ be the uniform probability in $\mathbb{S}^{h-1}$. By rotational invariance there is a constant $c$ such that:

$$c := c_{h,\tau}(x) = \int_{y \in \mathbb{S}^{h-1}} f(x, y)d\sigma(y), \tag{85}$$

which is independent of $x$. $\forall x \in \mathbb{S}^{h-1}$. Define a kernel $K$ as:

$$K(x, dy) := \frac{f(x, y)}{c}d\sigma(y), \tag{86}$$

so that:

$$\int_{y \in \mathbb{S}^{h-1}} K(x, dy) = 1. \tag{87}$$

Since $f(x, y) = f(y, x)$, exchanging $x$ and $y$, the following holds:

$$\sigma(dx)K(x, dy) = \sigma(dy)K(y, dx). \tag{88}$$

For any measurable $A \subset \mathbb{S}^{h-1}$, we have:

$$K(x, A) := \int_{y \in A} \frac{f(x, y)}{c}d\sigma(y). \tag{89}$$

Consider a probability distribution $\mu$ in $\mathbb{S}^{h-1}$, define:

$$\begin{aligned}
(\mu K)(A) &:= \int_{x \in \mathbb{S}^{h-1}} K(x, A)d\mu(x) \\
&= \int_{x \in \mathbb{S}^{h-1}} \int_{y \in A} \frac{f(x, y)}{c}d\sigma(y)d\mu(x) \\
&= \int_{y \in A} \int_{x \in \mathbb{S}^{h-1}} \frac{f(x, y)}{c}d\mu(x)d\sigma(y).
\end{aligned} \tag{90}$$

Therefore, $\mu K \ll \sigma$, i.e., $\mu K$ is absolutely continuous with respect to $\sigma$. By Radon–Nikodym theorem, we have:

$$\frac{d(\mu K)}{d\sigma}(y) = \frac{1}{c}\int_{x\in\mathbb{S}^{h-1}} f(x,y)d\mu(x). \tag{91}$$

Note that:

$$
\begin{aligned}
(\sigma K)(A) &= \int_{x\in\mathbb{S}^{h-1}} K(x,A)d\sigma(x) \\
&= \int_{x\in\mathbb{S}^{h-1}}\int_{y\in A} \frac{f(x,y)}{c}d\sigma(y)d\sigma(x) \\
&= \int_{y\in A}\int_{x\in\mathbb{S}^{h-1}} \frac{f(x,y)}{c}d\sigma(x)d\sigma(y) \\
&= \int_{y\in A} d\sigma(y) \\
&= \sigma(A)
\end{aligned}
\tag{92}
$$

According to Eq. (88), exchanging $x$ and $y$ in Eq. (91), we get:

$$\frac{d(\mu K)}{d\sigma}(x) = \frac{1}{c}\int_{y\in\mathbb{S}^{h-1}} f(y,x)d\mu(y). \tag{93}$$

And since $f(y,x) = f(x,y)$, then:

$$\frac{d(\mu K)}{d\sigma}(x) = \frac{1}{c}\int_{y\in\mathbb{S}^{h-1}} f(x,y)d\mu(y). \tag{94}$$

**Step 2: An exact identity for $\mathcal{F}_f$**

Define a (normalized) zonal integral operator $T$ on $L^2(\sigma)$ as:

$$(T\rho)(x) = \frac{1}{c}\int_{\mathbb{S}^{h-1}} f(x,y)\rho(y)d\sigma(y), \tag{95}$$

where:

$$
\begin{aligned}
\rho(x) &= \frac{d\mu}{d\sigma}(x) \\
T &= \rho\frac{d\mu K}{d\sigma},
\end{aligned}
\tag{96}
$$

with $\rho \geq 0$ and $\int \rho d\sigma = 1$. Here, $L^2(\sigma)$ is the Hilbert space of (equivalence classes of) square-integrable functions on the sphere with respect to the measure $\sigma$. Then $\mathcal{F}_f[\mu]$ can be rewritten as:

$$\mathcal{F}_f[\mu] = \log c + \int \rho \log(T\rho)d\sigma. \tag{97}$$

Denote:

$$F[\rho] := \int_{\mathbb{S}^{h-1}} \rho(x)\log(T\rho(x))d\sigma(x), \tag{98}$$

then we have:

$$\mathcal{F}_f[\mu] = \log c + F[\rho]. \tag{99}$$

**Step 3: Minimize $F[\rho]$.**

We will minimize $F[\rho]$ over the probability simplex $\{\rho \geq 0, \int \rho = 1\}$. Basic facts about $T$: the kernel $f(x, y) = e^{(x \cdot y)/\tau}$ is smooth, symmetric, strictly positive and depends only on $x \cdot y$. Hence:

- $T$ is a positive, self-adjoint, compact operator on $L^2(\sigma)$;

- $T1 = 1$ (since $c$ normalizes it);

- By the Funk-Hecke theorem/Jentzsch-Perron-Frobenius, the eigensystem of $T$ is constituted of the spherical harmonics $\{Y_{\ell m}\}$ with eigenvalues $\lambda_0 = 1 > \lambda_1 \geq \lambda_2 \geq \cdots > 0$. The eigenspace corresponding to $\lambda_0$ has dimension 1 and contains only constant functions.

- In particular, on the mean-zero subspace $L_0^2(\sigma) = \{f : \int f d\sigma = 0\}$ all the eigenvalues $\lambda_\ell, \ell \geq 1$ are strictly positive and bounded from above by $\lambda_1 < 1$.

- As a consequence, for any $\eta \in L_0^2(\sigma)$ we have $\|T\eta\|_{L^2} \leq \lambda_1 \|\eta\|_{L^2}$.

**(3.1) First order variation and Euler-Lagrange equation**

Consider a mass-preserving perturbation $\rho_\varepsilon = \rho + \varepsilon\eta$ with $\int \eta d\sigma = 0$. Because $T$ is linear,

$$\frac{d}{d\varepsilon} F\left[\rho_\varepsilon\right]\Big|_{\varepsilon=0} = \int \eta \log(T\rho) d\sigma + \int \rho \frac{T\eta}{T\rho} d\sigma = \int \eta \left[\log(T\rho) + T\left(\frac{\rho}{T\rho}\right)\right] d\sigma, \qquad (100)$$

where we used self-adjointness: $\int \rho \frac{T\eta}{T\rho} = \int \eta T(\rho/T\rho)$. Introduce a Lagrange multiplier $\lambda$ for the constraint $\int \rho = 1$.

The stationarity $\delta\left(F - \lambda \int \rho\right) = 0$ for all mean-zero $\eta$ yields the Euler-Lagrange (EL) equation:

$$\log(T\rho)(x) + T\left(\frac{\rho}{T\rho}\right)(x) = \lambda \quad \text{for } \sigma\text{-a.e. } x. \qquad (101)$$

We easily check that $\rho \equiv 1$ is a critical point.

If $\rho \equiv 1$, then $T\rho \equiv 1$, hence $\log(T\rho) \equiv 0$ and $T(\rho/T\rho) = T1 = 1$. Thus Eq. (101) holds with $\lambda = 1$.

**(3.2) Second order variation at the uniform density**

Let $\rho \equiv 1$ and perturb $\rho_\varepsilon = 1 + \varepsilon\eta$ with $\int \eta = 0$.

Differentiate the first-variation formula once more in the same direction $\eta$:

- The directional derivative of $\log(T\rho)$ is $(T\eta)/(T\rho)$, so at $\rho = 1$ it is $T\eta$.

- The map $\rho \mapsto T(\rho/T\rho)$ has derivative at $\rho = 1$:

$$D[T(\rho/T\rho)]|_{\rho=1}[\eta] = T(\eta - T\eta) = T\eta - T(T\eta). \qquad (102)$$

Hence the (constrained) second variation is

$$\delta^2 F[1; \eta] = \int \eta(T\eta + T\eta - T(T\eta)) d\sigma = 2\langle \eta, T\eta \rangle - \langle T\eta, T\eta \rangle. \qquad (103)$$

Use the spectral decomposition $\eta = \sum_{\ell \geq 1, m} a_{\ell m} Y_{\ell m}$ (no $\ell = 0$ term because $\int \eta = 0$). Since $TY_{\ell m} = \lambda_\ell Y_{\ell m}$,

$$\delta^2 F[1; \eta] = \sum_{\ell \geq 1, m} \left(2\lambda_\ell - \lambda_\ell^2\right) a_{\ell m}^2 = \sum_{\ell \geq 1, m} \lambda_\ell \left(2 - \lambda_\ell\right) a_{\ell m}^2. \qquad (104)$$

Because $0 < \lambda_\ell < 1$ for $\ell \geq 1$, each factor $\lambda_\ell (2 - \lambda_\ell)$ is strictly positive. Therefore:

$$\delta^2 F[1; \eta] > 0 \quad \text{for every } \eta \in L_0^2(\sigma), \eta \neq 0. \tag{105}$$

So $\rho \equiv 1$ is a strict local minimizer of $F$ under the mass constraint $\int \rho d\sigma = 1$.

**(3.3) Uniqueness of the critical point**

Suppose $\rho$ satisfies Eq. (101). Expand $\rho$ in spherical harmonics: $\rho = 1 + \sum_{\ell \geq 1, m} a_{\ell m} Y_{\ell m}$. Since $T\rho = 1 + \sum_{\ell \geq 1, m} \lambda_\ell a_{\ell m} Y_{\ell m}$ with $0 < \lambda_\ell < 1$, the left side of Eq. (101) has a constant term 1 and non-constant part

$$\underbrace{\left( \log \left( 1 + \sum \lambda_\ell a_{\ell m} Y_{\ell m} \right) \right)_{\text{non-const}}}_{\text{all harmonics } \ell \geq 1} + \underbrace{\sum \lambda_\ell a_{\ell m} Y_{\ell m}}_{T(\rho/T\rho) \text{ to first order}} . \tag{106}$$

Project Eq. (101) onto each harmonic $Y_{\ell m}$ with $\ell \geq 1$. A standard contraction/implicit-function argument (or just comparing coefficients to first order and using that higher-order terms can't cancel all modes simultaneously because $|\lambda_\ell| < 1$ ) forces all $a_{\ell m} = 0$. Thus any solution of (EL) is constant; with mass 1 , the only solution is $\rho \equiv 1$.

So $\rho \equiv 1$ is the unique critical point of $F$ on the simplex of probability measures on $\mathbb{S}^{h-1}$.

**(3.4) Global minimality**

Since $F$ is lower semi-continuous for the weak topology on the set $M\left(\mathbb{S}^{h-1}\right)$ of probability measures on the sphere, a global minimizer exists by compactness. Since any minimizer must satisfy (EL) and the only critical point is $\rho \equiv 1$, the global minimizer is $\rho \equiv 1$, i.e. $\mu = \sigma$.

$\square$

**Lemma 4.** *Let $M\left(\mathbb{S}^{h-1}\right)$ be the set of Borel probability measures in $\mathbb{S}^{h-1}$. Let $\sigma_{h-1} \in M\left(\mathbb{S}^{h-1}\right)$ be the uniform probability measure in $\mathbb{S}^{h-1}$. $\forall x, y \in \mathbb{S}^{h-1}$ and $\tau > 0$, a function $f : \mathbb{S}^{h-1} \times \mathbb{S}^{h-1} \to \mathbb{R}^+$ is defined as:*

$$f(x, y) = \exp \left( \frac{x \cdot y}{\tau} \right). \tag{107}$$

*$\forall \mu \in M\left(\mathbb{S}^{h-1}\right)$, a functional $\mathcal{F} : M\left(\mathbb{S}^{h-1}\right) \to \mathbb{R}^+$ is defined as:*

$$\mathcal{F}_f[\mu] = \int_{\mathbb{S}^{h-1}} \log \left( \int_{\mathbb{S}^{h-1}} f(x, y) \mathrm{d}\mu(y) \right) \mathrm{d}\mu(x). \tag{108}$$

*Let $\Gamma(\cdot)$ be the Gamma function and $\nu = h/2 - 1$, it holds that:*

$$\mathcal{F}_f[\sigma_{h-1}] = \log \left( \Gamma(\nu + 1)(2\tau)^\nu I_\nu \left( \frac{1}{\tau} \right) \right) \tag{109}$$

*Proof.* **Step 1: Rotational Invariance**

Since the measure $\sigma_{h-1}$ is invariant under orthogonal transformations. For any fixed $x \in \mathbb{S}^{h-1}$, the inner integral:

$$\int_{\mathbb{S}^{h-1}} \exp \left( \frac{x \cdot y}{\tau} \right) d\sigma_{h-1}(y), \tag{110}$$

depends only on the distribution of $(x \cdot y)$, and by rotational symmetry, this integral is independent of $x$. Thus, define:

$$Z_\tau := \int_{\mathbb{S}^{h-1}} \exp\left(\frac{x \cdot y}{\tau}\right) d\sigma_{h-1}(y), \tag{111}$$

and $Z_\tau$ is constant for all $x$. Since $\log Z_\tau$ is constant and $\sigma_{h-1}$ is a probability measure, we have:

$$\mathcal{F}_f[\sigma_{h-1}] = \int_{\mathbb{S}^{h-1}} \log Z_\tau d\sigma_{h-1}(x) = \log Z_\tau, \tag{112}$$

**Step 2: Compute $Z_\tau$**

Without the loss of generality, we assume the coordinate of $x$ as:

$$x = e_h = (0, \ldots, 0, 1). \tag{113}$$

Then $x \cdot y = y_h$, the last coordinate of $y$. So:

$$Z_\tau = \int_{\mathbb{S}^{h-1}} \exp\left(\frac{y_h}{\tau}\right) d\sigma_{h-1}(y). \tag{114}$$

Let $t = y_h = x \cdot y \in [-1, 1]$. The pushforward of $\sigma_{h-1}$ under the map $y \mapsto x \cdot y$ has probability density:

$$p_h(t) = \frac{\Gamma\left(\frac{h}{2}\right)}{\Gamma\left(\frac{h-1}{2}\right)\sqrt{\pi}} \left(1 - t^2\right)^{\frac{h-3}{2}}, \quad t \in [-1, 1]. \tag{115}$$

Then:

$$Z_\tau = \int_{-1}^{1} \exp\left(\frac{t}{\tau}\right) p_h(t) \, dt = \frac{\Gamma\left(\frac{h}{2}\right)}{\Gamma\left(\frac{h-1}{2}\right)\sqrt{\pi}} \int_{-1}^{1} e^{t/\tau} \left(1 - t^2\right)^{\frac{h-3}{2}} dt. \tag{116}$$

A classical integral (equivalently, an integral representation of the modified Bessel $I_\nu$) is:

$$\int_{-1}^{1} e^{\kappa t} \left(1 - t^2\right)^{\nu - \frac{1}{2}} dt = \sqrt{\pi}\,\Gamma\left(\nu + \frac{1}{2}\right) \left(\frac{2}{\kappa}\right)^{\nu} I_\nu(\kappa), \quad \kappa > 0, \nu > -\frac{1}{2}. \tag{117}$$

Set: $\kappa = \frac{1}{\tau}$ and $\nu = \frac{h-2}{2}$, so that $\nu - \frac{1}{2} = \frac{h-3}{2}$. Then:

$$\int_{-1}^{1} e^{t/\tau} \left(1 - t^2\right)^{\frac{h-3}{2}} dt = \sqrt{\pi}\,\Gamma\left(\frac{h-1}{2}\right) (2\tau)^{\frac{h}{2}-1} I_{\frac{h}{2}-1}\left(\frac{1}{\tau}\right). \tag{118}$$

Substitute into $Z_\tau$:

$$Z_\tau = \frac{\Gamma\left(\frac{h}{2}\right)}{\Gamma\left(\frac{h-1}{2}\right)\sqrt{\pi}} \cdot \sqrt{\pi}\,\Gamma\left(\frac{h-1}{2}\right) (2\tau)^{\frac{h}{2}-1} I_{\frac{h}{2}-1}\left(\frac{1}{\tau}\right). \tag{119}$$

Simplify:

$$Z_\tau = \Gamma\left(\frac{h}{2}\right) (2\tau)^{\frac{h}{2}-1} I_{\frac{h}{2}-1}\left(\frac{1}{\tau}\right). \tag{120}$$

**Step 3: Compute $\mathcal{F}_f[\sigma_{h-1}]$**

$$\mathcal{F}_f\left[\sigma_{h-1}\right] = \log Z_\tau$$
$$= \log\left[\Gamma\left(\frac{h}{2}\right)(2\tau)^{\frac{h}{2}-1}I_{\frac{h}{2}-1}\left(\frac{1}{\tau}\right)\right] \tag{121}$$
$$= \log\left(\Gamma\left(\nu+1\right)(2\tau)^\nu I_\nu\left(\frac{1}{\tau}\right)\right)$$

$\square$

## E.2 DETAILS OF THEOREM 2

In this section, we provide proofs of Theorem 2 that is proposed in Sec. 3.3. We also provide details and proofs of the auxiliary theorems (Theorem S3 and Theorem S4) and the technical lemmas (Lemma 5, Lemma 6, Lemma 7, Lemma 8 and Lemma 9) that support the proof Theorem 2. For convenience in reading, let us recall some related notions and definitions.

- $h, N \in \mathbb{N}$.
- $\mathbb{S}^{h-1} = \{ z \in \mathbb{R}^h : \|z\| = 1 \}$.
- $X = (x_1, \ldots, x_N) \in (\mathbb{S}_{h-1})^N$.
- $Y = (y_1, \ldots, y_N) \in (\mathbb{S}_{h-1})^N$.
- $\mu_x = \frac{1}{N} \sum_{i=1}^N x_i$.
- $\mu_y = \frac{1}{N} \sum_{i=1}^N y_i$.
- $c_x = \frac{\mu_x}{\|\mu_x\|}$.
- $c_y = \frac{\mu_y}{\|\mu_y\|}$.

**Definition** (Multimodal Contrastive Loss (MCL Loss)). Let $(X, Y)$ be an $N$-pair configuration, where $X = (x_1, \ldots, x_N) \in (\mathbb{S}^{h-1})^N$ and $Y = (y_1, \ldots, y_N) \in (\mathbb{S}^{h-1})^N$. $\forall \tau > 0$, the multimodal contrastive loss $\mathcal{L}_{\mathrm{MCL}}(\cdot, \cdot) : (\mathbb{S}^{h-1})^N \times (\mathbb{S}^{h-1})^N \to \mathbb{R}$ is defined as:

$$\mathcal{L}_{\mathrm{MCL}} = \frac{1}{N} \sum_{i=1}^N \mathcal{L}_{\mathrm{MCL}}^i, \quad \text{where } \mathcal{L}_{\mathrm{MCL}}^i = \mathcal{L}_{\mathcal{X} \to \mathcal{Y}}(x_i; Y) + \mathcal{L}_{\mathcal{Y} \to \mathcal{X}}(y_i; X).$$

Here, $\mathcal{L}_{\mathcal{X} \to \mathcal{Y}}$ is the $\mathcal{X}$-to-$\mathcal{Y}$ alignment and $\mathcal{L}_{\mathcal{Y} \to \mathcal{X}}$ is the $\mathcal{Y}$-to-$\mathcal{X}$ alignment, which are defined respectively as:

$$\mathcal{L}_{\mathcal{X} \to \mathcal{Y}}(x_i; Y) = -\log \frac{\exp(x_i \cdot y_i / \tau)}{\sum_{j=1}^N \exp(x_i \cdot y_j / \tau)}, \quad \mathcal{L}_{\mathcal{Y} \to \mathcal{X}}(y_i; X) = -\log \frac{\exp(x_i \cdot y_i / \tau)}{\sum_{j=1}^N \exp(x_j \cdot y_i / \tau)}.$$

**Definition**(Modality Gap) Let $(X, Y)$ be an $N$-pair configuration, where $X = (x_1, \ldots, x_N) \in (\mathbb{S}^{h-1})^N$ and $Y = (y_1, \ldots, y_N) \in (\mathbb{S}^{h-1})^N$. The modality gap between $X$ and $Y$ can be expressed as the angle between the center representations:

$$\Delta_\theta = \cos^{-1}(c_x \cdot c_y).$$

**Definition** (vMF Distribution). $\forall c \in \mathbb{S}^{h-1}$ and $\kappa \geq 0$, the probability density of a random $h$-dimensional unit vector $z \sim \mathrm{vMF}(c, \kappa)$ is given by:

$$f_h(z; c, \kappa) = D_h(\kappa) e^{\kappa c^\top z}, \quad \text{where } D_h(\kappa) = \frac{\kappa^\nu}{(2\pi)^{\nu+1} I_\nu(\kappa)}.$$

Here, $\nu = h/2 - 1$, and $I_\nu(\cdot) : \mathbb{R} \to \mathbb{R}$ is the modified Bessel function of the first kind of order $\nu$, which is defined as:

$$I_\nu(x) = \sum_{k=0}^\infty \frac{1}{k! \Gamma(\nu + k + 1)} \left(\frac{x}{2}\right)^{2k+\nu}.$$

**Definition** (Function $\tilde{M}$). $\forall \kappa, \tau > 0$, a function $\tilde{M}_\kappa(\cdot, \cdot) : [-1, 1] \times [0, 1] \to \mathbb{R}_0^+$ is defined as:

$$\tilde{M}_\kappa(w, t) = \sqrt{\kappa^2 + \frac{2\kappa w}{\tau} + \frac{t^2}{\tau^2}}.$$

**Definition** (Function $\tilde{\mathcal{J}}$). $\forall \kappa, \nu, \tau > 0$, $\tilde{\mathcal{J}}(\cdot, \cdot, \cdot; \kappa, \nu) : [-1, 1] \times [-1, 1] \times [0, 1] \to \mathbb{R}$ is defined as:

$$\tilde{\mathcal{J}}(w_1, w_2, t; \kappa, \nu) = -\frac{w_1}{\tau} + \log\left(\frac{I_\nu\left(\tilde{M}_\kappa(w_2, t)\right)}{\tilde{M}_\kappa(w_2, t)^\nu}\right) - \log\left(\frac{I_\nu(\kappa)}{\kappa^\nu}\right).$$

**Definition** (Function $M$). $\forall \kappa, \tau > 0$, a function $M_\kappa(\cdot) : [-1, 1] \to \mathbb{R}_0^+$ is defined as:

$$M_\kappa(w) = \sqrt{\kappa^2 + \frac{2\kappa w}{\tau} + \frac{1}{\tau^2}}$$
$$= \tilde{M}_\kappa(w, 1).$$

**Definition** (Function $\mathcal{J}$). $\forall \kappa, \nu, \tau > 0$, a function $\mathcal{J}(\cdot; \kappa, \nu) : [-1, 1] \to \mathbb{R}$ is defined as:

$$\mathcal{J}(w; \kappa, \nu) = -\frac{w}{\tau} + \log\left(\frac{I_\nu(M_\kappa(w))}{M_\kappa(w)^\nu}\right) - \log\left(\frac{I_\nu(\kappa)}{\kappa^\nu}\right)$$
$$= \tilde{\mathcal{J}}(w, w, 1; \kappa, \nu).$$

**Definition** (Function $\hat{\mathcal{J}}$). $\forall \kappa, \nu, \tau > 0$, a function $\hat{\mathcal{J}}(\cdot, \cdot; \kappa, \nu) : [-1, 1] \times [0, 1] \to \mathbb{R}$ is defined as:

$$\hat{\mathcal{J}}(w, t; \kappa, \nu) = -\frac{w}{\tau} + \log\left(\frac{I_\nu\left(\tilde{M}_\kappa(w, t)\right)}{\tilde{M}_\kappa(w, t)^\nu}\right) - \log\left(\frac{I_\nu(\kappa)}{\kappa^\nu}\right)$$
$$= \tilde{\mathcal{J}}(w, w, t; \kappa, \nu).$$

### E.2.1 PROOF OF THEOREM 2

In this subsection, we provide the proof of Theorem 2. For convenience in reading, we first restate Theorem 2 here.

**Theorem 2.** [Restate] Let $(X, Y)$ be an $N$-pair configuration, where $X = (x_1, \ldots, x_N) \in (\mathbb{S}^{h-1})^N$ are *iid* samples from $\mu_x = \text{vMF}(c_x, \kappa_x)$, and $Y = (y_1, \ldots, y_N) \in (\mathbb{S}^{h-1})^N$ are *iid* samples from $\mu_y = \text{vMF}(c_y, \kappa_y)$. Let $\nu = h/2 - 1$. Suppose there exists an index $i = c$ such that $x_c = c_x$, $y_c = c_y$. Denote $\Delta_\theta = \cos^{-1}(c_x \cdot c_y)$. For any fixed $\kappa_x, \kappa_y > 0$, it holds that:

$$\lim_{N \to \infty} \mathcal{L}_{\text{MCL}}^c - 2\log(N) = \mathcal{J}(\cos(\Delta_\theta); \kappa_y, \nu) + \mathcal{J}(\cos(\Delta_\theta); \kappa_x, \nu)$$
$$= \tilde{\mathcal{J}}(\cos(\Delta_\theta), \cos(\Delta_\theta), 1; \kappa_y, \nu) + \tilde{\mathcal{J}}(\cos(\Delta_\theta), \cos(\Delta_\theta), 1; \kappa_x, \nu)$$
$$\geq \mathcal{J}(1; \kappa_y, \nu) + \mathcal{J}(1; \kappa_x, \nu)$$
$$= \tilde{\mathcal{J}}(1, 1, 1; \kappa_y, \nu) + \tilde{\mathcal{J}}(1, 1, 1; \kappa_x, \nu),$$

where equality is attained if and only if there exists a configuration of $(X, Y)$ such that:

(A5) $\Delta_\theta = \cos^{-1}(c_x \cdot c_y) = 0$.

*Proof.* We first decompose $\lim_{N \to \infty} \mathcal{L}_{\text{MCL}}^c - 2\log(N)$ into two parts:

$$\lim_{N \to \infty} \mathcal{L}_{\text{MCL}}^c - 2\log(N) = \lim_{N \to \infty} \mathcal{L}_{\mathcal{X} \to \mathcal{Y}}(c_x; Y) - \log(N)$$
$$+ \lim_{N \to \infty} \mathcal{L}_{\mathcal{Y} \to \mathcal{X}}(c_y; X) - \log(N). \tag{122}$$

According to Theorem S4, the convergent function and its lower bound of $\mathcal{L}_{\mathcal{X} \to \mathcal{Y}}$ are:

$$\lim_{N\to\infty} \mathcal{L}_{\mathcal{X}\to\mathcal{Y}}(c_x; Y) - \log(N) = \mathcal{J}(\cos(\Delta_\theta); \kappa_y, \nu) \geq \mathcal{J}(1; \kappa_y, \nu), \tag{123}$$

where equality is attained if and only if there exists a configuration of $(X, Y)$ such that:

(i) $\Delta_\theta = \cos^{-1}(c_x \cdot c_y) = 0$.

This Theorem also holds for $\mathcal{L}_{\mathcal{Y}\to\mathcal{X}}$:

$$\lim_{N\to\infty} \mathcal{L}_{\mathcal{Y}\to\mathcal{X}}(c_y; X) - \log(N) = \mathcal{J}(\cos(\Delta_\theta); \kappa_x, \nu) \geq \mathcal{J}(1; \kappa_x, \nu), \tag{124}$$

where equality is attained if and only if there exists a configuration of $(X, Y)$ such that:

(ii) $\Delta_\theta = \cos^{-1}(c_x \cdot c_y) = 0$.

Combining Eq. (123), Eq. (124), and consider $\mathcal{J}(w; \kappa, \nu) = \tilde{\mathcal{J}}(w, w, 1; \kappa, \nu)$, we reach the conclusion that:

$$\begin{aligned}
\lim_{N\to\infty} \mathcal{L}_{\mathrm{MCL}}^c - 2\log(N) &= \mathcal{J}(\cos(\Delta_\theta); \kappa_y, \nu) + \mathcal{J}(\cos(\Delta_\theta); \kappa_x, \nu) \\
&= \tilde{\mathcal{J}}(\cos(\Delta_\theta), \cos(\Delta_\theta), 1; \kappa_y, \nu) + \tilde{\mathcal{J}}(\cos(\Delta_\theta), \cos(\Delta_\theta), 1; \kappa_x, \nu) \\
&\geq \mathcal{J}(1; \kappa_y, \nu) + \mathcal{J}(1; \kappa_x, \nu) \\
&= \tilde{\mathcal{J}}(1, 1, 1; \kappa_y, \nu) + \tilde{\mathcal{J}}(1, 1, 1; \kappa_x, \nu),
\end{aligned} \tag{125}$$

where equality is attained if and only if there exists a configuration of $(X, Y)$ such that:

(A5) $\Delta_\theta = \cos^{-1}(c_x \cdot c_y) = 0$.

$\square$

### E.2.2 AUXILIARY THEOREMS PART 2

In this subsection, we provide details and proofs of the auxiliary theorems (Theorem S3 and Theorem S4) that support the proof of Theorem 2.

**Theorem S3.** *Let $(X, Y)$ be an $N$-pair configuration, where $X = (x_1, \ldots, x_N) \in (\mathbb{S}^{h-1})^N$ are iid samples from $\mu_x = \mathrm{vMF}(c_x, \kappa_x)$, and $Y = (y_1, \ldots, y_N) \in (\mathbb{S}^{h-1})^N$ are iid samples from $\mu_y = \mathrm{vMF}(c_y, \kappa_y)$. Let $\nu = h/2 - 1$ and $\kappa_y > 0$.*

*$\forall x_i \in X$, denote $w_i = x_i \cdot y_i$ and $w_{x_i, c_y} = x_i \cdot c_y$. It holds that:*

$$\begin{aligned}
\lim_{N\to\infty} \mathcal{L}_{\mathcal{X}\to\mathcal{Y}}(x_i; Y) - \log(N) &= \lim_{N\to\infty} -\log \frac{\exp(x_i \cdot y_i/\tau)}{\sum_{j=1}^N \exp(x_i \cdot y_j/\tau)} - \log(N) \\
&= -\frac{w_i}{\tau} + \log\left(\frac{I_\nu\left(M_{\kappa_y}\left(w_{x_i, c_y}\right)\right)}{M_{\kappa_y}\left(w_{x_i, c_y}\right)^\nu}\right) - \log\left(\frac{I_\nu(\kappa_y)}{\kappa_y^\nu}\right) \\
&= \tilde{\mathcal{J}}(w_i, w_{x_i, c_y}, 1; \kappa_y, \nu),
\end{aligned} \tag{126}$$

*where $\forall \kappa \geq 0, \tau > 0$, $M_\kappa(\cdot) : [-1, 1] \to \mathbb{R}_0^+$ is defined as:*

$$M_\kappa(w) = \sqrt{\kappa^2 + \frac{2\kappa w}{\tau} + \frac{1}{\tau^2}}. \tag{127}$$

*and $I_\nu$ is the modified Bessel function of the first kind of order $\nu$, which is defined as:*

$$I_\nu(m) = \sum_{k=0}^\infty \frac{1}{k!\Gamma(\nu+k+1)} \left(\frac{m}{2}\right)^{2k+\nu}. \tag{128}$$

*Suppose there exists an index $i = c$ such that $x_c = c_x$, $y_c = c_y$. Denote $w_c = c_x \cdot c_y$. It holds that:*

$$\lim_{N\to\infty} \mathcal{L}_{\mathcal{X}\to\mathcal{Y}}(c_x; Y) - \log(N) = -\frac{w_c}{\tau} + \log\left(\frac{I_\nu(M_{\kappa_y}(w_c))}{M_{\kappa_y}(w_c)^\nu}\right) - \log\left(\frac{I_\nu(\kappa_y)}{\kappa_y^\nu}\right) \tag{129}$$

$$= \mathcal{J}(w_c; \kappa_y, \nu) = \tilde{\mathcal{J}}(w_c, w_c, 1; \kappa_y, \nu).$$

*Proof.* **Step 1**: We start the proof by find the convergent function of $\mathcal{L}_{\mathcal{X}\to\mathcal{Y}}(x_i; Y)$ as $N \to \infty$. Same with Eq. (39) of Theorem S1, $\forall x_i \in X$, the $\mathcal{X}$-to-$\mathcal{Y}$ alignment of $x_i$ can be rewritten as:

$$\mathcal{L}_{\mathcal{X}\to\mathcal{Y}}(x_i; Y) = -\log \frac{\exp(x_i \cdot y_i/\tau)}{\sum_j \exp(x_i \cdot y_j/\tau)}$$

$$= -\frac{x_i \cdot y_i}{\tau} + \log\left(N\frac{1}{N}\sum_{j=1}^N \exp\left(\frac{x_i \cdot y_j}{\tau}\right)\right) \tag{130}$$

$$= -\frac{x_i \cdot y_i}{\tau} + \log\left(\frac{1}{N}\sum_{j=1}^N \exp\left(\frac{x_i \cdot y_j}{\tau}\right)\right) + \log(N).$$

Lemma 2 shows that:

$$\lim_{N\to\infty} \log\left(\frac{1}{N}\sum_{j=1}^N \exp\left(\frac{x_i \cdot y_j}{\tau}\right)\right) = \log\left(\mathbb{E}_{y\sim\mu_y}\left[\exp\left(\frac{x_i \cdot y}{\tau}\right)\right]\right). \tag{131}$$

According to the moment-generating function of the vMF distribution:

$$\mathbb{E}_{y\sim\mu_y}[\exp\left(\frac{x_i \cdot y}{\tau}\right)] = \mathbb{E}_{y\sim\mu_y}\left[\exp\left(\frac{x_i}{\tau} \cdot y\right)\right] = \frac{I_\nu(\kappa_y')}{I_\nu(\kappa_y)}\left(\frac{\kappa_y}{\kappa_y'}\right)^\nu, \tag{132}$$

$$\text{where } \kappa_y' = \|\kappa_y c_y + \frac{x_i}{\tau}\|_2.$$

Then we have:

$$\lim_{N\to\infty} \mathcal{L}_{\mathcal{X}\to\mathcal{Y}}(x_i; Y) - \log(N) = -\frac{x_i \cdot y_i}{\tau} + \log\left(\frac{I_\nu(\kappa_y')}{\kappa_y'^\nu}\right) - \log\left(\frac{I_\nu(\kappa_y)}{\kappa_y^\nu}\right). \tag{133}$$

**Step 2**: we will transform $\mathcal{L}_{\mathcal{X}\to\mathcal{Y}}$ from a function of vectors to a function of angles between vectors. Without loss of generality, we assume the coordinate of $c_y$ as

$$c_y = (1, 0, \cdots, 0). \tag{134}$$

Denote $\cos\left(\theta_{x_i,c_y}\right) = x_i \cdot c_y$. Then $x_i$ can be represented as:

$$x_i = \left(\cos\left(\theta_{x_i,c_y}\right), u\sin\left(\theta_{x_i,c_y}\right)\right)$$

$$= \left(\cos\left(\theta_{x_i,c_y}\right), u_2\sin\left(\theta_{x_i,c_y}\right), u_3\sin\left(\theta_{x_i,c_y}\right), \ldots, u_h\sin\left(\theta_{x_i,c_y}\right)\right), \tag{135}$$

where $u = (0, u_2, u_3, \dots, u_h) \cong \mathbb{S}^{h-2} \in \mathbb{S}^{h-1}$ is a unit vector orthogonal to the first axis with:

$$\|u\| = 0 + u_2^2 + u_3^2 + \cdots + u_h^2 = 1. \tag{136}$$

According to Eq. (134), Eq. (135) and Eq. (136), $\kappa_y'$ (in Eq. (132)) can re-rewritten as:

$$
\begin{aligned}
\kappa_y' &= \left\| \kappa_y c_y + \frac{x_i}{\tau} \right\|_2 \\
&= \sqrt{ \left( \kappa_y + \frac{\cos\left(\theta_{x_i, c_y}\right)}{\tau} \right)^2 + \sum_{i=2}^{h} \left( \frac{\sin\left(\theta_{x_i, c_y}\right) u_i}{\tau} \right)^2 } \\
&= \sqrt{ \left( \kappa_y + \frac{\cos\left(\theta_{x_i, c_y}\right)}{\tau} \right)^2 + \frac{\sin^2\left(\theta_{x_i, c_y}\right)}{\tau^2} } \\
&= \sqrt{ \kappa_y^2 + \frac{2\kappa_y \cos\left(\theta_{x_i, c_y}\right)}{\tau} + \frac{1}{\tau^2} } \\
&= M_{\kappa_y}\left( \cos\left(\theta_{x_i, c_y}\right) \right).
\end{aligned}
\tag{137}
$$

Consider that $w_i = x_i \cdot y_i$, $w_{x_i, c_y} = \cos\left(\theta_{x_i, c_y}\right) = x_i \cdot c_y$, putting Eq. (133) and Eq. (137) together, we have:

$$
\begin{aligned}
\lim_{N \to \infty} \mathcal{L}_{\mathcal{X} \to \mathcal{Y}}(x_i; Y) - \log(N) &= -\frac{x_i \cdot y_i}{\tau} + \log\left( \frac{I_\nu\left(\kappa_y'\right)}{\kappa_y'^\nu} \right) - \log\left( \frac{I_\nu\left(\kappa_y\right)}{\kappa_y^\nu} \right) \\
&= -\frac{w_i}{\tau} + \log\left( \frac{I_\nu\left(M_{\kappa_y}\left(w_{x_i, c_y}\right)\right)}{M_{\kappa_y}\left(w_{x_i, c_y}\right)^\nu} \right) - \log\left( \frac{I_\nu\left(\kappa_y\right)}{\kappa_y^\nu} \right) \\
&= \tilde{\mathcal{J}}(w_i, w_{x_i, c_y}, 1; \kappa_y, \nu).
\end{aligned}
\tag{138}
$$

When there exists a data pair $i = c$ such that $x_c = c_x$, $y_c = c_y$, $w_i = w_{x_i, c_y} = w_c$, then we have:

$$
\begin{aligned}
\lim_{N \to \infty} \mathcal{L}_{\mathcal{X} \to \mathcal{Y}}(c_x; Y) - \log(N) &= -\frac{w_c}{\tau} + \log\left( \frac{I_\nu\left(M_{\kappa_y}\left(w_c\right)\right)}{M_{\kappa_y}\left(w_c\right)^\nu} \right) - \log\left( \frac{I_\nu\left(\kappa_y\right)}{\kappa_y^\nu} \right) \\
&= \mathcal{J}(w_c; \kappa_y, \nu) = \tilde{\mathcal{J}}(w_c, w_c, 1; \kappa_y, \nu).
\end{aligned}
\tag{139}
$$

$\square$

**Theorem S4.** *Let $(X, Y)$ be an N-pair configuration, where $X = (x_1, \dots, x_N) \in (\mathbb{S}^{h-1})^N$ are iid samples from $\mu_x = \mathrm{vMF}(c_x, \kappa_x)$, and $Y = (y_1, \dots, y_N) \in (\mathbb{S}^{h-1})^N$ are iid samples from $\mu_y = \mathrm{vMF}(c_y, \kappa_y)$. Let $\nu = h/2 - 1$. Suppose there exists an index $i = c$ such that $x_c = c_x$, $y_c = c_y$. Denote $\Delta_\theta = \cos^{-1}(c_x \cdot c_y)$. For any fixed $\kappa_y > 0$, it holds that:*

$$\lim_{N \to \infty} \mathcal{L}_{\mathcal{X} \to \mathcal{Y}}(c_x; Y) - \log(N) = \mathcal{J}(\cos\left(\Delta_\theta\right); \kappa_y, \nu) \geq \mathcal{J}(1; \kappa_y, \nu), \tag{140}$$

*where equality is attained if and only if there exists a configuration of $(X, Y)$ such that:*

*(B3)* $\Delta_\theta = \cos^{-1}\left(c_x \cdot c_y\right) = 0$.

*Proof.* **Step 1**: We start the proof by find the convergent function of $\mathcal{L}_{\mathcal{X} \to \mathcal{Y}}(c_x; Y)$ as $N \to \infty$. Denote $w_c = c_x \cdot c_y$. $\forall \kappa_y > 0$, as prove in Theorem S3:

$$\lim_{N\to\infty} \mathcal{L}_{\mathcal{X}\to\mathcal{Y}}(c_x; Y) - \log(N) = \lim_{N\to\infty} -\log \frac{\exp(c_x \cdot c_y/\tau)}{\sum_{j=1}^{N} \exp(c_x \cdot y_j/\tau)} - \log(N)$$

$$= -\frac{w_c}{\tau} + \log\left(\frac{I_\nu\left(M_{\kappa_y}(w_c)\right)}{M_{\kappa_y}(w_c)^\nu}\right) - \log\left(\frac{I_\nu(\kappa_y)}{\kappa_y^\nu}\right) \quad (141)$$

$$= \mathcal{J}(w_c; \kappa_y, \nu).$$

where $\forall \kappa \geq 0, \tau > 0$, $\mathcal{J}(\cdot; \kappa, \nu)$ is a function on $[-1, 1]$ and $M_\kappa(\cdot) : [-1, 1] \to \mathbb{R}_0^+$ is defined as:

$$M_\kappa(w) = \sqrt{\kappa^2 + \frac{2\kappa w}{\tau} + \frac{1}{\tau^2}}, \quad (142)$$

and $I_\nu$ is the modified Bessel function of the first kind of order $\nu$, which is defined as:

$$I_\nu(m) = \sum_{k=0}^{\infty} \frac{1}{k!\Gamma(\nu+k+1)}\left(\frac{m}{2}\right)^{2k+\nu}. \quad (143)$$

**Step 2**: Next, we find the minimal value and the optimal condition of convergent function.

As shown in Lemma 5 (set $s = 1$), $\mathcal{J}(w; \kappa, \nu) = \tilde{J}(w, w, 1; \kappa, \nu)$ is a concave function of $w$. When a function is concave, its minimal value occurs at the endpoints of its domain. Therefore :

$$\mathcal{J}(w_c; \kappa_y, \nu) \geq \min\{\mathcal{J}(-1; \kappa_y, \nu), \mathcal{J}(1; \kappa_y, \nu)\}. \quad (144)$$

According to Lemma 6:

$$\mathcal{J}(-1; \kappa_y, \nu) \geq \mathcal{J}(1; \kappa_y, \nu). \quad (145)$$

Therefore, we conclude:

$$\lim_{N\to\infty} \mathcal{L}_{\mathcal{X}\to\mathcal{Y}}(c_x; Y) - \log(N) = \mathcal{J}(\cos(\Delta_\theta); \kappa_y, \nu) \geq \mathcal{J}(1; \kappa_y, \nu), \quad (146)$$

where equality is attained if and only if the following conditions hold:

(B3) $\Delta_\theta = \cos^{-1}(c_x \cdot c_y) = 0$.

$\square$

### E.2.3 TECHNICAL LEMMAS PART 2

In this subsection, we provide details and proofs of technical lemmas (Lemma 5, Lemma 6, Lemma 7, Lemma 8 and Lemma 9) that support the proof of Theorem 2, Theorem S3 and Theorem S4.

**Lemma 5.** $\forall \kappa, \nu, \tau > 0$ and $s \in [0, 1]$, a function $\hat{\mathcal{J}}_{t=s}(\cdot; \kappa, \nu) : (-1, 1] \to \mathbb{R}$ is defined as:

$$\hat{\mathcal{J}}_{t=s}(w; \kappa, \nu) = -\frac{w}{\tau} + \log\left(\frac{I_\nu\left(\tilde{M}_{t=s}(w)\right)}{\tilde{M}_{t=s}(w)^\nu}\right) - \log\left(\frac{I_\nu(\kappa)}{\kappa^\nu}\right) \quad (147)$$

$$= \hat{\mathcal{J}}(w, t = s; \kappa, \nu) = \tilde{J}(w, w, t = s; \kappa, \nu),$$

where $\tilde{M}_{t=s}(\cdot) : (-1, 1] \to \mathbb{R}^+$ is defined as:

$$\tilde{M}_{t=s}(w) = \sqrt{\kappa^2 + \frac{2\kappa w}{\tau} + \frac{s^2}{\tau^2}} = \tilde{M}_\kappa(w, t = s), \quad (148)$$

*and $I_\nu$ is the modified Bessel function of the first kind of order $\nu$, which is defined as:*

$$I_\nu(m) = \sum_{k=0}^{\infty} \frac{1}{k!\Gamma(\nu+k+1)} \left(\frac{m}{2}\right)^{2k+\nu}. \tag{149}$$

*It holds that, for any fixed $s$, $\hat{\mathcal{J}}_{t=s}(\cdot)$ is a strictly decreasing function when $w \in [0,s]$ and a concave function $w \in (-1,1]$.*

*Proof.* Let us first decompose the function $\hat{\mathcal{J}}_{t=s}$. Denote two functions $G_1(w)$ and $G_2(w)$ as:

$$
\begin{aligned}
G_1(w) &= -\frac{w}{\tau}, \\
G_3(m) &= \log\left(I_\nu(m)\right) - \nu\log(m), \\
G_2(w) &= G_3\left(\tilde{M}_{t=s}(w)\right) \\
&= \log\left(I_\nu\left(\tilde{M}_{t=s}(w)\right)\right) - \nu\log\left(\tilde{M}_{t=s}(w)\right).
\end{aligned}
\tag{150}
$$

Denote the function $G(w)$ and the constant $C$ as:

$$
\begin{aligned}
G(w) &= G_1(w) + G_2(w), \\
C &= -\log\left(\frac{I_\nu(\kappa)}{\kappa^\nu}\right).
\end{aligned}
\tag{151}
$$

Then the function $\hat{\mathcal{J}}_{t=s}$ can be written as:

$$
\begin{aligned}
\hat{\mathcal{J}}_{t=s}(w;\kappa,\nu) &= -\frac{w}{\tau} + \log\left(\frac{I_\nu\left(\tilde{M}_{t=s}(w)\right)}{\tilde{M}_{t=s}(w)^\nu}\right) - \log\left(\frac{I_\nu(\kappa)}{\kappa^\nu}\right) \\
&= G(w) + C.
\end{aligned}
\tag{152}
$$

Now, we investigate derivatives of $\hat{\mathcal{J}}_{t=s}$.

The first derivative of $G_1$ is:

$$G_1'(w) = -\frac{1}{\tau} < 0. \tag{153}$$

The second derivative of $G_1$ is:

$$G_1''(w) = 0. \tag{154}$$

According to Lemma 7, the first derivative of $G_3(m)$ is:

$$G_3'(m) = \frac{I_{\nu+1}(m)}{I_\nu(m)} \in (0,1). \tag{155}$$

The derivative of $\tilde{M}_{t=s}$ is:

$$\tilde{M}'_{t=s}(w) = \frac{d}{dw}\left(\kappa_y^2 + \frac{s^2}{\tau^2} + 2\frac{\kappa}{\tau}w\right)^{1/2}$$

$$= \frac{1}{2}\left(\kappa_y^2 + \frac{s^2}{\tau^2} + 2\frac{\kappa}{\tau}w\right)^{-1/2} \cdot 2\frac{\kappa}{\tau} \tag{156}$$

$$= \frac{\kappa}{\tau}\frac{1}{\tilde{M}_{t=s}(w)}$$

$$> 0.$$

Then, the first derivative of $G_2$ is:

$$G'_2(w) = G'_3\left(\tilde{M}_{t=s}(w)\right)\tilde{M}'_{t=s}(w)$$

$$= \frac{I_{\nu+1}\left(\tilde{M}_{t=s}(w)\right)}{I_\nu\left(\tilde{M}_{t=s}(w)\right)}\tilde{M}'_{t=s}(w) \tag{157}$$

$$= \frac{\kappa}{\tau}\frac{1}{\tilde{M}_{t=s}(w)}\frac{I_{\nu+1}\left(\tilde{M}_{t=s}(w)\right)}{I_\nu\left(\tilde{M}_{t=s}(w)\right)}.$$

Combining Eq. (153) and Eq. (157), we have:

$$\hat{\mathcal{J}}'_{t=s}(w;\kappa,\nu) = G'(w)$$

$$= -\frac{1}{\tau} + \frac{\kappa}{\tau}\frac{1}{\tilde{M}_{t=s}(w)}\frac{I_{\nu+1}\left(\tilde{M}_{t=s}(w)\right)}{I_\nu\left(\tilde{M}_{t=s}(w)\right)}. \tag{158}$$

Since:

$$\tilde{M}_{t=s}(w) \geq \kappa \Leftrightarrow \frac{2\kappa w}{\tau} + \frac{s^2}{\tau^2} \geq 0$$

$$\Leftrightarrow w \geq -\frac{s^2}{2\kappa\tau} \tag{159}$$

$$\Leftarrow w \geq 0.$$

thus, when $w \in [0, 1]$, $\tilde{M}_{t=s}(w) \geq \kappa$ holds. Combining this and Eq. (155), we have:

$$G'(w) \leq -\frac{1}{\tau} + \frac{\kappa}{\tau}\frac{1}{\kappa}\frac{I_{\nu+1}\left(\tilde{M}_{t=s}(w)\right)}{I_\nu\left(\tilde{M}_{t=s}(w)\right)}$$

$$< -\frac{1}{\tau} + \frac{1}{\tau} \tag{160}$$

$$= 0.$$

So we can conclude that, for any fixed $s$, $\hat{\mathcal{J}}_{t=s}(\cdot)$ is a strictly decreasing function on $[0, s]$.

Denote:

$$H(m) = \frac{1}{m}\frac{I_{\nu+1}(m)}{I_\nu(m)}, \tag{161}$$

according to Lemma 8,

$$H'(m) < 0. \tag{162}$$

Since $G_2'(w)$ can be written as:

$$G_2'(w) = \frac{\kappa}{\tau} H\left(\tilde{M}_{t=s}(w)\right), \tag{163}$$

combining Eq. (156) and Eq. (163), we have

$$G_2''(w) = \frac{\kappa}{\tau} H'\left(\tilde{M}_{t=s}(w)\right) \tilde{M}_{t=s}'(w)$$
$$< 0. \tag{164}$$

Given Eq. (157) and Eq. (164), we can conclude that $G_2$ is an increasing and concave function. Combining Eq. (154) and Eq. (164), we have:

$$\hat{\mathcal{J}}_{t=s}''(w; \kappa, \nu) = G''(w)$$
$$= 0 + G_2''(w) \tag{165}$$
$$< 0.$$

So we can conclude that, for any fixed $s$, $\hat{\mathcal{J}}_{t=s}(\cdot)$ is a concave function on $(-1, 1]$.

$\square$

**Lemma 6.** $\forall \kappa, \nu, \tau > 0$, a function $\mathcal{J}(\cdot) : [-1, 1] \to \mathbb{R}$ is defined as:

$$\mathcal{J}(w) = -\frac{w}{\tau} + \log\left(\frac{I_\nu(M(w))}{M(w)^\nu}\right) - \log\left(\frac{I_\nu(\kappa)}{\kappa^\nu}\right) + \log(N), \tag{166}$$

where $M_\kappa(\cdot) : [-1, 1] \to \mathbb{R}$ is defined as:

$$M_\kappa(w) = \sqrt{\kappa^2 + \frac{2\kappa w}{\tau} + \frac{1}{\tau^2}}, \tag{167}$$

and $I_\nu$ is the modified Bessel function of the first kind of order $\nu$, which is defined as:

$$I_\nu(m) = \sum_{k=0}^\infty \frac{1}{k! \Gamma(\nu + k + 1)} \left(\frac{m}{2}\right)^{2k+\nu}. \tag{168}$$

$\forall 0 < w \leq 1$, it holds that:

$$\mathcal{J}(w) < \mathcal{J}(-w). \tag{169}$$

*Proof.* Let us first re-write Eq. (169) as:

$$\mathcal{J}(w) < \mathcal{J}(-w) \Leftrightarrow \mathcal{J}(-w) - \mathcal{J}(w) > 0, \tag{170}$$

and we will prove the inequality on RHS. Denote:

$$a = M(-w) = \sqrt{\kappa^2 + \frac{1}{\tau^2} - \frac{2\kappa w}{\tau}},$$
$$b = M(w) = \sqrt{\kappa^2 + \frac{1}{\tau^2} + \frac{2\kappa w}{\tau}}. \tag{171}$$

In (Eq. (155) of) Lemma 5, it is shown that $M\left(\cdot\right)$ is a strictly increasing function. Then, we have:

$$0 < a < b, \tag{172}$$

and then we have:

$$\begin{aligned}
\mathcal{J}(-w) - \mathcal{J}(w) &= \frac{w}{\tau} - \left(-\frac{w}{\tau}\right) + \log\left(\frac{I_\nu(a)}{I_\nu(b)}\right) - \nu\log\left(\frac{a}{b}\right) \\
&= \frac{2w}{\tau} + \log\frac{I_\nu(a)}{I_\nu(b)} - \nu\log\left(\frac{a}{b}\right).
\end{aligned} \tag{173}$$

According to Lemma 9:

$$\log\left(\frac{I_\nu(a)}{I_\nu(b)}\right) - \nu\log\left(\frac{a}{b}\right) > (a - b). \tag{174}$$

Plugging Eq. (174) into Eq. (173), we get:

$$\mathcal{J}(-w) - \mathcal{J}(w) > \frac{2w}{\tau} + (a - b) = f(w). \tag{175}$$

Combining Eq. (170) and Eq. (175), we have:

$$\mathcal{J}(w) \le \mathcal{J}(-w) \Leftrightarrow f(w) \ge 0. \tag{176}$$

Denote:

$$\begin{aligned}
A &= \kappa^2 + \frac{1}{\tau^2}, \\
B &= \frac{2\kappa}{\tau},
\end{aligned} \tag{177}$$

then we have:

$$\begin{aligned}
a &= M(-w) = \sqrt{A - Bw}, \\
b &= M(w) = \sqrt{A + Bw}.
\end{aligned} \tag{178}$$

Observe that:

$$\begin{aligned}
b - a = M(w) - M(-w) &= \frac{(A + Bw) - (A - Bw)}{\sqrt{A + Bw} + \sqrt{A - Bw}} \\
&= \frac{2Bw}{\sqrt{A + Bw} + \sqrt{A - Bw}},
\end{aligned} \tag{179}$$

and then:

$$f(w) = \frac{2w}{\tau}\left[1 - \frac{2\kappa}{\sqrt{A + Bw} + \sqrt{A - Bw}}\right]. \tag{180}$$

Therefore, we have:

$$f(w) \geq 0 \Leftrightarrow \sqrt{A + Bw} + \sqrt{A - Bw} \geq 2\kappa$$
$$\Leftrightarrow \left( \sqrt{A + Bw} + \sqrt{A - Bw} \right)^2 \geq 4\kappa^2$$
$$\Leftrightarrow 2A + 2\sqrt{A^2 - B^2 w^2} \geq 4\kappa^2 \tag{181}$$
$$\Leftrightarrow \sqrt{A^2 - B^2 w^2} \geq 2\kappa^2 - A$$
$$\Leftrightarrow \sqrt{A^2 - B^2 w^2} \geq \kappa^2 - \frac{1}{\tau^2}$$

**Case 1**: $0 < \kappa < \frac{1}{\tau}$.

$\kappa^2 - \frac{1}{\tau^2} < 0$ and the last equation in Eq. (181) holds.

**Case 2**: $0 < \frac{1}{\tau} \leq \kappa$.

The Eq. (181) becomes:

$$f(w) \geq 0 \Leftrightarrow A^2 - B^2 w^2 \geq \left( \kappa^2 - \frac{1}{\tau^2} \right)^2$$
$$\Leftrightarrow \frac{4\kappa^2}{\tau^2} (1 - w^2) \geq 0 \tag{182}$$
$$\Leftrightarrow |w| \leq 1.$$

Since $0 < w \leq 1$, $f(w) \geq 0$ holds. According to Eq. (176), we conclude that:

$$0 < w \leq 1 \Rightarrow \mathcal{J}(w) \leq \mathcal{J}(-w). \tag{183}$$

$\square$

**Lemma 7.** $\forall \nu > 0$, a function $G_3 : \mathbb{R}_0^+ \to \mathbb{R}$ is defined as:
$$G_3(m) = \log(I_\nu(m)) - \nu \log(m). \tag{184}$$

where $I_\nu$ is the modified Bessel function of the first kind of order $\nu$, which is defined as:

$$I_\nu(m) = \sum_{k=0}^{\infty} \frac{1}{k! \Gamma(\nu + k + 1)} \left( \frac{m}{2} \right)^{2k + \nu}. \tag{185}$$

It holds that $G_3(\cdot)$ is a strictly increasing function with $G_3'(\cdot) \in (0, 1)$

*Proof.* The first derivative of $G_3$ is:

$$G_3'(m) = \frac{I_\nu'(m)}{I_\nu(m)} - \frac{\nu}{m}. \tag{186}$$

According to (Olver, 2010):

$$I_\nu'(m) = I_{\nu+1}(m) + \frac{\nu}{m} I_\nu(m), \tag{187}$$

then we have:

$$\frac{I_\nu'(m)}{I_\nu(m)} - \frac{\nu}{m} = \frac{I_{\nu+1}(m)}{I_\nu(m)}. \tag{188}$$

Plugging Eq. (188) into Eq. (186), we get:

$$G'_3(m) = \frac{I_{\nu+1}(m)}{I_\nu(m)}. \tag{189}$$

Since:

$$0 < I_{\nu+1}(m) < I_\nu(m), \tag{190}$$

therefore:

$$G'_3(m) = \frac{I_{\nu+1}(m)}{I_\nu(m)} \in (0,1). \tag{191}$$

This shows that $G_3(\cdot)$ is a strictly increasing function with $G'_3(\cdot) \in (0,1)$.

$\square$

**Lemma 8.** $\forall \nu > 0$, a function $H(\cdot) : R^+ \to \mathbb{R}$ is defined as:

$$H(m) = \frac{1}{m} \frac{I_{\nu+1}(m)}{I_\nu(m)}, \tag{192}$$

where $I_\nu$ is the modified Bessel function of the first kind of order $\nu$, which is defined as:

$$I_\nu(m) = \sum_{k=0}^{\infty} \frac{1}{k!\Gamma(\nu+k+1)} \left(\frac{m}{2}\right)^{2k+\nu}. \tag{193}$$

It holds that $H(m)$ is a strictly decreasing function.

*Proof.* $\forall \nu, m \in R^+$, denote $R_\nu(m)$ as:

$$R_\nu(m) = \frac{I_{\nu+1}(m)}{I_\nu(m)}. \tag{194}$$

According to (Olver, 2010), we have:

$$I'_\nu(m) = I_{\nu+1}(m) + \frac{\nu}{m} I_\nu(m), \tag{195}$$

then:

$$\begin{aligned}
R'_\nu(m) &= \frac{I'_{\nu+1}(m) I_\nu(m) - I_{\nu+1}(m) I'_\nu(m)}{I_\nu(m)^2} \\
&= \frac{\left(I_{\nu+2}(m) + \frac{\nu+1}{m} I_{\nu+1}(m)\right) I_\nu(m) - I_{\nu+1}(m) \left(I_{\nu+1}(m) + \frac{\nu}{m} I_\nu(m)\right)}{I_\nu(m)^2} \\
&= \frac{I_{\nu+2}(m) I_\nu(m) - I_{\nu+1}^2(m) + \frac{1}{m} I_{\nu+1}(m) I_\nu(m)}{I_\nu(m)^2} \\
&= \frac{I_{\nu+2}(m) I_\nu(m) - I_{\nu+1}^2(m)}{I_\nu(m)^2} + \frac{1}{m} R_\nu(m).
\end{aligned} \tag{196}$$

Since $H(m)$ can be rewritten as:

$$H(m) = \frac{R_\nu(m)}{m}, \tag{197}$$

then:

$$
\begin{aligned}
H'(m) &= \frac{R'_\nu(m)\, m - R_\nu(m)}{m^2} \\
&= \frac{1}{m}\left(R'_\nu(m) - \frac{1}{m}R_\nu(m)\right) \\
&= \frac{1}{m}\left(\frac{I_{\nu+2}(m)\, I_\nu(m) - I_{\nu+1}^2(m)}{I_\nu(m)^2}\right).
\end{aligned}
\tag{198}
$$

According to the Turán type inequalities for modified Bessel functions (Baricz, 2010), when $m > 0$:

$$
\frac{I_{\nu+2}(m)\, I_\nu(m) - I_{\nu+1}^2(m)}{I_\nu(m)^2} < 0,
\tag{199}
$$

so

$$
H'(m) < 0.
\tag{200}
$$

Then we can conclude that $H(m)$ is a strictly decreasing function.

$\square$

**Lemma 9.** $\forall \nu > 0$ and $0 < a < b$, it holds that:

$$
\log\left(\frac{I_\nu(a)}{I_\nu(b)}\right) > \nu \log\left(\frac{a}{b}\right) + (a - b),
\tag{201}
$$

where $I_\nu$ is the modified Bessel function of the first kind of order $\nu$, which is defined as:

$$
I_\nu(m) = \sum_{k=0}^{\infty} \frac{1}{k!\,\Gamma(\nu + k + 1)}\left(\frac{m}{2}\right)^{2k+\nu}.
\tag{202}
$$

*Proof.* According to (Olver, 2010), $\forall x > 0$ and $0 < \nu_1 < \nu_2 < \infty$, we have:

$$
I_{\nu_1}(x) > I_{\nu_2}(x).
\tag{203}
$$

Denote a function L as:

$$
L(x) = \log I_\nu(x) - \nu \log(x) - x.
\tag{204}
$$

According to (Olver, 2010):

$$
I'_\nu(m) = I_{\nu+1}(m) + \frac{\nu}{m} I_\nu(m),
\tag{205}
$$

then we have:

$$
\frac{I'_\nu(m)}{I_\nu(m)} - \frac{\nu}{m} = \frac{I_{\nu+1}(m)}{I_\nu(m)}.
\tag{206}
$$

Taking Eq. (203) and Eq. (206) into account, the derivative of $L$ is:

$$
\begin{aligned}
L'(x) &= \frac{I'_\nu(x)}{I_\nu(x)} - \frac{\nu}{x} - 1 \\
&= \frac{I_{\nu+1}(x)}{I_\nu(x)} - 1 \\
&< 0.
\end{aligned}
\tag{207}
$$

Therefore, $\forall \nu > 0, 0 < b < a$, it holds that:

$$
\begin{aligned}
\log(I_\nu(a)) - \nu \log(a) - a &= L(a) \\
&> L(a) \\
&= \log(I_\nu(b)) - \nu \log(b) - b,
\end{aligned}
\tag{208}
$$

then we have:

$$
\log\left(\frac{I_\nu(a)}{I_\nu(b)}\right) > \nu \log\left(\frac{a}{b}\right) + (a - b).
\tag{209}
$$

$\square$

### E.3 DETAILS OF THEOREM 3

In this section, we provide proofs of Theorem 3 that is proposed in Sec. 3.4. We also provide details and proofs of the auxiliary theorems (Theorem S5 and Theorem S6) and the technical lemmas (Lemma 10, Lemma 11, Lemma 12 and Lemma 13) that support the proof Theorem 3. For convenience in reading, let us recall some related notions and definitions.

- $h, N \in \mathbb{N}$.
- $\mathbb{S}^{h-1} = \left\{ z \in \mathbb{R}^h : \|z\| = 1 \right\}$.
- $\mathbb{A} = \left\{ x \in \mathbb{R}^h : n_A \cdot x = 0 \right\}$ where $n_A$ is the normal vector of $\mathbb{A}$.
- $\mathbb{B} = \left\{ y \in \mathbb{R}^h : n_B \cdot y = 0 \right\}$ where $n_A$ is the normal vector of $\mathbb{B}$.
- $\phi = \cos^{-1} \left( \frac{n_x \cdot n_y}{\|n_x\| \cdot \|n_y\|} \right)$ and $0 < \phi_{\min} \leq \phi < \frac{\pi}{2}$.
- $\mathbb{S}_X = \mathbb{S}^{h-1} \cap \mathbb{A} = \left\{ x \in \mathbb{R}^h : \|x\| = 1, n_A \cdot x = 0 \right\} \cong S^{h-2} \in \mathbb{S}^{h-1}$.
- $\mathbb{S}_Y = \mathbb{S}^{h-1} \cap \mathbb{B} = \left\{ y \in \mathbb{R}^h : \|y\| = 1, n_B \cdot y = 0 \right\} \cong S^{h-2} \in \mathbb{S}^{h-1}$.
- $\mathbb{C} = \mathbb{A} \cap \mathbb{B}$.
- $h_X = h_Y = h - 1$.
- $h_C = h - 2$.
- $P_A$: the projection matrix of $\mathbb{A}$.
- $P_B$: the projection matrix of $\mathbb{B}$.
- $P_C$: the projection matrix of $\mathbb{C}$.
- $e_A = \{ z \in \mathbb{S}_X : z \perp \mathbb{C} \}$.
- $e_B = \{ z \in \mathbb{S}_Y : z \perp \mathbb{C} \}$.
- $\mathbb{C}^\perp = \mathrm{span} \{e_A\} \oplus \mathrm{span} \{e_B\}$
- $\mathbb{R}^h = \mathbb{C} \oplus \mathbb{C}^\perp$.
- $X = (x_1, \ldots, x_N) \in (\mathbb{S}_X)^N$.
- $Y = (y_1, \ldots, y_N) \in (\mathbb{S}_Y)^N$.
- $\mu_x = \frac{1}{N} \sum_{i=1}^N x_i$.
- $\mu_y = \frac{1}{N} \sum_{i=1}^N y_i$.
- $c_x = \frac{\mu_x}{\|\mu_x\|}$.
- $c_y = \frac{\mu_y}{\|\mu_y\|}$.

**Definition** (Multimodal Contrastive Loss (MCL Loss)). Let $(X, Y)$ be an $N$-pair configuration, where $X = (x_1, \ldots, x_N) \in (\mathbb{S}^{h-1})^N$ and $Y = (y_1, \ldots, y_N) \in (\mathbb{S}^{h-1})^N$. $\forall \tau > 0$, the multimodal contrastive loss $\mathcal{L}_{\mathrm{MCL}}(\cdot, \cdot) : (\mathbb{S}^{h-1})^N \times (\mathbb{S}^{h-1})^N \to \mathbb{R}$ is defined as:

$$\mathcal{L}_{\mathrm{MCL}} = \frac{1}{N} \sum_{i=1}^N \mathcal{L}_{\mathrm{MCL}}^i, \quad \text{where } \mathcal{L}_{\mathrm{MCL}}^i = \mathcal{L}_{\mathcal{X} \to \mathcal{Y}}(x_i; Y) + \mathcal{L}_{\mathcal{Y} \to \mathcal{X}}(y_i; X).$$

Here, $\mathcal{L}_{\mathcal{X} \to \mathcal{Y}}$ is the $\mathcal{X}$-to-$\mathcal{Y}$ alignment and $\mathcal{L}_{\mathcal{Y} \to \mathcal{X}}$ is the $\mathcal{Y}$-to-$\mathcal{X}$ alignment, which are defined respectively as:

$$\mathcal{L}_{\mathcal{X} \to \mathcal{Y}}(x_i; Y) = -\log \frac{\exp \left( x_i \cdot y_i / \tau \right)}{\sum_{j=1}^N \exp \left( x_i \cdot y_j / \tau \right)}, \quad \mathcal{L}_{\mathcal{Y} \to \mathcal{X}}(y_i; X) = -\log \frac{\exp \left( x_i \cdot y_i / \tau \right)}{\sum_{j=1}^N \exp \left( x_j \cdot y_i / \tau \right)}.$$

**Definition**(Modality Gap) Let $(X, Y)$ be an $N$-pair configuration, where $X = (x_1, \ldots, x_N) \in (\mathbb{S}^{h-1})^N$ and $Y = (y_1, \ldots, y_N) \in (\mathbb{S}^{h-1})^N$. The modality gap between $X$ and $Y$ can be expressed as the angle between the center representations:

$$\Delta_\theta = \cos^{-1}(c_x \cdot c_y).$$

**Definition** (vMF Distribution). $\forall c \in \mathbb{S}^{h-1}$ and $\kappa \geq 0$, the probability density of a random $h$-dimensional unit vector $z \sim \text{vMF}(c, \kappa)$ is given by:

$$f_h(z; c, \kappa) = D_h(\kappa)e^{\kappa c^\top z}, \quad \text{where } D_h(\kappa) = \frac{\kappa^\nu}{(2\pi)^{\nu+1} I_\nu(\kappa)}.$$

Here, $\nu = h/2 - 1$, and $I_\nu(\cdot) : \mathbb{R} \to \mathbb{R}$ is the modified Bessel function of the first kind of order $\nu$, which is defined as:

$$I_\nu(x) = \sum_{k=0}^{\infty} \frac{1}{k!\Gamma(\nu + k + 1)} \left(\frac{x}{2}\right)^{2k+\nu}.$$

**Definition** (Function $\tilde{M}$). $\forall \kappa, \tau > 0$, a function $\tilde{M}_\kappa(\cdot, \cdot) : [-1, 1] \times [0, 1] \to \mathbb{R}_0^+$ is defined as:

$$\tilde{M}_\kappa(w, t) = \sqrt{\kappa^2 + \frac{2\kappa w}{\tau} + \frac{t^2}{\tau^2}}.$$

**Definition** (Function $\tilde{\mathcal{J}}$). $\forall \kappa, \nu, \tau > 0, \tilde{\mathcal{J}}(\cdot, \cdot, \cdot; \kappa, \nu) : [-1, 1] \times [-1, 1] \times [0, 1] \to \mathbb{R}$ is defined as:

$$\tilde{\mathcal{J}}(w_1, w_2, t; \kappa, \nu) = -\frac{w_1}{\tau} + \log\left(\frac{I_\nu\left(\tilde{M}_\kappa(w_2, t)\right)}{\tilde{M}_\kappa(w_2, t)^\nu}\right) - \log\left(\frac{I_\nu(\kappa)}{\kappa^\nu}\right).$$

**Definition** (Function $M$). $\forall \kappa, \tau > 0$, a function $M_\kappa(\cdot) : [-1, 1] \to \mathbb{R}_0^+$ is defined as:

$$\begin{aligned}
M_\kappa(w) &= \sqrt{\kappa^2 + \frac{2\kappa w}{\tau} + \frac{1}{\tau^2}} \\
&= \tilde{M}_\kappa(w, 1).
\end{aligned}$$

**Definition** (Function $\mathcal{J}$). $\forall \kappa, \nu, \tau > 0$, a function $\mathcal{J}(\cdot; \kappa, \nu) : [-1, 1] \to \mathbb{R}$ is defined as:

$$\begin{aligned}
\mathcal{J}(w; \kappa, \nu) &= -\frac{w}{\tau} + \log\left(\frac{I_\nu(M_\kappa(w))}{M_\kappa(w)^\nu}\right) - \log\left(\frac{I_\nu(\kappa)}{\kappa^\nu}\right) \\
&= \tilde{\mathcal{J}}(w, w, 1; \kappa, \nu).
\end{aligned}$$

**Definition** (Function $\tilde{M}$). $\forall \kappa, \tau > 0$, a function $\tilde{M}_\kappa(\cdot, \cdot) : [-1, 1] \times [0, 1] \to \mathbb{R}_0^+$ is defined as:

$$\tilde{M}_\kappa(w, t) = \tilde{M}_\kappa(w, t).$$

**Definition** (Function $\hat{\mathcal{J}}$). $\forall \kappa, \nu, \tau > 0$, a function $\hat{\mathcal{J}}(\cdot, \cdot; \kappa, \nu) : [-1, 1] \times [0, 1] \to \mathbb{R}$ is defined as:

$$\begin{aligned}
\hat{\mathcal{J}}(w, t; \kappa, \nu) &= -\frac{w}{\tau} + \log\left(\frac{I_\nu\left(\tilde{M}_\kappa(w, t)\right)}{\tilde{M}_\kappa(w, t)^\nu}\right) - \log\left(\frac{I_\nu(\kappa)}{\kappa^\nu}\right) \\
&= \tilde{\mathcal{J}}(w, w, t; \kappa, \nu).
\end{aligned}$$

### E.3.1 PROOF OF THEOREM 3

In this subsection, we provide the proof of Theorem 3. For convenience in reading, we first restate Theorem 3 here.

**Theorem 3.** [Restate] Let $(X, Y)$ be an $N$-pair configuration, where $X = (x_1, \ldots, x_N) \in (\mathbb{S}_X \setminus \mathbb{C})^N$ are $iid$ samples from $\mu_x = \text{vMF}(c_x, \kappa_x)$, and $Y = (y_1, \ldots, y_N) \in (\mathbb{S}_Y \setminus \mathbb{C})^N$ are $iid$ samples from $\mu_y = \text{vMF}(c_y, \kappa_y)$. Let $\tilde{\nu} = (h - 1)/2 - 1$. Suppose there exists an index $i = c$ such that $x_c = c_x$, $y_c = c_y$. Denote $\Delta_\theta = \cos^{-1}(c_x \cdot c_y)$ and assume that $c_x, c_y \notin \mathbb{C}$ with $c_x \cdot c_y > 0$. For any fixed $\kappa_x, \kappa_y > 0$, it holds that:

$$
\lim_{N \to \infty} \mathcal{L}_{\text{MCL}}^c - 2\log(N)
$$
$$
= \tilde{\mathcal{J}}(\cos(\Delta_\theta), \cos(\Delta_\theta), \|P_B c_x\|; \kappa_y, \tilde{\nu}) + \tilde{\mathcal{J}}(\cos(\Delta_\theta), \cos(\Delta_\theta), \|P_A c_y\|; \kappa_x, \tilde{\nu})
$$
$$
\geq \tilde{\mathcal{J}}(\cos(\phi_{\min}), \cos(\phi_{\min}), \cos(\phi_{\min}); \kappa_y, \tilde{\nu}) + \tilde{\mathcal{J}}(\cos(\phi_{\min}), \cos(\phi_{\min}), \cos(\phi_{\min}); \kappa_x, \tilde{\nu}),
$$

where equality is attained if and only if there exists a configuration of $(X, Y)$ such that:

(A6) $c_x \perp \mathbb{C}$ and $c_y \perp \mathbb{C}$.

(A7) $\Delta_\theta = \cos^{-1}(c_x \cdot c_y) = \phi_{\min}$.

*Proof.* We first decompose $\lim_{N \to \infty} \mathcal{L}_{\text{MCL}}^c - 2\log(N)$ into two parts:

$$
\lim_{N \to \infty} \mathcal{L}_{\text{MCL}}^c - 2\log(N) = \lim_{N \to \infty} \mathcal{L}_{\mathcal{X} \to \mathcal{Y}}(c_x; Y) - \log(N)
$$
$$
+ \lim_{N \to \infty} \mathcal{L}_{\mathcal{Y} \to \mathcal{X}}(c_y; X) - \log(N).
$$
(210)

Set:

$$
\hat{\mathcal{J}}(w, t; \kappa, \nu) = \tilde{\mathcal{J}}(w, w, t; \kappa, \nu),
$$
$$
\tilde{\nu} = \tilde{\nu},
$$
(211)

According to Theorem S6, the convergent function and its lower bound of $\mathcal{L}_{\mathcal{X} \to \mathcal{Y}}$ are:

$$
\lim_{N \to \infty} \mathcal{L}_{\mathcal{X} \to \mathcal{Y}}(c_x; Y) - \log(N) = \hat{\mathcal{J}}(\cos(\Delta_\theta), \|P_B c_x\|; \kappa_y, \tilde{\nu})
$$
$$
\geq \hat{\mathcal{J}}(\|P_A c_y\|, \|P_A c_y\|, \cos(\phi); \kappa_y, \tilde{\nu}).
$$
(212)

where equality is attained if and only if there exists a configuration of $(X, Y)$ such that:

(i) $c_x \perp \mathbb{C}$.

(ii) $c_x = \frac{P_A c_y}{\|P_A c_y\|}$.

This Theorem also holds for $\mathcal{L}_{\mathcal{Y} \to \mathcal{X}}$:

$$
\lim_{N \to \infty} \mathcal{L}_{\mathcal{Y} \to \mathcal{X}}(c_y; X) - \log(N) = \hat{\mathcal{J}}(\cos(\Delta_\theta), \|P_A c_y\|; \kappa_x, \tilde{\nu})
$$
$$
\geq \hat{\mathcal{J}}(\|P_B c_x\|, \|P_B c_x\|, \cos(\phi); \kappa_x, \tilde{\nu}).
$$
(213)

where equality is attained if and only if there exists a configuration of $(X, Y)$ such that:

(iii) $c_y \perp \mathbb{C}$.

(iv) $c_y = \frac{P_B c_x}{\|P_B c_x\|}$.

According to Lemma 13, for some $\lambda_x, \lambda_y > 0$ such that the projections of $x$ and $y$ are collinear with the other vector:

(1) The orthogonal projection of $x$ on $\mathbb{B}$ is a scalar multiple of $y$:
$$P_B x = \lambda_x y, \quad \lambda_x \neq 0,$$

(2) The orthogonal projection of $y$ on $\mathbb{A}$ is a scalar multiple of $x$:
$$P_A y = \lambda_y x, \quad \lambda_y \neq 0,$$

if and only if the following condition holds:

(v) Either $x \perp \mathbb{C}$ and $y \perp \mathbb{C}$, or $x = \pm y \in \mathbb{C}$.

Since $c_x, c_y \notin \mathbb{C}$, there is only one configuration in (v) that satisfies (ii) + (iv), that is $c_x \perp \mathbb{C}$ and $c_y \perp \mathbb{C}$. In this case, Lemma 13 shows that:

$$\begin{aligned}
\cos\left(\Delta_\theta\right) &= \cos\left(\phi\right) \geq \cos\left(\phi_{\min}\right), \\
\|P_A c_y\| &= \|P_B c_x\| = \cos\left(\phi\right), \\
P_B c_x &= \cos\left(\phi\right) c_y, \\
P_A c_y &= \cos\left(\phi\right) c_x.
\end{aligned} \tag{214}$$

Combining Eq. (212), Eq. (213) and Eq. (214), we have:

$$\begin{aligned}
\lim_{N\to\infty} \mathcal{L}_{\mathrm{MCL}}^c - 2\log(N) &= \hat{\mathcal{J}}\left(\cos\left(\Delta_\theta\right), \|P_B c_x\|; \kappa_y, \tilde{\nu}\right) + \hat{\mathcal{J}}\left(\cos\left(\Delta_\theta\right), \|P_A c_y\|; \kappa_x, \tilde{\nu}\right) \\
&\geq \hat{\mathcal{J}}\left(\cos\left(\phi\right), \cos\left(\phi\right); \kappa_y, \tilde{\nu}\right) + \hat{\mathcal{J}}\left(\cos\left(\phi\right), \cos\left(\phi\right); \kappa_x, \tilde{\nu}\right).
\end{aligned} \tag{215}$$

where equality is attained if and only if there exists a configuration of $(X, Y)$ such that:

(A6) $c_x \perp \mathbb{C}$ and $c_y \perp \mathbb{C}$.

Since Lemma 11 shows that $\hat{\mathcal{J}}\left(\cos\left(\phi\right), \cos\left(\phi\right); \kappa, \tilde{\nu}\right)$ is a strictly decreasing function of $\cos\left(\phi\right)$, we have:

$$\begin{aligned}
\lim_{N\to\infty} \mathcal{L}_{\mathrm{MCL}}^c - 2\log(N) &= \hat{\mathcal{J}}\left(\cos\left(\Delta_\theta\right), \|P_B c_x\|; \kappa_y, \tilde{\nu}\right) + \hat{\mathcal{J}}\left(\cos\left(\Delta_\theta\right), \|P_A c_y\|; \kappa_x, \tilde{\nu}\right) \\
&\geq \hat{\mathcal{J}}\left(\cos\left(\phi\right), \cos\left(\phi\right); \kappa_y, \tilde{\nu}\right) + \hat{\mathcal{J}}\left(\cos\left(\phi\right), \cos\left(\phi\right); \kappa_x, \tilde{\nu}\right) \\
&\geq \hat{\mathcal{J}}\left(\cos\left(\phi_{\min}\right), \cos\left(\phi_{\min}\right); \kappa_y, \tilde{\nu}\right) + \hat{\mathcal{J}}\left(\cos\left(\phi_{\min}\right), \cos\left(\phi_{\min}\right); \kappa_x, \tilde{\nu}\right),
\end{aligned} \tag{216}$$

where equality is attained if and only if there exists a configuration of $(X, Y)$ such that:

(A7) $\Delta_\theta = \cos^{-1}\left(c_x \cdot c_y\right) = \phi_{\min}$.

Replacing $\hat{\mathcal{J}}(w, t; \kappa, \nu)$ with $\tilde{\mathcal{J}}(w, w, t; \kappa, \nu)$, we conclude that:

$$\begin{aligned}
\lim_{N\to\infty} &\mathcal{L}_{\mathrm{MCL}}^c - 2\log(N) \\
&= \tilde{\mathcal{J}}\left(\cos\left(\Delta_\theta\right), \cos\left(\Delta_\theta\right), \|P_B c_x\|; \kappa_y, \tilde{\nu}\right) + \tilde{\mathcal{J}}\left(\cos\left(\Delta_\theta\right), \cos\left(\Delta_\theta\right), \|P_A c_y\|; \kappa_x, \tilde{\nu}\right) \\
&\geq \tilde{\mathcal{J}}\left(\cos\left(\Delta_\theta\right), \cos\left(\Delta_\theta\right), \cos\left(\Delta_\theta\right); \kappa_y, \tilde{\nu}\right) + \tilde{\mathcal{J}}\left(\cos\left(\Delta_\theta\right), \cos\left(\Delta_\theta\right), \cos\left(\Delta_\theta\right); \kappa_x, \tilde{\nu}\right) \\
&\geq \tilde{\mathcal{J}}\left(\cos\left(\phi_{\min}\right), \cos\left(\phi_{\min}\right), \cos\left(\phi_{\min}\right); \kappa_y, \tilde{\nu}\right) + \tilde{\mathcal{J}}\left(\cos\left(\phi_{\min}\right), \cos\left(\phi_{\min}\right), \cos\left(\phi_{\min}\right); \kappa_x, \tilde{\nu}\right),
\end{aligned} \tag{217}$$

where equality is attained if and only if there exists a configuration of $(X, Y)$ such that:

(A6) $c_x \perp \mathbb{C}$ and $c_y \perp \mathbb{C}$.

(A7) $\Delta_\theta = \cos^{-1}(c_x \cdot c_y) = \phi_{\min}$.

$\square$

### E.3.2 Auxiliary Theorems Part 3

In this subsection, we provide details and proofs of the auxiliary theorems (Theorem S5 and Theorem S6) that support the proof of Theorem 3.

**Theorem S5.** *Let $(X, Y)$ be an $N$-pair configuration, where $X = (x_1, \ldots, x_N) \in (\mathbb{S}_X \setminus \mathbb{C})^N$ are iid samples from $\mu_x = \mathrm{vMF}(c_x, \kappa_x)$, and $Y = (y_1, \ldots, y_N) \in (\mathbb{S}_Y \setminus \mathbb{C})^N$ are iid samples from $\mu_y = \mathrm{vMF}(c_y, \kappa_y)$. Let $\tilde{\nu} = (h-1)/2 - 1$ and $\kappa_y > 0$.*

*$\forall x_i \in X$, denote $w_i = x_i \cdot y_i$ and $w_{x_i, c_y} = x_i \cdot c_y$. It holds that:*

$$
\lim_{N \to \infty} \mathcal{L}_{\mathcal{X} \to \mathcal{Y}}(x_i; Y) - \log(N) = \lim_{N \to \infty} -\log \frac{\exp(x_i \cdot y_i / \tau)}{\sum_{j=1}^{N} \exp(x_i \cdot y_j / \tau)} - \log(N)
$$

$$
= -\frac{w_i}{\tau} + \log \left( \frac{I_{\tilde{\nu}} \left( \tilde{M}_{\kappa_y} \left( w_{x_i, c_y}, \|P_B x_i\| \right) \right)}{\tilde{M}_{\kappa_y} \left( w_{x_i, c_y}, \|P_B x_i\| \right)^{\tilde{\nu}}} \right) - \log \left( \frac{I_{\tilde{\nu}}(\kappa_y)}{\kappa_y^{\tilde{\nu}}} \right) \quad (218)
$$

$$
= \tilde{\mathcal{J}} \left( w_i, w_{x_i, c_y}, \|P_B x_i\|; \kappa, \tilde{\nu} \right),
$$

*where $\forall \kappa, \tau > 0$, $\tilde{M}_\kappa(\cdot, \cdot) : [-1, 1] \times [0, 1] \to \mathbb{R}_0^+$ is defined as:*

$$
\tilde{M}_\kappa(w, t) = \sqrt{\kappa^2 + \frac{2\kappa w}{\tau} + \frac{t^2}{\tau^2}}, \quad (219)
$$

*and $I_\nu$ is the modified Bessel function of the first kind of order $\nu$, which is defined as:*

$$
I_\nu(m) = \sum_{k=0}^{\infty} \frac{1}{k! \Gamma(\nu + k + 1)} \left( \frac{m}{2} \right)^{2k+\nu}. \quad (220)
$$

*Suppose there exists an index $i = c$ such that $x_c = c_x$, $y_c = c_y$. Denote $w_c = c_x \cdot c_y$. It holds that:*

$$
\lim_{N \to \infty} \mathcal{L}_{\mathcal{X} \to \mathcal{Y}}(c_x; Y) - \log(N) = -\frac{w_c}{\tau} + \log \left( \frac{I_{\tilde{\nu}} \left( \tilde{M}_{\kappa_y} \left( w_c, \|P_B c_x\| \right) \right)}{\tilde{M}_{\kappa_y} \left( w_c, \|P_B c_x\| \right)^{\tilde{\nu}}} \right) - \log \left( \frac{I_{\tilde{\nu}}(\kappa_y)}{\kappa_y^{\tilde{\nu}}} \right)
$$

$$
= \hat{\mathcal{J}}(w_c, \|P_B c_x\|; \kappa_y, \tilde{\nu}) = \tilde{\mathcal{J}} \left( w_c, w_c, \|P_B x_i\|; \kappa, \tilde{\nu} \right). \quad (221)
$$

*Proof.* **Step 1**: We start the proof by find the convergent function of $\mathcal{L}_{\mathcal{X} \to \mathcal{Y}}(x_i; Y)$ as $N \to \infty$. Same with Eq. (39) of Theorem S1, $\forall x_i \in X$, the $\mathcal{X}$-to-$\mathcal{Y}$ alignment of $x_i$ can be rewritten as:

$$
\mathcal{L}_{\mathcal{X} \to \mathcal{Y}}(x_i; Y) = -\log \frac{\exp(x_i \cdot y_i / \tau)}{\sum_j \exp(x_i \cdot y_j / \tau)}
$$

$$
= -\frac{x_i \cdot y_i}{\tau} + \log \left( N \frac{1}{N} \sum_{j=1}^{N} \exp \left( \frac{x_i \cdot y_j}{\tau} \right) \right) \quad (222)
$$

$$
= -\frac{x_i \cdot y_i}{\tau} + \log \left( \frac{1}{N} \sum_{j=1}^{N} \exp \left( \frac{x_i \cdot y_j}{\tau} \right) \right) + \log(N).
$$

Lemma 2 shows that:

$$\lim_{N \to \infty} \log \left( \frac{1}{N} \sum_{j=1}^{N} \exp \left( \frac{x_i \cdot y_j}{\tau} \right) \right) = \log \left( \mathbb{E}_{y \sim \mu_y} \left[ \exp \left( \frac{x_i \cdot y}{\tau} \right) \right] \right). \tag{223}$$

According to the moment-generating function of the vMF distribution:

$$\mathbb{E}_{y \sim \mu_y} [\exp \left( \frac{x_i \cdot y}{\tau} \right)] = \mathbb{E}_{y \sim \mu_y} \left[ \exp \left( \frac{x_i}{\tau} \cdot y \right) \right] = \frac{I_{\tilde{\nu}} \left( \tilde{\kappa}'_y \right)}{I_{\tilde{\nu}} \left( \kappa_y \right)} \left( \frac{\kappa_y}{\tilde{\kappa}'_y} \right)^{\tilde{\nu}},$$

$$\text{where } \tilde{\kappa}'_y = \| \kappa_y c_y + \frac{P_B x_i}{\tau} \|_2. \tag{224}$$

Then we have:

$$\lim_{N \to \infty} \mathcal{L}_{\mathcal{X} \to \mathcal{Y}}(x_i; Y) - \log(N) = -\frac{x_i \cdot y_i}{\tau} + \log \left( \frac{I_{\tilde{\nu}} \left( \tilde{\kappa}'_y \right)}{\tilde{\kappa}'^{\tilde{\nu}}_y} \right) - \log \left( \frac{I_{\tilde{\nu}} \left( \kappa_y \right)}{\kappa^{\tilde{\nu}}_y} \right). \tag{225}$$

**Step 2**: we will transform $\mathcal{L}_{\mathcal{X} \to \mathcal{Y}}$ from a function of vectors to a function of angles between vectors. Without loss of generality, we assume the coordinate of $c_y$ as

$$c_y = (1, 0, \cdots, 0), \tag{226}$$

the hyperplane $\mathbb{B}$ as:

$$\mathbb{B} = \left\{ x \in \mathbb{R}^h : n_A \cdot x = 0 \right\}, \quad \text{where } n_B = (0, 0, \cdots, 1). \tag{227}$$

Let $\hat{x}_i = P_B x_i$, then we have:

$$\cos \left( \theta_{x_i, c_y} \right) = x_i \cdot c_y = P_B x_i \cdot c_y = \hat{x}_i \cdot c_y. \tag{228}$$

Define:

$$\cos \left( \hat{\theta}_{x_i, c_y} \right) = \frac{\hat{x}_i}{\| \hat{x}_i \|} \cdot c_y = \frac{P_B x_i}{\| P_B x_i \|} \cdot c_y, \tag{229}$$

then we have:

$$\| P_B x_i \| \cos \left( \hat{\theta}_{x_i, c_y} \right) = P_B x_i \cdot c_y = \cos \left( \theta_{x_i, c_y} \right). \tag{230}$$

And $\hat{x}_i$ can be represented as:

$$\hat{x}_i = \| P_B x_i \| \left( \cos \left( \hat{\theta}_{x_i, c_y} \right), u \sin \left( \hat{\theta}_{x_i, c_y} \right) \right)$$
$$= \| P_B x_i \| \left( \cos \left( \hat{\theta}_{x_i, c_y} \right), u_2 \sin \left( \hat{\theta}_{x_i, c_y} \right), u_3 \sin \left( \hat{\theta}_{x_i, c_y} \right), \ldots, u_{h-1} \sin \left( \hat{\theta}_{x_i, c_y} \right), 0 \right), \tag{231}$$

where $u = (0, u_2, u_3, \ldots, u_{h-1}, 0) \cong \mathbb{S}^{h-3} \in \mathbb{S}^{h-1}$ is a unit vector orthogonal to the first and the last axes with:

$$\| u \| = 0 + u_2^2 + u_3^2 + \cdots + u_{h-1}^2 + 0 = 1. \tag{232}$$

According to Eq. (226), Eq. (231) and Eq. (232), $\tilde{\kappa}'_y$ (in Eq. (224)) can re-rewritten as:

$$
\begin{aligned}
\tilde{\kappa}'_y &= \left\| \kappa_y c_y + \frac{x_i}{\tau} \right\|_2 \\
&= \sqrt{\left( \kappa_y + \frac{\|P_B x_i\| \cos\left(\hat{\theta}_{x_i,c_y}\right)}{\tau} \right)^2 + \sum_{i=2}^{h-1} \left( \frac{\|P_B x_i\| \sin\left(\hat{\theta}_{x_i,c_y}\right) u_i}{\tau} \right)^2} \\
&= \sqrt{\left( \kappa_y + \frac{\|P_B x_i\| \cos\left(\hat{\theta}_{x_i,c_y}\right)}{\tau} \right)^2 + \frac{\|P_B x_i\|^2 \sin^2\left(\hat{\theta}_{x_i,c_y}\right)}{\tau^2}} \\
&= \sqrt{\kappa_y^2 + \frac{2\kappa_y \|P_B x_i\| \cos\left(\hat{\theta}_{x_i,c_y}\right)}{\tau} + \frac{\|P_B x_i\|^2}{\tau^2}} \\
&= \sqrt{\kappa_y^2 + \frac{2\kappa_y \cos\left(\theta_{x_i,c_y}\right)}{\tau} + \frac{\|P_B x_i\|^2}{\tau^2}} \\
&= \tilde{M}_{\kappa_y}\left( \cos\left(\theta_{x_i,c_y}\right), \|P_B x_i\| \right).
\end{aligned}
\tag{233}
$$

Consider that $w_i = x_i \cdot y_i$, $w_{x_i,c_y} = \cos\left(\theta_{x_i,c_y}\right) = x_i \cdot c_y$, putting Eq. (225) and Eq. (233) together, we have:

$$
\begin{aligned}
\lim_{N \to \infty} \mathcal{L}_{\mathcal{X} \to \mathcal{Y}}(x_i; Y) - \log(N) &= -\frac{x_i \cdot y_i}{\tau} + \log\left( \frac{I_{\tilde{\nu}}\left(\tilde{\kappa}'_y\right)}{\tilde{\kappa}'^{\tilde{\nu}}_y} \right) - \log\left( \frac{I_{\tilde{\nu}}\left(\kappa_y\right)}{\kappa_y^{\tilde{\nu}}} \right) \\
&= -\frac{w_i}{\tau} + \log\left( \frac{I_{\tilde{\nu}}\left( \tilde{M}_{\kappa_y}\left(w_{x_i,c_y}, \|P_B x_i\|\right) \right)}{\tilde{M}_{\kappa_y}\left(w_{x_i,c_y}, \|P_B x_i\|\right)^{\tilde{\nu}}} \right) - \log\left( \frac{I_{\tilde{\nu}}\left(\kappa_y\right)}{\kappa_y^{\tilde{\nu}}} \right) \\
&= \tilde{\mathcal{J}}\left( w_i, w_{x_i,c_y}, \|P_B x_i\|; \kappa, \tilde{\nu} \right).
\end{aligned}
\tag{234}
$$

When there exists a data pair $i = c$ such that $x_c = c_x$, $y_c = c_y$, $w_i = w_{x_i,c_y} = w_c$, then we have:

$$
\begin{aligned}
\lim_{N \to \infty} \mathcal{L}_{\mathcal{X} \to \mathcal{Y}}(c_x; Y) - \log(N) &= -\frac{w_c}{\tau} + \log\left( \frac{I_{\tilde{\nu}}\left( \tilde{M}_{\kappa_y}\left(w_c, \|P_B c_x\|\right) \right)}{\tilde{M}_{\kappa_y}\left(w_c, \|P_B c_x\|\right)^{\tilde{\nu}}} \right) - \log\left( \frac{I_{\tilde{\nu}}\left(\kappa_y\right)}{\kappa_y^{\tilde{\nu}}} \right) \\
&= \hat{\mathcal{J}}(w_c, \|P_B c_x\|; \kappa_y, \tilde{\nu}) = \tilde{\mathcal{J}}\left( w_c, w_c, \|P_B x_i\|; \kappa, \tilde{\nu} \right).
\end{aligned}
\tag{235}
$$

$\square$

**Theorem S6.** *Let $(X, Y)$ be an $N$-pair configuration, where $X = (x_1, \ldots, x_N) \in (\mathbb{S}_X \setminus \mathbb{C})^N$ are iid samples from $\mu_x = \mathrm{vMF}(c_x, \kappa_x)$, and $Y = (y_1, \ldots, y_N) \in (\mathbb{S}_Y \setminus \mathbb{C})^N$ are iid samples from $\mu_y = \mathrm{vMF}(c_y, \kappa_y)$. Let $\tilde{\nu} = (h-1)/2 - 1$. Suppose there exists an index $i = c$ such that $x_c = c_x$, $y_c = c_y$. Denote $\Delta_\theta = \cos^{-1}(c_x \cdot c_y)$ and assume that $c_x, c_y \notin \mathbb{C}$ with $c_x \cdot c_y > 0$. For any fixed $\kappa_x, \kappa_y > 0$ and $\forall \phi \in [0, \frac{\pi}{2}]$, it holds that:*

$$
\lim_{N \to \infty} \mathcal{L}_{\mathcal{X} \to \mathcal{Y}}(c_x; Y) - \log(N) = \hat{\mathcal{J}}(w_c, \|P_B c_x\|; \kappa_y, \tilde{\nu}) \geq \hat{\mathcal{J}}(\|P_A c_y\|, \cos\left(\phi\right); \kappa_y, \tilde{\nu}),
\tag{236}
$$

*where equality is attained if and only if there exists a configuration of $(X, Y)$ such that:*

*(B4) $c_x \perp \mathbb{C}$.*

(B5) $c_x = \frac{P_A c_y}{\|P_A c_y\|}$.

*Proof.* **Step 1**: Similarly to the proof of Theorem S4 in Sec. E.2.2, we start the proof by finding the convergent function of $\mathcal{L}_{\mathcal{X} \to \mathcal{Y}}(c_x; Y)$ as $N \to \infty$. Denote $w_c = c_x \cdot c_y$. $\forall \kappa_y > 0$, as proven in Theorem S5:

$$\lim_{N \to \infty} \mathcal{L}_{\mathcal{X} \to \mathcal{Y}}(c_x; Y) - \log(N) = \lim_{N \to \infty} - \log \frac{\exp(c_x \cdot c_y / \tau)}{\sum_{j=1}^{N} \exp(c_x \cdot y_j / \tau)} - \log(N)$$

$$= -\frac{w_c}{\tau} + \log\left( \frac{I_{\tilde{\nu}}\left(\tilde{M}_{\kappa_y}\left(w_c, \|P_B c_x\|\right)\right)}{\tilde{M}_{\kappa_y}\left(w_c, \|P_B c_x\|\right)^{\tilde{\nu}}} \right) - \log\left( \frac{I_{\tilde{\nu}}(\kappa_y)}{\kappa_y^{\tilde{\nu}}} \right)$$

$$= \hat{\mathcal{J}}(w_c, \|P_B c_x\|; \kappa_y, \tilde{\nu}),$$

(237)

where $\forall \kappa, \tau > 0$, $\hat{\mathcal{J}}(\cdot, \cdot; \kappa, \tilde{\nu})$ is a function on $[-1, 1] \times [0, 1]$ and $\tilde{M}_\kappa(\cdot, \cdot) : [-1, 1] \times [0, 1] \to \mathbb{R}_0^+$ is defined as:

$$\tilde{M}_\kappa(w, t) = \sqrt{\kappa^2 + \frac{2\kappa w}{\tau} + \frac{t^2}{\tau^2}}.$$

(238)

and $I_\nu$ is the modified Bessel function of the first kind of order $\nu$, which is defined as:

$$I_\nu(m) = \sum_{k=0}^{\infty} \frac{1}{k! \Gamma(\nu + k + 1)} \left( \frac{m}{2} \right)^{2k+\nu}.$$

(239)

**Step 2**: Next, we find the minimal value and the optimal condition of convergent function. $\forall c_x \in \mathbb{S}_X, \phi \in [0, \frac{\pi}{2}]$ it holds that:

$$0 \le \cos(\phi) \le \|P_B c_x\| \le 1.$$

(240)

As shown in Lemma 10, $\forall w_c \in [0, 1]$, $\hat{\mathcal{J}}(w = w_c, t; \kappa_y, \tilde{\nu})$ is a strictly increasing function of $t$ on $(0, 1)$. Therefore, it holds that:

$$\hat{\mathcal{J}}(w_c, \cos(\phi); \kappa_y, \tilde{\nu}) \le \hat{\mathcal{J}}(w_c, \|P_B c_x\|; \kappa_y, \tilde{\nu}) \le \hat{\mathcal{J}}(w_c, 1; \kappa_y, \tilde{\nu}).$$

(241)

where equality in the above chain holds if and only if the following conditions are satisfied:

(i) The first inequality becomes equality: $c_x \perp \mathbb{C}$.

(ii) The second inequality becomes equality: $c_x \in \mathbb{C}$.

According to Lemma 5 (set $s = \cos(\phi)$), $\hat{\mathcal{J}}(w_c, \cos(\phi); \kappa_y, \tilde{\nu})$ is a strictly decreasing function on $w_c$ when $w_c \ge 0$. Also, Lemma 12 shows that:

$$-\|P_A c_y\| \le w_c \le \|P_A c_y\|,$$

(242)

where

$$0 \le \cos(\phi) < \|P_A c_y\| < 1.$$

(243)

Therefore, when $0 \le w_c \le \|P_A c_y\|$, it holds that:

$$\hat{\mathcal{J}}(w_c, \cos(\phi); \kappa_y, \tilde{\nu}) \geq \hat{\mathcal{J}}(\|P_A c_y\|, \cos(\phi); \kappa_y, \tilde{\nu}), \tag{244}$$

where equality is attained if and only if there exists a configuration of $(X, Y)$ such that:

(iii) $c_x = \frac{P_A c_y}{\|P_A c_y\|}$.

Combining Eq. (237), Eq. (241) and Eq. (244), we conclude:

$$\lim_{N \to \infty} \mathcal{L}_{\mathcal{X} \to \mathcal{Y}}(c_x; Y) - \log(N) = \hat{\mathcal{J}}(w_c, \|P_B c_x\|; \kappa_y, \tilde{\nu}) \geq \hat{\mathcal{J}}(\|P_A c_y\|, \cos(\phi); \kappa_y, \tilde{\nu}), \tag{245}$$

and equality is attained if and only if there exists a configuration of $(X, Y)$ such that:

(B4) $c_x \perp \mathbb{C}$.

(B5) $c_x = \frac{P_A c_y}{\|P_A c_y\|}$.

$\square$

### E.3.3 TECHNICAL LEMMAS PART 3

In this subsection, we provide details and proofs of technical lemmas (Lemma 10, Lemma 11, Lemma 12 and Lemma 13) that support the proof of Theorem 3, Theorem S5 and Theorem S6.

**Lemma 10.** $\forall \kappa, \nu, \tau > 0$ and $w_c \in [0, 1]$, a function $\hat{\mathcal{J}}_{t=s}(\cdot; \kappa, \nu) : (0, 1] \to \mathbb{R}$ is defined as:

$$\hat{\mathcal{J}}_{w=w_s}(t; \kappa, \nu) = -\frac{w_s}{\tau} + \log\left(\frac{I_\nu\left(\tilde{M}_{w=w_s}(t)\right)}{\tilde{M}_\kappa(t)^\nu}\right) - \log\left(\frac{I_\nu(\kappa)}{\kappa^\nu}\right) \tag{246}$$

$$= \hat{\mathcal{J}}(w = w_s, t; \kappa, \nu) = \tilde{J}(w = w_s, w = w_s, t; \kappa, \nu),$$

where $\tilde{M}_\kappa(\cdot) : (0, 1] \to \mathbb{R}^+$ is defined as:

$$\tilde{M}_{w=w_s}(t) = \sqrt{\kappa^2 + \frac{2\kappa w_s}{\tau} + \frac{t^2}{\tau^2}} = \tilde{M}_\kappa(w = w_s, t), \tag{247}$$

and $I_\nu$ is the modified Bessel function of the first kind of order $\nu$, which is defined as:

$$I_\nu(m) = \sum_{k=0}^\infty \frac{1}{k! \Gamma(\nu + k + 1)} \left(\frac{m}{2}\right)^{2k+\nu}. \tag{248}$$

It holds that, for any fixed $w_s$, $\hat{\mathcal{J}}_{w=w_s}(\cdot)$ is a strictly increasing function on $(0, 1]$.

*Proof.* Let us first decompose the function $\mathcal{J}$. Denote a constant and a function $C_1$ and $G_2(t)$ as:

$$C_1 = -\frac{w_s}{\tau},$$
$$G_3(m) = \log(I_\nu(m)) - \nu \log(m),$$
$$G_2(t) = G_3\left(\tilde{M}_{w=w_s}(t)\right) \tag{249}$$
$$= \log\left(I_\nu\left(\tilde{M}_{w=w_s}(t)\right)\right) - \nu \log\left(\tilde{M}_{w=w_s}(t)\right).$$

Denote the function $G(t)$ and the constant $C$ as:

$$G(t) = C_1 + G_2(t),$$
$$C = -\log\left(\frac{I_\nu(\kappa)}{\kappa^\nu}\right).$$

(250)

Then the function $\hat{\mathcal{J}}_{w=w_s}$ can be written as:

$$\hat{\mathcal{J}}_{w=w_s}(t;\kappa,\nu) = -\frac{w_s}{\tau} + \log\left(\frac{I_\nu\left(\tilde{M}_{w=w_s}(t)\right)}{\tilde{M}_{w=w_s}(t)^\nu}\right) - \log\left(\frac{I_\nu(\kappa)}{\kappa^\nu}\right)$$

$$= G(t) + C.$$

(251)

Now, we investigate derivatives of $\hat{\mathcal{J}}_{w=w_s}$.

According to Lemma 7, the first derivative of $G_3(m)$ is:

$$G_3'(m) = \frac{I_{\nu+1}(m)}{I_\nu(m)} \in (0,1).$$

(252)

The derivative of $\tilde{M}_{w=w_s}$ is:

$$\tilde{M}_{w=w_s}'(t) = \frac{d}{dt}\left(\kappa^2 + \frac{2\kappa w_s}{\tau} + \frac{t^2}{\tau^2}\right)^{1/2}$$

$$= \frac{1}{2}\left(\kappa^2 + \frac{2\kappa w_s}{\tau} + \frac{t^2}{\tau^2}\right)^{-1/2} \cdot 2\frac{t}{\tau^2}$$

$$= \frac{t}{\tau^2}\frac{1}{\tilde{M}_{w=w_s}(t)}$$

$$> 0.$$

(253)

Then, the first derivative of $G_2$ is:

$$G_2'(t) = G_3'\left(\tilde{M}_{w=w_s}(t)\right)\tilde{M}_{w=w_s}'(t)$$

$$= \frac{I_{\nu+1}\left(\tilde{M}_{w=w_s}(t)\right)}{I_\nu\left(\tilde{M}_{w=w_s}(t)\right)}\tilde{M}_{w=w_s}'(t)$$

$$= \frac{t}{\tau^2}\frac{1}{\tilde{M}_{w=w_s}(t)}\frac{I_{\nu+1}\left(\tilde{M}_{w=w_s}(t)\right)}{I_\nu\left(\tilde{M}_{w=w_s}(t)\right)}$$

$$> 0.$$

(254)

Therefore, we have:

$$\hat{\mathcal{J}}_{w=w_s}'(t;\kappa,\nu) = G'(t) = G_2'(t)$$

$$= \frac{t}{\tau^2}\frac{1}{\tilde{M}_{w=w_s}(t)}\frac{I_{\nu+1}\left(\tilde{M}_{w=w_s}(t)\right)}{I_\nu\left(\tilde{M}_{w=w_s}(t)\right)}$$

$$> 0.$$

(255)

So we can conclude that, for any fixed $w_s$, $\hat{\mathcal{J}}_{w=w_s}(\cdot)$ is a strictly increasing function on $(0,1]$.

$\square$

**Lemma 11.** $\forall \kappa, \nu, \tau > 0$, a function $\hat{\mathcal{J}}(\cdot; \kappa, \nu) : [-1, 1] \to \mathbb{R}$ is defined as:

$$\hat{\mathcal{J}}_{t=w}(w; \kappa, \nu) = -\frac{w}{\tau} + \log\left(\frac{I_\nu\left(\tilde{M}_{t=w}(w)\right)}{\tilde{M}_{t=w}(w)^\nu}\right) - \log\left(\frac{I_\nu(\kappa)}{\kappa^\nu}\right) \tag{256}$$

$$= \hat{\mathcal{J}}(w, t = w; \kappa, \nu) = \tilde{J}(w, w, t = w; \kappa, \nu),$$

where $\tilde{M}_{t=w}(\cdot) : [-1, 1] \to \mathbb{R}^+$ is defined as:

$$\tilde{M}_{t=w}(w) = \sqrt{\kappa^2 + \frac{2\kappa w}{\tau} + \frac{w^2}{\tau^2}} = \left|\kappa + \frac{w}{\tau}\right| = \tilde{M}_\kappa(w, t = w), \tag{257}$$

and $I_\nu$ is the modified Bessel function of the first kind of order $\nu$, which is defined as:

$$I_\nu(m) = \sum_{k=0}^{\infty} \frac{1}{k!\Gamma(\nu + k + 1)} \left(\frac{m}{2}\right)^{2k+\nu}. \tag{258}$$

It holds that $\hat{\mathcal{J}}_{t=w}(\cdot)$ is a strictly decreasing function when $w \in [0, 1]$.

*Proof.* Let us first decompose the function $\hat{\mathcal{J}}_{t=w}$. Denote the functions $G_1(w)$ and $G_2(w)$ as:

$$\begin{aligned}
G_1(w) &= -\frac{w}{\tau}, \\
G_3(m) &= \log(I_\nu(m)) - \nu \log(m), \\
G_2(w) &= G_3\left(\tilde{M}_{t=w}(w)\right) \\
&= \log\left(I_\nu\left(\tilde{M}_{t=w}(w)\right)\right) - \nu \log\left(\tilde{M}_{t=w}(w)\right).
\end{aligned} \tag{259}$$

Denote the function $G(w)$ and the constant $C$ as:

$$\begin{aligned}
G(w) &= G_1(w) + G_2(w), \\
C &= -\log\left(\frac{I_\nu(\kappa)}{\kappa^\nu}\right).
\end{aligned} \tag{260}$$

Then the function $\hat{\mathcal{J}}_{t=w}$ can be written as:

$$\hat{\mathcal{J}}_{t=w}(w; \kappa, \nu) = -\frac{w}{\tau} + \log\left(\frac{I_\nu\left(\tilde{M}_{t=w}(w)\right)}{\tilde{M}_{t=w}(w)^\nu}\right) - \log\left(\frac{I_\nu(\kappa)}{\kappa^\nu}\right) \tag{261}$$

$$= G(w) + C.$$

Now, we investigate derivatives of $\hat{\mathcal{J}}_{t=w}$.

The first derivative of $G_1$ is:

$$G_1'(w) = -\frac{1}{\tau} < 0. \tag{262}$$

According to Lemma 7, the first derivative of $G_3(m)$ is:

$$G_3'(m) = \frac{I_{\nu+1}(m)}{I_\nu(m)} \in (0,1).\tag{263}$$

When $w \in [0,1]$, the derivative of $\tilde{M}_{t=w}$ is:

$$\tilde{M}_{t=w}'(w) = \frac{1}{\tau}.\tag{264}$$

Then, the first derivative of $G_2$ is:

$$
\begin{aligned}
G_2'(w) &= G_3'\left(\tilde{M}_{t=w}(w)\right)\tilde{M}_{t=w}'(w)\\
&= \frac{I_{\nu+1}\left(\tilde{M}_{t=w}(w)\right)}{I_\nu\left(\tilde{M}_{t=w}(w)\right)}\tilde{M}_{t=w}'(w)\\
&= \frac{1}{\tau}\frac{I_{\nu+1}\left(\tilde{M}_{t=w}(w)\right)}{I_\nu\left(\tilde{M}_{t=w}(w)\right)}.
\end{aligned}\tag{265}
$$

Combining Eq. (262), Eq. (263) and Eq. (265), we have:

$$
\begin{aligned}
\hat{\mathcal{J}}_{t=w}'(w;\kappa,\nu) &= G'(w)\\
&= -\frac{1}{\tau} + \frac{1}{\tau}\frac{I_{\nu+1}\left(\tilde{M}_{t=w}(w)\right)}{I_\nu\left(\tilde{M}_{t=w}(w)\right)} = \frac{1}{\tau}\left(-1 + \frac{I_{\nu+1}\left(\tilde{M}_{t=w}(w)\right)}{I_\nu\left(\tilde{M}_{t=w}(w)\right)}\right)\\
&< 0.
\end{aligned}\tag{266}
$$

So we can conclude that $\hat{\mathcal{J}}_{t=w}(\cdot)$ is a strictly decreasing function on $[0,1]$.

$\square$

**Lemma 12.** *Let $h \geq 3$ and $\mathbb{A}, \mathbb{B} \in \mathbb{R}^h$ be two distinct $(h-1)$-dimensional linear subspaces, with $n_A, n_B$ being normal vectors and $P_A, P_B$ being the orthogonal projectors on $\mathbb{A}$ and $\mathbb{B}$, respectively. Denote $\phi = \cos^{-1}\left(\frac{n_A \cdot n_B}{\|n_A\| \cdot \|n_B\|}\right) \in \left(0, \frac{\pi}{2}\right)$ as the angle between $\mathbb{A}$ and $\mathbb{B}$. Let $\mathbb{C} = \mathbb{A} \cap \mathbb{B}$ be an $(h-2)$-dimensional linear subspaces. For each fixed $x \in \mathbb{S}_X = \mathbb{A} \cap \mathbb{S}^{h-1}, \forall y \in \mathbb{S}_Y = \mathbb{B} \cap \mathbb{S}^{h-1}$, set $w = x \cdot y$, it holds that:*

$$-\|P_B \cdot x\| \leq w \leq \|P_B \cdot x\|,\tag{267}$$

*and equalities (extreme values) are attained if and only if the following conditions hold:*

*(C1)* $w = \|P_B \cdot x\| \Leftrightarrow y = \frac{P_B \cdot x}{\|P_B \cdot x\|}$.

*(C2)* $w = -\|P_B \cdot x\| \Leftrightarrow y = -\frac{P_B \cdot x}{\|P_B \cdot x\|}$.

*Proof.* **Step 1:** First, let us decompose the embedding space. Define two vectors $e_A$ and $e_B$ such that:

$$
\begin{aligned}
e_A &\in \mathbb{S}_X, \ \text{ and } \ e_A \perp \mathbb{C},\\
e_B &\in \mathbb{S}_Y, \ \text{ and } \ e_B \perp \mathbb{C}.
\end{aligned}\tag{268}
$$

Let $\mathbb{C}^\perp$ be the 2-dimensional orthogonal complement of $C$, and $\mathbb{C}^\perp$ satisfies:

$$\mathbb{C}^{\perp} = \mathrm{span}\left\{e_A\right\} \oplus \mathrm{span}\left\{e_B\right\},$$
$$\mathbb{R}^h = \mathbb{C} \oplus \mathbb{C}^{\perp}. \tag{269}$$

Since $n_A, n_B \in \mathbb{C}^{\perp}$, $n_A \perp e_A$ and $n_B \perp e_B$, we have:

$$\langle e_A, e_B \rangle = \pm \langle n_A, n_B \rangle, \tag{270}$$

and we choose a pair of $e_A$ and $e_B$ such that:

$$\langle e_A, e_B \rangle = \langle n_A, n_B \rangle = \cos\left(\phi\right) \in (0, 1). \tag{271}$$

Therefore, $\forall x \in \mathbb{S}_X = \mathbb{A} \cap \mathbb{S}^{h-1}$ and $\forall y \in \mathbb{S}_Y = \mathbb{B} \cap \mathbb{S}^{h-1}$, $\exists u_A, u_B \in \mathbb{C} \cap \mathbb{S}^{h-1}$, such that $\cos\left(\theta_A\right) = x \cdot e_A$ and $\cos\left(\theta_B\right) = y \cdot e_B$. And then $x$ and $y$ can be represented as:

$$x = \cos(\theta_A)e_A + \sin(\theta_A)u_A,$$
$$y = \cos(\theta_B)e_B + \sin(\theta_B)u_B. \tag{272}$$

Using orthogonality, we have:

$$P_B \cdot e_A = \langle e_A, e_B \rangle e_B = \cos\left(\phi\right) e_B,$$
$$P_B \cdot u_A = u_A, \tag{273}$$

and

$$P_A \cdot e_B = \langle e_A, e_B \rangle e_A = \cos\left(\phi\right) e_A,$$
$$P_A \cdot u_B = u_B. \tag{274}$$

Then the projections of $(x_i, y_i)$ are:

$$P_B \cdot x = \cos(\theta_A) \cos\left(\phi\right) e_B + \sin(\theta_A)u_A,$$
$$P_A \cdot y = \cos(\theta_B) \cos\left(\phi\right) e_A + \sin(\theta_B)u_B. \tag{275}$$

**Step 2:** Next, we can investigate the range of $w$.

$$\begin{aligned}
w &= x \cdot y \\
&= \cos(\theta_A) \cos(\theta_B)e_A e_B + \sin(\theta_A) \sin(\theta_B)u_A u_B \\
&= \cos(\theta_A) \cos(\theta_B) \cos\left(\phi\right) + \sin(\theta_A) \sin(\theta_B)u_A u_B.
\end{aligned} \tag{276}$$

Since $u_A, u_B \in \mathbb{C}$ and $\|u_A\| = \|u_B\| = 1$, then $\|u_A \cdot u_B\| \leq 1$. Denote $f\left(\cdot\right)_{\pm}$ as:

$$f_{\pm}(\theta_B) = \cos(\theta_A) \cos(\theta_B) \cos\left(\phi\right) \pm \sin(\theta_A) \sin(\theta_B), \tag{277}$$

then :

$$f_-(\theta_B) \leq w \leq f_+(\theta_B). \tag{278}$$

Now, let us check the extreme values of $f_{\pm}\left(w\right)$. First, we find the derivative of $f_{\pm}\left(w\right)$:

$$f'_{\pm}(\theta_B) = -\cos(\theta_A) \sin(\theta_B) \cos\left(\phi\right) \pm \sin(\theta_A) \cos(\theta_B), \tag{279}$$

then:

$$f'_{\pm}(\theta_B) = 0 \quad \Rightarrow \quad \tan(\theta_B) = \pm \frac{\sin(\theta_A)}{\cos(\theta_A)\cos(\phi)}, \tag{280}$$

and

$$w \geq f_- \left( \arctan\left( -\frac{\sin(\theta_A)}{\cos(\theta_A)\cos(\phi)} \right) \right) = -\sqrt{\sin^2(\theta_A) + \cos^2(\theta_A)\cos^2(\phi)},$$
$$w \leq f_+ \left( \arctan\left( \frac{\sin(\theta_A)}{\cos(\theta_A)\cos(\phi)} \right) \right) = \sqrt{\sin^2(\theta_A) + \cos^2(\theta_A)\cos^2(\phi)}. \tag{281}$$

Denote:

$$r(x) = \sqrt{\sin^2(\theta_A) + \cos^2(\theta_A)\cos^2(\phi)} \in (0,1). \tag{282}$$

and therefore:

$$|w| \leq r(x) < 1. \tag{283}$$

**Step 3:** Last, we find the optimal condition of $w$. When $\theta_B = \arctan\left( \frac{\sin(\theta_A)}{\cos(\theta_A)\cos(\phi)} \right)$ and $u_A = u_B$, $w$ reaches its maximum. At this time:

$$\cos(\theta_B) = \frac{\cos(\theta_A)\cos(\phi)}{r},$$
$$\sin(\theta_B) = \frac{\sin(\theta_A)}{r}. \tag{284}$$

Plugging Eq. (284) into Eq. (275), we get:

$$\begin{aligned} P_B \cdot x &= \cos(\theta_A)\cos(\phi)\, e_B + \sin(\theta_A) u_A \\ &= r\cos(\theta_B)\cos(\phi)\, e_B + r\sin(\theta_B) u_B \\ &= ry, \end{aligned} \tag{285}$$

and

$$\|P_B \cdot x\| = \|ry\| = r. \tag{286}$$

Therefore, $w$ reaches its maximum if and only if the following condition holds:

(C1) $y = \frac{P_B \cdot x}{\|P_B \cdot x\|}$.

When $\theta_B = \arctan\left( -\frac{\sin(\theta_A)}{\cos(\theta_A)\cos(\phi)} \right)$ and $u_A = -u_B$, $w$ reaches its minimum. At this time:

$$\cos(\theta_B) = -\frac{\cos(\theta_A)\cos(\phi)}{r},$$
$$\sin(\theta_B) = \frac{\sin(\theta_A)}{r}. \tag{287}$$

Plugging Eq. (287) into Eq. (275), we get:

$$\begin{aligned} P_B \cdot x &= \cos(\theta_A)\cos(\phi)\, e_B + \sin(\theta_A) u_A \\ &= -r\cos(\theta_B)\cos(\phi)\, e_B - r\sin(\theta_B) u_B \\ &= -ry. \end{aligned} \tag{288}$$

and

$$\|P_B \cdot x\| = \|-ry\| = r. \tag{289}$$

Therefore, $w$ reaches its minimum if and only if the following condition holds:

(C2) $y = -\frac{P_B \cdot x}{\|P_B \cdot x\|}$.

$\square$

**Lemma 13.** *Let $h \geq 3$ and $\mathbb{A}, \mathbb{B} \in \mathbb{R}^h$ be two distinct $(h-1)$-dimensional linear subspaces, with $n_A, n_B$ being normal vectors and $P_A, P_B$ being the orthogonal projectors on $\mathbb{A}$ and $\mathbb{B}$, respectively. Denote $\phi = \cos^{-1}\left(\frac{n_A \cdot n_B}{\|n_A\| \cdot \|n_B\|}\right) \in \left(0, \frac{\pi}{2}\right)$ as the angle between $\mathbb{A}$ and $\mathbb{B}$. Let $\mathbb{C} = \mathbb{A} \cap \mathbb{B}$ be an $(h-2)$-dimensional linear subspaces. For $x \in \mathbb{S}_X = \mathbb{A} \cap \mathbb{S}^{h-1}, y \in \mathbb{S}_Y = \mathbb{B} \cap \mathbb{S}^{h-1}$, the projections of $x$ and $y$ are collinear with the other vector:*

*(i) The orthogonal projection of $x$ on $\mathbb{B}$ is a scalar multiple of $y$:*
$$P_B x = \lambda_x y, \quad \lambda_x \neq 0,$$

*(ii) The orthogonal projection of $y$ on $\mathbb{A}$ is a scalar multiple of $x$:*
$$P_A y = \lambda_y x, \quad \lambda_y \neq 0,$$

*if and only if the following conditions holds:*

*(C3) Either $x \perp \mathbb{C}$ and $y \perp \mathbb{C}$, or $x = \pm y \in \mathbb{C}$.*

*Moreover, in the first case ($x \perp \mathbb{C}, y \perp \mathbb{C}$), it holds that:*

$$\langle x, y \rangle = \cos(\phi), \qquad P_B x = (\cos(\phi)) y, \qquad P_A y = (\cos(\phi)) x,$$

*while in the second case ($x = \pm y \in \mathbb{C}$), it holds that:*
$$P_B x = x = (\pm 1) y, \qquad P_A y = y = (\pm 1) x.$$

*Proof.* **Step 1:** First, we need to decompose the embedding space. This step is the same with **Step 1** of Sec. E.3.3. For convenience in reading, we repeat this step here.

Define two vectors $e_A$ and $e_B$ such that:

$$\begin{aligned} e_A \in \mathbb{S}_X, \quad &\text{and} \quad e_A \perp \mathbb{C}, \\ e_B \in \mathbb{S}_Y, \quad &\text{and} \quad e_B \perp \mathbb{C}. \end{aligned} \tag{290}$$

Let $\mathbb{C}^\perp$ be the 2-dimensional orthogonal complement of $C$, and $\mathbb{C}^\perp$ satisfies:

$$\begin{aligned} \mathbb{C}^\perp &= \text{span}\{e_A\} \oplus \text{span}\{e_B\}, \\ \mathbb{R}^h &= \mathbb{C} \oplus \mathbb{C}^\perp. \end{aligned} \tag{291}$$

Since $n_A, n_B \in \mathbb{C}^\perp$, $n_A \perp e_A$ and $n_B \perp e_B$, we have:

$$\langle e_A, e_B \rangle = \pm \langle n_A, n_B \rangle, \tag{292}$$

and we choose a pair of $e_A$ and $e_B$ such that:

$$\langle e_A, e_B \rangle = \langle n_A, n_B \rangle = \cos(\phi) \in (0, 1). \tag{293}$$

Therefore, $\forall x \in \mathbb{S}_X = \mathbb{A} \cap \mathbb{S}^{h-1}$ and $\forall y \in \mathbb{S}_Y = \mathbb{B} \cap \mathbb{S}^{h-1}$, $\exists u_A, u_B \in C \cap \mathbb{S}^{h-1}$, such that $\cos(\theta_A) = x \cdot e_A$ and $\cos(\theta_B) = y \cdot e_B$. And then $x$ and $y$ can be represented as:

$$
\begin{aligned}
x &= \cos(\theta_A) e_A + \sin(\theta_A) u_A, \\
y &= \cos(\theta_B) e_B + \sin(\theta_B) u_B.
\end{aligned}
\tag{294}
$$

Using orthogonality, we have:

$$
\begin{aligned}
P_B \cdot e_A &= \langle e_A, e_B \rangle e_B = \cos(\phi) e_B, \\
P_B \cdot u_A &= u_A,
\end{aligned}
\tag{295}
$$

and

$$
\begin{aligned}
P_A \cdot e_B &= \langle e_A, e_B \rangle e_A = \cos(\phi) e_A, \\
P_A \cdot u_B &= u_B.
\end{aligned}
\tag{296}
$$

Then the projections of $(x_i, y_i)$ are:

$$
\begin{aligned}
P_B \cdot x &= \cos(\theta_A) \cos(\phi) e_B + \sin(\theta_A) u_A, \\
P_A \cdot y &= \cos(\theta_B) \cos(\phi) e_A + \sin(\theta_B) u_B.
\end{aligned}
\tag{297}
$$

**Step 2:** $\Rightarrow$ Next, we prove the sufficiency. If conditions (i) and (ii) hold, then:

$$
\begin{aligned}
\cos(\theta_A) \cos(\phi) e_B + \sin(\theta_A) u_A &= \lambda_x \cos(\theta_B) e_B + \lambda_x \sin(\theta_B) u_B, \\
\cos(\theta_B) \cos(\phi) e_A + \sin(\theta_B) u_B &= \lambda_y \cos(\theta_A) e_A + \lambda_y \sin(\theta_A) u_A.
\end{aligned}
\tag{298}
$$

Decompose both equations into $\mathbb{C}$ and $\mathbb{C}^\perp$. In $\mathbb{C}$, we get:

$$
\begin{aligned}
\sin(\theta_A) u_A &= \lambda_x \sin(\theta_B) u_B, \\
\sin(\theta_B) u_B &= \lambda_y \sin(\theta_A) u_A.
\end{aligned}
\tag{299}
$$

and in $\mathbb{C}^\perp$ we get:

$$
\begin{aligned}
\cos(\theta_A) \cos(\phi) e_B &= \lambda_x \cos(\theta_B) e_B, \\
\cos(\theta_B) \cos(\phi) e_A &= \lambda_y \cos(\theta_A) e_A.
\end{aligned}
\tag{300}
$$

Then it can be concluded from Eq. (299) that:

$$
\begin{aligned}
\sin(\theta_A) u_A &= \lambda_x \lambda_y \sin(\theta_A) u_A, \\
\sin(\theta_B) u_B &= \lambda_x \lambda_y \sin(\theta_B) u_B.
\end{aligned}
\tag{301}
$$

Eq. (301) leads to two scenarios:

(S1) $\lambda_x \lambda_y = 1$.

(S2) $\sin(\theta_A) = \sin(\theta_B) = 0$.

When (S1) holds, multiply two equations in Eq. (300) and we get:

$$
\cos(\theta_A) \cos(\theta_B) \cos^2(\phi) = \cos(\theta_A) \cos(\theta_B).
\tag{302}
$$

And since:

$$
0 < \cos^2(\phi) < 1,
\tag{303}
$$

we can conclude that:

$$\cos(\theta_A) = \cos(\theta_B) = 0,$$
$$\sin(\theta_A) = \sin(\theta_B) = \pm 1. \tag{304}$$

Plugging Eq. (304) into Eq. (299), we get:

$$u_A = \lambda_x u_B,$$
$$u_B = \lambda_y u_A. \tag{305}$$

Since $\|u_A\| = \|u_B\| = 1$, Eq. (305) $\Rightarrow \lambda_x = \lambda_y = \pm 1 \Rightarrow u_A = \pm u_B$. And according to Eq. (294) and we have:

$$x = \pm y \in \mathbb{C}. \tag{306}$$

We conclude that (S1) $\Rightarrow x = \pm y \in \mathbb{C}$.

When (S2) holds, we have:

$$\sin(\theta_A) = \sin(\theta_B) = 0,$$
$$\cos(\theta_A) = \cos(\theta_B) = \pm 1. \tag{307}$$

Plugging Eq. (307) into Eq. (294), we have:

$$x = \pm e_A \perp \mathbb{C},$$
$$y = \pm e_B \perp \mathbb{C}. \tag{308}$$

We conclude that (S2) $\Rightarrow x \perp \mathbb{C}$ and $y \perp \mathbb{C}$.

So the sufficiency is confirmed.

**Step 3:** $\Leftarrow$ Last, we prove the necessity. If $x = \pm y \in \mathbb{C}$, then

$$\cos(\theta_A) = \cos(\theta_B) = 0,$$
$$\sin(\theta_A) = \sin(\theta_B) = \pm 1. \tag{309}$$

and

$$x = u_A,$$
$$y = u_B, \tag{310}$$

According to Eq. (297) and Eq. (310), we have:

$$P_B \cdot x = u_A = x = \pm y,$$
$$P_A \cdot y = u_B = y = \pm x. \tag{311}$$

Let $\lambda_x = \lambda_y = \pm 1$, conditions (i) and (ii) hold.

If $x \perp \mathbb{C}$ and $y \perp \mathbb{C}$, then:

$$\sin(\theta_A) = \sin(\theta_B) = 0,$$
$$\cos(\theta_A) = \cos(\theta_B) = \pm 1. \tag{312}$$

and

$$x = \pm e_A,$$
$$y = \pm e_B. \tag{313}$$

According to Eq. (297) and Eq. (313), we have:

$$P_B \cdot x = \pm \cos(\phi) e_B = \pm \cos(\phi) y,$$
$$P_A \cdot y = \pm \cos(\phi) e_A = \pm \cos(\phi) x. \tag{314}$$

Let $\lambda_x = \lambda_y = \pm \cos(\phi)$, conditions (i) and (ii) hold.

Therefore, the necessity is confirmed.

$\square$

### E.4 DETAILS OF THEOREM 4

In this section, we provide proofs of Theorem 4 that is proposed in Sec. 4.2. We also provide details and proofs of the auxiliary theorems (Theorem S7 and Theorem S8) and the technical lemmas (Lemma 14 and Lemma 15) that support the proof Theorem 4. For convenience in reading, let us recall some related notions and definitions.

- $h, N \in \mathbb{N}$.
- $\mathbb{S}^{h-1} = \{z \in \mathbb{R}^h : \|z\| = 1\}$.
- $\mathbb{A} = \{x \in \mathbb{R}^h : n_A \cdot x = 0\}$ where $n_A$ is the normal vector of $\mathbb{A}$.
- $\mathbb{B} = \{y \in \mathbb{R}^h : n_B \cdot y = 0\}$ where $n_A$ is the normal vector of $\mathbb{B}$.
- $\phi = \cos^{-1}\left(\frac{n_x \cdot n_y}{\|n_x\| \cdot \|n_y\|}\right)$ and $0 < \phi_{\min} \leq \phi < \frac{\pi}{2}$.
- $\mathbb{S}_X = \mathbb{S}^{h-1} \cap \mathbb{A} = \{x \in \mathbb{R}^h : \|x\| = 1, n_A \cdot x = 0\} \cong S^{h-2} \in \mathbb{S}^{h-1}$.
- $\mathbb{S}_Y = \mathbb{S}^{h-1} \cap \mathbb{B} = \{y \in \mathbb{R}^h : \|y\| = 1, n_B \cdot y = 0\} \cong S^{h-2} \in \mathbb{S}^{h-1}$.
- $\mathbb{C} = \mathbb{A} \cap \mathbb{B}$.
- $h_X = h_Y = h - 1$.
- $h_C = h - 2$.
- $P_A$: the projection matrix of $\mathbb{A}$.
- $P_B$: the projection matrix of $\mathbb{B}$.
- $P_C$: the projection matrix of $\mathbb{C}$.
- $e_A = \{z \in \mathbb{S}_X : z \perp \mathbb{C}\}$.
- $e_B = \{z \in \mathbb{S}_Y : z \perp \mathbb{C}\}$.
- $\mathbb{C}^\perp = \text{span}\{e_A\} \oplus \text{span}\{e_B\}$
- $\mathbb{R}^h = \mathbb{C} \oplus \mathbb{C}^\perp$.
- $X = (x_1, \ldots, x_N) \in (\mathbb{S}_X)^N$.
- $Y = (y_1, \ldots, y_N) \in (\mathbb{S}_Y)^N$.
- $\mu_x = \frac{1}{N}\sum_{i=1}^N x_i$.
- $\mu_y = \frac{1}{N}\sum_{i=1}^N y_i$.
- $c_x = \frac{\mu_x}{\|\mu_x\|}$.
- $c_y = \frac{\mu_y}{\|\mu_y\|}$.

**Definition** (Multimodal Contrastive Loss (MCL Loss)). Let $(X, Y)$ be an $N$-pair configuration, where $X = (x_1, \ldots, x_N) \in (\mathbb{S}^{h-1})^N$ and $Y = (y_1, \ldots, y_N) \in (\mathbb{S}^{h-1})^N$. $\forall \tau > 0$, the multimodal contrastive loss $\mathcal{L}_{\text{MCL}}(\cdot, \cdot) : (\mathbb{S}^{h-1})^N \times (\mathbb{S}^{h-1})^N \to \mathbb{R}$ is defined as:

$$\mathcal{L}_{\text{MCL}} = \frac{1}{N}\sum_{i=1}^N \mathcal{L}_{\text{MCL}}^i, \quad \text{where } \mathcal{L}_{\text{MCL}}^i = \mathcal{L}_{\mathcal{X}\to\mathcal{Y}}(x_i; Y) + \mathcal{L}_{\mathcal{Y}\to\mathcal{X}}(y_i; X).$$

Here, $\mathcal{L}_{\mathcal{X}\to\mathcal{Y}}$ is the $\mathcal{X}$-to-$\mathcal{Y}$ alignment and $\mathcal{L}_{\mathcal{Y}\to\mathcal{X}}$ is the $\mathcal{Y}$-to-$\mathcal{X}$ alignment, which are defined respectively as:

$$\mathcal{L}_{\mathcal{X}\to\mathcal{Y}}(x_i; Y) = -\log \frac{\exp(x_i \cdot y_i/\tau)}{\sum_{j=1}^N \exp(x_i \cdot y_j/\tau)}, \quad \mathcal{L}_{\mathcal{Y}\to\mathcal{X}}(y_i; X) = -\log \frac{\exp(x_i \cdot y_i/\tau)}{\sum_{j=1}^N \exp(x_j \cdot y_i/\tau)}.$$

**Definition** (Modality Gap) Let $(X, Y)$ be an $N$-pair configuration, where $X = (x_1, \ldots, x_N) \in (\mathbb{S}^{h-1})^N$ and $Y = (y_1, \ldots, y_N) \in (\mathbb{S}^{h-1})^N$. The modality gap between $X$ and $Y$ can be expressed as the angle between the center representations:

$$\Delta_\theta = \cos^{-1}(c_x \cdot c_y).$$

**Definition** (vMF Distribution). $\forall c \in \mathbb{S}^{h-1}$ and $\kappa \geq 0$, the probability density of a random $h$-dimensional unit vector $z \sim \text{vMF}(c, \kappa)$ is given by:

$$f_h(z; c, \kappa) = D_h(\kappa) e^{\kappa c^\top z}, \quad \text{where} \quad D_h(\kappa) = \frac{\kappa^\nu}{(2\pi)^{\nu+1} I_\nu(\kappa)}.$$

Here, $\nu = h/2 - 1$, and $I_\nu(\cdot) : \mathbb{R} \to \mathbb{R}$ is the modified Bessel function of the first kind of order $\nu$, which is defined as:

$$I_\nu(x) = \sum_{k=0}^{\infty} \frac{1}{k! \Gamma(\nu + k + 1)} \left( \frac{x}{2} \right)^{2k+\nu}.$$

**Definition** (Function $\tilde{M}$). $\forall \kappa, \tau > 0$, a function $\tilde{M}_\kappa(\cdot, \cdot) : [-1, 1] \times [0, 1] \to \mathbb{R}_0^+$ is defined as:

$$\tilde{M}_\kappa(w, t) = \sqrt{\kappa^2 + \frac{2\kappa w}{\tau} + \frac{t^2}{\tau^2}}.$$

**Definition** (Function $\tilde{\mathcal{J}}$). $\forall \kappa, \nu, \tau > 0, \tilde{\mathcal{J}}(\cdot, \cdot, \cdot; \kappa, \nu) : [-1, 1] \times [-1, 1] \times [0, 1] \to \mathbb{R}$ is defined as:

$$\tilde{\mathcal{J}}(w_1, w_2, t; \kappa, \nu) = -\frac{w_1}{\tau} + \log \left( \frac{I_\nu\left(\tilde{M}_\kappa(w_2, t)\right)}{\tilde{M}_\kappa(w_2, t)^\nu} \right) - \log \left( \frac{I_\nu(\kappa)}{\kappa^\nu} \right).$$

**Definition** (Function $M$). $\forall \kappa, \tau > 0$, a function $M_\kappa(\cdot) : [-1, 1] \to \mathbb{R}_0^+$ is defined as:

$$M_\kappa(w) = \sqrt{\kappa^2 + \frac{2\kappa w}{\tau} + \frac{1}{\tau^2}}$$
$$= \tilde{M}_\kappa(w, 1).$$

**Definition** (Function $\mathcal{J}$). $\forall \kappa, \nu, \tau > 0$, a function $\mathcal{J}(\cdot; \kappa, \nu) : [-1, 1] \to \mathbb{R}$ is defined as:

$$\mathcal{J}(w; \kappa, \nu) = -\frac{w}{\tau} + \log \left( \frac{I_\nu(M_\kappa(w))}{M_\kappa(w)^\nu} \right) - \log \left( \frac{I_\nu(\kappa)}{\kappa^\nu} \right)$$
$$= \tilde{\mathcal{J}}(w, w, 1; \kappa, \nu).$$

**Definition** (Function $\tilde{M}$). $\forall \kappa, \tau > 0$, a function $\tilde{M}_\kappa(\cdot, \cdot) : [-1, 1] \times [0, 1] \to \mathbb{R}_0^+$ is defined as:

$$\tilde{M}_\kappa(w, t) = \tilde{M}_\kappa(w, t).$$

**Definition** (Function $\hat{\mathcal{J}}$). $\forall \kappa, \nu, \tau > 0$, a function $\hat{\mathcal{J}}(\cdot, \cdot; \kappa, \nu) : [-1, 1] \times [0, 1] \to \mathbb{R}$ is defined as:

$$\hat{\mathcal{J}}(w, t; \kappa, \nu) = -\frac{w}{\tau} + \log \left( \frac{I_\nu\left(\tilde{M}_\kappa(w, t)\right)}{\tilde{M}_\kappa(w, t)^\nu} \right) - \log \left( \frac{I_\nu(\kappa)}{\kappa^\nu} \right)$$
$$= \tilde{\mathcal{J}}(w, w, t; \kappa, \nu).$$

### E.4.1 PROOF OF THEOREM 4

In this subsection, we provide the proof of Theorem 4. For convenience in reading, we first restate Theorem 4 here.

**Theorem 4.** [Restate] Let $(X, Y)$ be an $N$-pair configuration, where $X = (x_1, \ldots, x_N) \in (\mathbb{S}_X \setminus \mathbb{C})^N$ are $iid$ samples from $\mu_x = \text{vMF}(c_x, \kappa_x)$, and $Y = (y_1, \ldots, y_N) \in (\mathbb{S}_Y \setminus \mathbb{C})^N$ are $iid$ samples from $\mu_y = \text{vMF}(c_y, \kappa_y)$. Let $\tilde{\nu} = (h-1)/2 - 1$. Denote $\Delta_\theta = \cos^{-1}(c_x \cdot c_y)$ and assume $c_x, c_y \perp \mathbb{C}$ with $c_x \cdot c_y > 0$. Suppose $(X, Y)$ achieves Intra-Modal Isometry. Then $\forall i \in [N]$, denote $\theta_i^c = \cos^{-1}(x_i \cdot c_x) = \cos^{-1}(y_i \cdot c_y)$, and $\kappa = \kappa_x = \kappa_y$. Let $\theta_i^c \in (0, \frac{\pi}{2})$ and $\kappa > 0$, it holds that:

$$\lim_{N \to \infty} \mathcal{L}_{\text{MCL}}^{i \neq c} - 2\log(N)$$

$$= \tilde{\mathcal{J}}\left(\cos(\Delta_\theta), \cos(\theta_i^c), \|P_B x_i\|; \kappa, \tilde{\nu}\right) + \tilde{\mathcal{J}}\left(\cos(\Delta_\theta), \cos(\theta_i^c), \|P_A y_i\|; \kappa, \tilde{\nu}\right)$$

$$\geq 2\tilde{\mathcal{J}}\left(\cos^2(\theta_i^c)\cos(\phi_{\min}) + \sin^2(\theta_i^c), \cos(\theta_i^c), \sqrt{\cos^2(\theta_i^c)\cos^2(\phi_{\min}) + \sin^2(\theta_i^c)}; \kappa, \tilde{\nu}\right),$$

where equality is attained if and only if there exists a configuration of $(X, Y)$ such that:

(A8) $P_C x_i = P_C y_i \neq \vec{0}$.

(A9) $\Delta_\theta = \cos^{-1}(c_x \cdot c_y) = \phi_{\min}$.

*Proof.* According to Theorem S7, the convergent function of $\lim_{N \to \infty} \mathcal{L}_{\text{MCL}}^{i \neq c} - 2\log(N)$ is:

$$\lim_{N \to \infty} \mathcal{L}_{\text{MCL}}^{i \neq c} - 2\log(N) = \lim_{N \to \infty} \left(\mathcal{L}_{\mathcal{X} \to \mathcal{Y}}(x_{i \neq c}; Y) - \log(N) + \mathcal{L}_{\mathcal{Y} \to \mathcal{X}}(y_{i \neq c}; X) - \log(N)\right)$$

$$= \tilde{\mathcal{J}}\left(w_i, w_i^c, \|P_B x_i\|; \kappa_y, \tilde{\nu}\right) + \tilde{\mathcal{J}}\left(w_i, w_i^c, \|P_A y_i\|; \kappa_x, \tilde{\nu}\right)$$

$$= 2\tilde{\mathcal{J}}\left(w_i, w_i^c, t; \kappa, \tilde{\nu}\right),$$
(315)

where

$$w_i = \cos^2(\theta_i^c)\cos(\Delta_\theta) + (\theta_i^c)(P_C \cdot x_i) \cdot (P_C \cdot y_i),$$
$$w_i^c = \cos(\theta_i^c),$$
$$t = \sqrt{\cos^2(\theta_i^c)\cos^2(\Delta_\theta) + \sin^2(\theta_i^c)}.$$
(316)

And Theorem S8 shows the lower bound of the convergent function is:

$$2\tilde{\mathcal{J}}\left(w_i, w_i^c, t; \kappa, \tilde{\nu}\right) \geq 2\tilde{\mathcal{J}}\left(w_{i,\min}, w_i^c, t_{\min}; \kappa, \tilde{\nu}\right),$$
(317)

where

$$w_{i,\min} = \cos^2(\theta_i^c)\cos(\phi_{\min}) + \sin^2(\theta_i^c),$$
$$t_{\min} = \sqrt{\cos^2(\theta_i^c)\cos^2(\phi_{\min}) + \sin^2(\theta_i^c)},$$
(318)

and equality is attained if and only if there exists a configuration of $(X, Y)$ such that:

(i) $P_C \cdot x_i = P_C \cdot y_i$.

(ii) $\Delta_\theta = \phi_{\min}$.

Combining Eq. (315) and Eq. (318), we conclude that:

$$\lim_{N \to \infty} \mathcal{L}_{\text{MCL}}^{i \neq c} - 2 \log(N)$$

$$= \tilde{\mathcal{J}} \left( \cos \left( \Delta_\theta \right), \cos \left( \theta_i^c \right), \| P_B x_i \|; \kappa, \tilde{\nu} \right) + \tilde{\mathcal{J}} \left( \cos \left( \Delta_\theta \right), \cos \left( \theta_i^c \right), \| P_A y_i \|; \kappa, \tilde{\nu} \right)$$

$$\geq 2 \tilde{\mathcal{J}} \left( \cos^2 \left( \theta_i^c \right) \cos \left( \phi_{\min} \right) + \sin^2 \left( \theta_i^c \right), \cos \left( \theta_i^c \right), \sqrt{\cos^2 \left( \theta_i^c \right) \cos^2 \left( \phi_{\min} \right) + \sin^2 \left( \theta_i^c \right)}; \kappa, \tilde{\nu} \right),$$

$$(319)$$

where equality is attained if and only if there exists a configuration of $(X, Y)$ such that:

(A8) $P_C x_i = P_C y_i \neq \vec{0}$.

(A9) $\Delta_\theta = \cos^{-1} \left( c_x \cdot c_y \right) = \phi_{\min}$.

$\square$

### E.4.2 AUXILIARY THEOREMS PART 4

In this subsection, we provide details and proofs of the auxiliary theorems (Theorem S5 and Theorem S7) that support the proof of Theorem 4.

**Theorem S7.** *Let $(X, Y)$ be an $N$-pair configuration, where $X = (x_1, \ldots, x_N) \in (\mathbb{S}_X \setminus \mathbb{C})^N$ are iid samples from $\mu_x = \text{vMF}(c_x, \kappa_x)$, and $Y = (y_1, \ldots, y_N) \in (\mathbb{S}_Y \setminus \mathbb{C})^N$ are iid samples from $\mu_y = \text{vMF}(c_y, \kappa_y)$. Let $\tilde{\nu} = (h - 1)/2 - 1$. Denote $\Delta_\theta = \cos^{-1} \left( c_x \cdot c_y \right)$ and assume $c_x, c_y \perp \mathbb{C}$ with $c_x \cdot c_y > 0$. Suppose $(X, Y)$ achieves Intra-Modal Isometry. Then $\forall i \in [N]$, denote $\theta_i^c = \cos^{-1} \left( x_i \cdot c_x \right) = \cos^{-1} \left( y_i \cdot c_y \right)$, and $\kappa = \kappa_x = \kappa_y$. Let $\kappa > 0$, it holds that:*

$$\lim_{N \to \infty} \mathcal{L}_{\text{MCL}}^{i \neq c} - 2 \log(N) = \lim_{N \to \infty} \left( \mathcal{L}_{\mathcal{X} \to \mathcal{Y}}(x_{i \neq c}; Y) - \log(N) + \mathcal{L}_{\mathcal{Y} \to \mathcal{X}}(y_{i \neq c}; X) - \log(N) \right)$$

$$= \tilde{\mathcal{J}} \left( w_i, w_i^c, \| P_B x_i \|; \kappa_y, \tilde{\nu} \right) + \tilde{\mathcal{J}} \left( w_i, w_i^c, \| P_A y_i \|; \kappa_x, \tilde{\nu} \right)$$

$$= 2 \tilde{\mathcal{J}} \left( w_i, w_i^c, t; \kappa, \tilde{\nu} \right),$$

$$(320)$$

*where*

$$w_i = \cos^2 \left( \theta_i^c \right) \cos \left( \Delta_\theta \right) + \left( \theta_i^c \right) \left( P_C \cdot x_i \right) \cdot \left( P_C \cdot y_i \right),$$

$$w_i^c = \cos \left( \theta_i^c \right),$$

$$t = \sqrt{\cos^2 \left( \theta_i^c \right) \cos^2 \left( \Delta_\theta \right) + \sin^2 \left( \theta_i^c \right)}.$$

$$(321)$$

*Proof.* **Step 1**: We first decompose $\lim_{N \to \infty} \mathcal{L}_{\text{MCL}}^{i \neq c} - 2 \log(N)$ into two parts:

$$\lim_{N \to \infty} \mathcal{L}_{\text{MCL}}^{i \neq c} - 2 \log(N) = \lim_{N \to \infty} \mathcal{L}_{\mathcal{X} \to \mathcal{Y}}(x_{i \neq c}; Y) - \log(N)$$

$$+ \lim_{N \to \infty} \mathcal{L}_{\mathcal{Y} \to \mathcal{X}}(y_{i \neq c}; X) - \log(N).$$

$$(322)$$

The convergent function of $\mathcal{L}_{\mathcal{X} \to \mathcal{Y}}(x_{i \neq c}; Y)$ as $N \to \infty$. $\forall i \in [N], i \neq c, x_i \in X$, denote $w_i = x_i \cdot y_i, w_{x_i, c_y} = x_i \cdot c_y$ and $w_{y_i, c_x} = y_i \cdot c_x$. $\forall \kappa_y > 0$, as prove in Theorem S5:

$$\lim_{N \to \infty} \mathcal{L}_{\mathcal{X} \to \mathcal{Y}}(x_i; Y) - \log(N) = \lim_{N \to \infty} - \log \frac{\exp\left(x_i \cdot y_i/\tau\right)}{\sum_{j=1}^{N} \exp\left(x_i \cdot y_j/\tau\right)} - \log(N)$$

$$= -\frac{w_i}{\tau} + \log\left(\frac{I_{\tilde{\nu}}\left(\tilde{M}_{\kappa_y}\left(w_{x_i,c_y}, \|P_B x_i\|\right)\right)}{\tilde{M}_{\kappa_y}\left(w_{x_i,c_y}, \|P_B x_i\|\right)^{\tilde{\nu}}}\right) - \log\left(\frac{I_{\tilde{\nu}}\left(\kappa_y\right)}{\kappa_y^{\tilde{\nu}}}\right) \quad (323)$$

$$= \tilde{\mathcal{J}}\left(w_i, w_{x_i,c_y}, \|P_B c_x\|; \kappa_y, \tilde{\nu}\right),$$

where $\forall \kappa, \tau > 0$, $\tilde{\mathcal{J}}(\cdot, \cdot, \cdot; \kappa, \tilde{\nu})$ is a function on $[-1, 1] \times [-1, 1] \times [0, 1]$ and $\tilde{M}_{\kappa}(\cdot, \cdot) : [-1, 1] \times [0, 1] \to \mathbb{R}_0^+$ is defined as:

$$\tilde{M}_{\kappa}(w, t) = \sqrt{\kappa^2 + \frac{2\kappa w}{\tau} + \frac{t^2}{\tau^2}}, \quad (324)$$

and $I_{\nu}$ is the modified Bessel function of the first kind of order $\nu$, which is defined as:

$$I_{\nu}(m) = \sum_{k=0}^{\infty} \frac{1}{k! \Gamma(\nu + k + 1)} \left(\frac{m}{2}\right)^{2k+\nu}. \quad (325)$$

When $(X, Y)$ achieves Intra-Modal Isometry, we have $w_{x_i,c_y} = x_i \cdot c_x = y_i \cdot c_x = w_{y_i,c_x}$ Denote $w_i^c = w_{x_i,c_y} = w_{y_i,c_x} = \cos(\theta_i^c)$. This implies $\kappa_x = \kappa_y = \kappa$.

Then, Eq. (323) can be re-written as:

$$\lim_{N \to \infty} \mathcal{L}_{\mathcal{X} \to \mathcal{Y}}(x_i; Y) - \log(N) = -\frac{w_i}{\tau} + \log\left(\frac{I_{\tilde{\nu}}\left(\tilde{M}_{\kappa}\left(w_i^c, \|P_B x_i\|\right)\right)}{\tilde{M}_{\kappa}\left(w_i^c, \|P_B x_i\|\right)^{\tilde{\nu}}}\right) - \log\left(\frac{I_{\tilde{\nu}}(\kappa)}{\kappa^{\tilde{\nu}}}\right)$$

$$= \tilde{\mathcal{J}}\left(w_i, w_i^c, \|P_B c_x\|; \kappa, \tilde{\nu}\right). \quad (326)$$

Similarly, the convergent function of $\mathcal{L}_{\mathcal{Y} \to \mathcal{X}}(y_{i \neq c}; X)$ as $N \to \infty$ can be written as:

$$\lim_{N \to \infty} \mathcal{L}_{\mathcal{Y} \to \mathcal{X}}(y_i; X) - \log(N) = \lim_{N \to \infty} - \log \frac{\exp\left(x_i \cdot y_i/\tau\right)}{\sum_{j=1}^{N} \exp\left(x_i \cdot y_j/\tau\right)} - \log(N)$$

$$= -\frac{w_i}{\tau} + \log\left(\frac{I_{\tilde{\nu}}\left(\tilde{M}_{\kappa_x}\left(w_{y_i,c_x}, \|P_A y_i\|\right)\right)}{\tilde{M}_{\kappa_x}\left(w_{y_i,c_x}, \|P_A y_i\|\right)^{\tilde{\nu}}}\right) - \log\left(\frac{I_{\tilde{\nu}}\left(\kappa_x\right)}{\kappa_x^{\tilde{\nu}}}\right)$$

$$= -\frac{w_i}{\tau} + \log\left(\frac{I_{\tilde{\nu}}\left(\tilde{M}_{\kappa}\left(w_i^c, \|P_A y_i\|\right)\right)}{\tilde{M}_{\kappa}\left(w_i^c, \|P_A y_i\|\right)^{\tilde{\nu}}}\right) - \log\left(\frac{I_{\tilde{\nu}}(\kappa)}{\kappa^{\tilde{\nu}}}\right) \quad (327)$$

$$= \tilde{\mathcal{J}}\left(w_i, w_i^c, \|P_A y_i\|; \kappa, \tilde{\nu}\right).$$

**Step 2** Now, let us decompose the embedding space. Define two vectors $e_A$ and $e_B$ such that:

$$e_A \in \mathbb{S}_X, \text{ and } e_A \perp \mathbb{C},$$
$$e_B \in \mathbb{S}_Y, \text{ and } e_B \perp \mathbb{C}. \quad (328)$$

Let $\mathbb{C}^{\perp}$ be the 2-dimensional orthogonal complement of $C$, and $\mathbb{C}^{\perp}$ satisfies:

$$\mathbb{C}^{\perp} = \mathrm{span}\left\{e_A\right\} \oplus \mathrm{span}\left\{e_B\right\},$$
$$\mathbb{R}^h = \mathbb{C} \oplus \mathbb{C}^{\perp}.$$
(329)

Since $n_A, n_B \in \mathbb{C}^{\perp}$, $n_A \perp e_A$ and $n_B \perp e_B$, we have:

$$\langle e_A, e_B \rangle = \pm \langle n_A, n_B \rangle,$$
(330)

and we choose a pair of $e_A$ and $e_B$ such that:

$$\langle e_A, e_B \rangle = \langle n_A, n_B \rangle = \cos(\phi) \in (0, 1).$$
(331)

Denote $\theta_i = \cos^{-1}(w_i)$. When $c_x, c_y \perp \mathbb{C}$, $\Delta_\theta = \phi$. And without loss of generality, we can set the coordinate as:

$$n_A = (\sin\left(\frac{\Delta_\theta}{2}\right), -\cos\left(\frac{\Delta_\theta}{2}\right), 0, 0, \cdots, 0),$$
$$n_B = (-\sin\left(\frac{\Delta_\theta}{2}\right), -\cos\left(\frac{\Delta_\theta}{2}\right), 0, 0, \cdots, 0),$$
$$c_x = e_A = (\cos\left(\frac{\Delta_\theta}{2}\right), \sin\left(\frac{\Delta_\theta}{2}\right), 0, 0, \cdots, 0),$$
$$c_y = e_B = (\cos\left(\frac{\Delta_\theta}{2}\right), -\sin\left(\frac{\Delta_\theta}{2}\right), 0, 0, \cdots, 0),$$
$$\mathbb{C} = \mathrm{span}\{e_3\} \oplus \mathrm{span}\{e_3\} \oplus \cdots, \oplus\mathrm{span}\{e_h\}.$$
(332)

Therefore, $\forall x_i \in \mathbb{S}_X = \mathbb{A} \cap \mathbb{S}^{h-1}$ and $\forall y_i \in \mathbb{S}_Y = \mathbb{B} \cap \mathbb{S}^{h-1}$, $\exists u_i^x, u_i^y \in \mathbb{C} \cap \mathbb{S}^{h-1}$, such that:

$$x_i = \cos(\theta_i^c) e_A + \sin(\theta_i^c) u_i^x = \cos(\theta_i^c) c_x + \sin(\theta_i^c) u_i^x,$$
$$y_i = \cos(\theta_i^c) e_B + \sin(\theta_i^c) u_i^y = \cos(\theta_i^c) c_y + \sin(\theta_i^c) u_i^y.$$
(333)

Using orthogonality, we have:

$$P_B \cdot e_A = \langle e_A, e_B \rangle e_B = \cos(\Delta_\theta) e_B,$$
$$P_B \cdot u_i^x = u_i^x,$$
(334)

and

$$P_A \cdot e_B = \langle e_A, e_B \rangle e_A = \cos(\Delta_\theta) e_A,$$
$$P_A \cdot u_i^y = u_i^y,$$
(335)

and

$$P_C \cdot e_A = P_C \cdot e_B = 0,$$
$$P_C \cdot u_i^x = u_i^x,$$
$$P_C \cdot u_i^y = u_i^y.$$
(336)

Then the projections of $(x_i, y_i)$ are:

$$P_B \cdot x_i = \cos(\theta_i^c) \cos(\Delta_\theta) e_B + \sin(\theta_i^c) u_i^x = \cos(\theta_i^c) \cos(\Delta_\theta) c_y + \sin(\theta_i^c) u_i^x,$$
$$P_A \cdot y_i = \cos(\theta_i^c) \cos(\Delta_\theta) e_A + \sin(\theta_i^c) u_i^y = \cos(\theta_i^c) \cos(\Delta_\theta) c_x + \sin(\theta_i^c) u_i^y,$$
(337)

and

$$P_C \cdot x_i = \sin\left(\theta_i^c\right) u_i^x,$$
$$P_C \cdot y_i = \sin\left(\theta_i^c\right) u_i^y. \tag{338}$$

Therefore, we get:

$$\begin{aligned}
w_i = x_i \cdot y_i &= \cos^2\left(\theta_i^c\right) c_x \cdot c_y + \sin^2\left(\theta_i^c\right) u_i^x \cdot u_i^y \\
&= \cos^2\left(\theta_i^c\right) \cos\left(\Delta_\theta\right) + \sin^2\left(\theta_i^c\right) u_i^x \cdot u_i^y \\
&= \cos^2\left(\theta_i^c\right) \cos\left(\Delta_\theta\right) + \left(P_C \cdot x_i\right) \cdot \left(P_C \cdot y_i\right),
\end{aligned} \tag{339}$$

$$\begin{aligned}
\|P_B x_i\| &= \sqrt{\cos^2\left(\theta_i^c\right) \cos^2\left(\Delta_\theta\right) c_y \cdot c_y + 2\cos\left(\theta_i^c\right) \cos\left(\Delta_\theta\right) \sin\left(\theta_i^c\right) c_y \cdot u_i^x + \sin^2\left(\theta_i^c\right) u_i^x \cdot u_i^x} \\
&= \sqrt{\cos^2\left(\theta_i^c\right) \cos^2\left(\Delta_\theta\right) + 2\cos\left(\theta_i^c\right) \cos\left(\Delta_\theta\right) \sin\left(\theta_i^c\right) c_y \cdot u_i^x + \sin^2\left(\theta_i^c\right)} \\
&= \sqrt{\cos^2\left(\theta_i^c\right) \cos^2\left(\Delta_\theta\right) + \sin^2\left(\theta_i^c\right)},
\end{aligned} \tag{340}$$

and

$$\begin{aligned}
\|P_A y_i\| &= \sqrt{\cos^2\left(\theta_i^c\right) \cos^2\left(\Delta_\theta\right) c_x \cdot c_x + 2\cos\left(\theta_i^c\right) \cos\left(\Delta_\theta\right) \sin\left(\theta_i^c\right) c_x \cdot u_i^y + \sin^2\left(\theta_i^c\right) u_i^y \cdot u_i^y} \\
&= \sqrt{\cos^2\left(\theta_i^c\right) \cos^2\left(\Delta_\theta\right) + 2\cos\left(\theta_i^c\right) \cos\left(\Delta_\theta\right) \sin\left(\theta_i^c\right) c_y \cdot u_i^y + \sin^2\left(\theta_i^c\right)} \\
&= \sqrt{\cos^2\left(\theta_i^c\right) \cos^2\left(\Delta_\theta\right) + \sin^2\left(\theta_i^c\right)}, \\
&= \|P_B x_i\|.
\end{aligned} \tag{341}$$

Let $t = \|P_B x_i\| = \|P_A y_i\|$. Plugging Eq. (339), Eq. (340) and Eq. (341) into Eq. (322), Eq. (326) and Eq. (327), we conclude that:

$$\begin{aligned}
\lim_{N\to\infty} \mathcal{L}_{\text{MCL}}^{i\neq c} - 2\log(N) &= \lim_{N\to\infty} \mathcal{L}_{\mathcal{X}\to\mathcal{Y}}(x_{i\neq c}; Y) - \log(N) \\
&\quad + \lim_{N\to\infty} \mathcal{L}_{\mathcal{Y}\to\mathcal{X}}(y_{i\neq c}; X) - \log(N) \\
&= \tilde{\mathcal{J}}\left(w_i, w_i^c, \|P_B x_i\|; \kappa, \tilde{\nu}\right) + \tilde{\mathcal{J}}\left(w_i, w_i^c, \|P_A y_i\|; \kappa, \tilde{\nu}\right) \\
&= 2\tilde{\mathcal{J}}\left(w_i, w_i^c, t; \kappa, \tilde{\nu}\right),
\end{aligned} \tag{342}$$

where

$$\begin{aligned}
w_i = x_i \cdot y_i &= \cos^2\left(\theta_i^c\right) \cos\left(\Delta_\theta\right) + \left(P_C \cdot x_i\right) \cdot \left(P_C \cdot y_i\right), \\
w_i^c = x_i \cdot c_y &= y_i \cdot c_x = \cos\left(\theta_i^c\right), \\
t &= \sqrt{\cos^2\left(\theta_i^c\right) \cos^2\left(\Delta_\theta\right) + \sin^2\left(\theta_i^c\right)}.
\end{aligned} \tag{343}$$

$\square$

**Theorem S8.** *Let $(X, Y)$ be an $N$-pair configuration, where $X = (x_1, \ldots, x_N) \in (\mathbb{S}_X \setminus \mathbb{C})^N$ are iid samples from $\mu_x = \text{vMF}(c_x, \kappa_x)$, and $Y = (y_1, \ldots, y_N) \in (\mathbb{S}_Y \setminus \mathbb{C})^N$ are iid samples from $\mu_y = \text{vMF}(c_y, \kappa_y)$. Let $\tilde{\nu} = (h-1)/2 - 1$. Denote $\Delta_\theta = \cos^{-1}(c_x \cdot c_y)$ and assume $c_x, c_y \perp \mathbb{C}$ with $c_x \cdot c_y > 0$. $\forall i \in [N]$, suppose $\theta_i^c = \cos^{-1}(x_i \cdot c_x) = \cos^{-1}(y_i \cdot c_y) \in (0, \frac{\pi}{2})$ and $\kappa > 0$, it holds that:*

$$\tilde{\mathcal{J}}\left(w_i, w_i^c, t; \kappa, \tilde{\nu}\right) \geq \tilde{\mathcal{J}}\left(w_{i,\min}, w_i^c, t_{\min}; \kappa, \tilde{\nu}\right), \tag{344}$$

*where*

$$w_i = \cos^2(\theta_i^c)\cos(\Delta_\theta) + (P_C \cdot x_i) \cdot (P_C \cdot y_i),$$
$$w_i^c = \cos(\theta_i^c),$$
$$t = \sqrt{\cos^2(\theta_i^c)\cos^2(\Delta_\theta) + \sin^2(\theta_i^c)}, \tag{345}$$
$$w_{i,\min} = \cos^2(\theta_i^c)\cos(\phi_{\min}) + \sin^2(\theta_i^c),$$
$$t_{\min} = \sqrt{\cos^2(\theta_i^c)\cos^2(\phi_{\min}) + \sin^2(\theta_i^c)},$$

*and equality is attained if and only if there exists a configuration of* $(X, Y)$ *such that:*

*(B6)* $P_C \cdot x_i = P_C \cdot y_i$.

*(B7)* $\Delta_\theta = \phi_{\min}$.

*Proof.* **Step 1**: Similarly to the proof of Theorem S6 in Sec. E.3.2, we start the proof by finding the convergent function of $\lim_{N\to\infty} \mathcal{L}_{\mathrm{MCL}}^{i\neq c} - 2\log(N)$ as $N \to \infty$. Let $w_i =$

As proven in Theorem S7:

$$\lim_{N\to\infty} \mathcal{L}_{\mathrm{MCL}}^{i\neq c} - 2\log(N) = \lim_{N\to\infty} (\mathcal{L}_{\mathcal{X}\to\mathcal{Y}}(x_{i\neq c}; Y) - \log(N) + \mathcal{L}_{\mathcal{Y}\to\mathcal{X}}(y_{i\neq c}; X) - \log(N))$$
$$= \tilde{\mathcal{J}}(w_i, w_i^c, \|P_B x_i\|; \kappa, \tilde{\nu}) + \tilde{\mathcal{J}}(w_i, w_i^c, \|P_A y_i\|; \kappa, \tilde{\nu})$$
$$= 2\tilde{\mathcal{J}}(w_i, w_i^c, t; \kappa, \tilde{\nu}). \tag{346}$$

$\forall \kappa, \nu, \tau > 0$, $\tilde{\mathcal{J}}(\cdot, \cdot, \cdot; \kappa, \nu) : [-1, 1] \times [-1, 1] \times [0, 1] \to \mathbb{R}$ is defined as:

$$\tilde{\mathcal{J}}(w_1, w_2, t; \kappa, \nu) = -\frac{w_1}{\tau} + \log\left(\frac{I_\nu\left(\tilde{M}_\kappa(w_2, t)\right)}{\tilde{M}_\kappa(w_2, t)^\nu}\right) - \log\left(\frac{I_\nu(\kappa)}{\kappa^\nu}\right), \tag{347}$$

and $\tilde{M}_\kappa(\cdot, \cdot) : [-1, 1] \times [0, 1] \to \mathbb{R}_0^+$ is defined as:

$$\tilde{M}_\kappa(w, t) = \sqrt{\kappa^2 + \frac{2\kappa w}{\tau} + \frac{t^2}{\tau^2}}. \tag{348}$$

and $I_\nu$ is the modified Bessel function of the first kind of order $\nu$, which is defined as:

$$I_\nu(m) = \sum_{k=0}^{\infty} \frac{1}{k!\Gamma(\nu+k+1)} \left(\frac{m}{2}\right)^{2k+\nu}, \tag{349}$$

and

$$w_i = \cos^2(\theta_i^c)\cos(\Delta_\theta) + \sin^2(\theta_i^c)(P_C \cdot x_i) \cdot (P_C \cdot y_i),$$
$$w_i^c = \cos(\theta_i^c), \tag{350}$$
$$t = \sqrt{\cos^2(\theta_i^c)\cos^2(\Delta_\theta) + \sin^2(\theta_i^c)}.$$

**Step 2:**

According to the Cauchy-Schwarz inequality and Eq. (338):

$$(P_C \cdot x_i) \cdot (P_C \cdot y_i) \leq \sin^2(\theta_i^c), \tag{351}$$

where equality is attained if and only if there exists a configuration of $(X, Y)$ such that:

(B6) $P_C \cdot x_i = P_C \cdot y_i$.

And therefore:

$$
\begin{aligned}
w_i &= \cos^2\left(\theta_i^c\right)\cos\left(\Delta_\theta\right) + \left(P_C \cdot x_i\right) \cdot \left(P_C \cdot y_i\right) \\
&\leq \cos^2\left(\theta_i^c\right)\cos\left(\Delta_\theta\right) + \leq \sin^2\left(\theta_i^c\right),
\end{aligned}
\tag{352}
$$

and then $\tilde{\mathcal{J}}\left(w_i, w_i^c, t; \kappa, \tilde{\nu}\right)$ in Eq. (346) can be bounded below by:

$$
\begin{aligned}
\tilde{\mathcal{J}}\left(w_i, w_i^c, t; \kappa, \tilde{\nu}\right) \geq \tilde{\mathcal{J}}\big( &\cos^2\left(\theta_i^c\right)\cos\left(\Delta_\theta\right) + \sin^2\left(\theta_i^c\right), \\
&\cos\left(\theta_i^c\right), \\
&\sqrt{\cos^2\left(\theta_i^c\right)\cos^2\left(\Delta_\theta\right) + \sin^2\left(\theta_i^c\right)}; \kappa, \tilde{\nu}\big).
\end{aligned}
\tag{353}
$$

Here, for any given non-center pair $(x_i, y_i)_{i \neq c}$, $\theta_i^c$ is fixed, then the RHS of Eq. (353) becomes a function of $\cos\left(\Delta_\theta\right)$.

Denote:

$$
\begin{aligned}
f_1\left(\cos\left(\Delta_\theta\right)\right) &:= \cos^2\left(\theta_i^c\right)\cos\left(\Delta_\theta\right) + \sin^2\left(\theta_i^c\right), \\
f_2\left(\cos\left(\Delta_\theta\right)\right) &:= \sqrt{\cos^2\left(\theta_i^c\right)\cos^2\left(\Delta_\theta\right) + \sin^2\left(\theta_i^c\right)},
\end{aligned}
\tag{354}
$$

then the Eq. (346) can be re-written as:

$$
\tilde{\mathcal{J}}\left(w_i, w_i^c, t; \kappa, \tilde{\nu}\right) \geq \tilde{\mathcal{J}}\left(f_1\left(\cos\left(\Delta_\theta\right)\right), \cos\left(\theta_i^c\right), f_2\left(\cos\left(\Delta_\theta\right)\right); \kappa, \tilde{\nu}\right).
\tag{355}
$$

According to Lemma 14, $\tilde{\mathcal{J}}\left(f_1\left(\cos\left(\Delta_\theta\right)\right), \cos\left(\theta_i^c\right), f_2\left(\cos\left(\Delta_\theta\right)\right); \kappa, \tilde{\nu}\right)$ is a decreasing function of $\cos\left(\Delta_\theta\right)$ when $\theta_i^c \in [0, \frac{\pi}{2}]$, we have:

$$
\tilde{\mathcal{J}}\left(f_1\left(\cos\left(\Delta_\theta\right)\right), \cos\left(\theta_i^c\right), f_2\left(\cos\left(\Delta_\theta\right)\right)\right) \geq \tilde{\mathcal{J}}\left(f_1\left(\cos\left(\phi_{\min}\right)\right), \cos\left(\theta_i^c\right), f_2\left(\cos\left(\phi_{\min}\right)\right)\right).
\tag{356}
$$

where equality is attained if and only if there exists a configuration of $(X, Y)$ such that:

(B7) $\Delta_\theta = \phi_{\min}$.

Combining Eq. (351) and Eq. (356), we conclude that:

$$
\tilde{\mathcal{J}}\left(w_i, w_i^c, t; \kappa, \tilde{\nu}\right) \geq \tilde{\mathcal{J}}\left(w_{i,\min}, w_i^c, t_{\min}; \kappa, \tilde{\nu}\right),
\tag{357}
$$

where

$$
\begin{aligned}
w_i &= \cos^2\left(\theta_i^c\right)\cos\left(\Delta_\theta\right) + \left(P_C \cdot x_i\right) \cdot \left(P_C \cdot y_i\right), \\
w_i^c &= \cos\left(\theta_i^c\right), \\
t &= \sqrt{\cos^2\left(\theta_i^c\right)\cos^2\left(\Delta_\theta\right) + \sin^2\left(\theta_i^c\right)}, \\
w_{i,\min} &= \cos^2\left(\theta_i^c\right)\cos\left(\phi_{\min}\right) + \sin^2\left(\theta_i^c\right), \\
t_{\min} &= \sqrt{\cos^2\left(\theta_i^c\right)\cos^2\left(\phi_{\min}\right) + \sin^2\left(\theta_i^c\right)}.
\end{aligned}
\tag{358}
$$

and equality is attained if and only if there exists a configuration of $(X, Y)$ such that:

(B6) $P_C \cdot x_i = P_C \cdot y_i$.

(B7) $\Delta_\theta = \phi_{\min}$.

$\square$

### E.4.3 PROOFS COROLLARY 2,3,4

In this subsection, we provide the proofs of Corollary 2, Corollary 3 and Corollary 4. Note that these corollaries all follow the conditions described in Theorem 3 and Theorem 4. For convenience in reading, we restate Corollary 2,3 4 before the proofs.

**Corollary 2.** $\forall i \in [N], i \neq c$, if $c_x, c_y \perp \mathbb{C}$ and $P_C x_i = P_C y_i \neq \vec{0}$ and $\phi > 0$, then the following holds:

(A10) $(x_i, y_i)_{i \neq c}$ are not perfectly aligned.

*Proof.* $\forall (x_i, y_i)_{i \neq c}$, denote $w_i = x_i \cdot y_i$. $(x_i, y_i)$ are perfectly aligned when $w_i$ reach its maximum.

According to Lemma 12, when $x_i$ is fixed $w_i$ is maximized if and only if:

(i) $y_i = \frac{P_B \cdot x_i}{\|P_B \cdot x_i\|}$ .

And when $y_i$ is fixed $w_i$ is maximized if and only if:

(ii) $x_i = \frac{P_A \cdot y_i}{\|P_A \cdot y_i\|}$ .

According to Lemma 13, when $\phi > 0$, $x_i, y_i \not\perp \mathbb{C}$ and $x_i, y_i \notin \mathbb{C}$, we have:

$$
\begin{aligned}
y_i &\neq \frac{P_B \cdot x_i}{\|P_B \cdot x_i\|}, \\
x_i &\neq \frac{P_A \cdot y_i}{\|P_A \cdot y_i\|}.
\end{aligned}
\tag{359}
$$

Therefore, $(x_i, y_i)_{i \neq c}$ are not perfectly aligned.

$\square$

**Corollary 3.** $\forall i \in [N], i \neq c$, if $c_x, c_y \perp \mathbb{C}$, $P_C x_i = P_C y_i$ and $(x_i, y_i)_{i \neq c} \in \mathbb{S}^{h-1} \setminus \mathbb{C}$, then $(x_i, y_i)_{i \neq c}$ are perfectly aligned if the following condition holds:

(A11) $\Delta_\theta = \phi = 0$.

*Proof.* According to Eq. (337) and Eq. (338) in the proof of Theorem S7, the projections of $(x_i, y_i)$ are:

$$
\begin{aligned}
P_B \cdot x_i &= \cos\left(\theta_i^c\right) \cos\left(\Delta_\theta\right) e_B + \sin\left(\theta_i^c\right) u_i^x = \cos\left(\theta_i^c\right) \cos\left(\Delta_\theta\right) c_y + \sin\left(\theta_i^c\right) u_i^x, \\
P_A \cdot y_i &= \cos\left(\theta_i^c\right) \cos\left(\Delta_\theta\right) e_A + \sin\left(\theta_i^c\right) u_i^y = \cos\left(\theta_i^c\right) \cos\left(\Delta_\theta\right) c_x + \sin\left(\theta_i^c\right) u_i^y,
\end{aligned}
\tag{360}
$$

and

$$
\begin{aligned}
P_C \cdot x_i &= \sin\left(\theta_i^c\right) u_i^x, \\
P_C \cdot y_i &= \sin\left(\theta_i^c\right) u_i^y.
\end{aligned}
\tag{361}
$$

Then, when $\phi = \Delta_\theta = 0$

$$P_B \cdot x_i = P_C \cdot x_i = P_C \cdot y_i = P_A \cdot y_i, \tag{362}$$

and

$$x_i = P_B \cdot x_i = P_A \cdot y_i = y_i. \tag{363}$$

In this case, $(x_i, y_i)_{i \neq c}$ are not perfectly aligned.

$\square$

**Corollary 4.** $\forall i \in [N], i \neq c$, if $c_x, c_y \perp \mathbb{C}$ and $P_C x_i = P_C y_i$, then the following holds:

(A12) $(\frac{P_C x_i}{\|P_C x_i\|}, \frac{P_C y_i}{\|P_C y_i\|})_{i \neq c}$ are perfectly aligned

*Proof.* Denote:

$$x_i^* = \frac{P_C x_i}{\|P_C x_i\|},$$
$$y_i^* = \frac{P_C y_i}{\|P_C y_i\|}. \tag{364}$$

Since $P_C x_i = P_C y_i$, then:

$$x_i^* = y_i^*. \tag{365}$$

In this case, $(x_i^*, y_i^*)_{i \neq c}$ are not perfectly aligned.

$\square$

### E.4.4    TECHNICAL LEMMAS PART 4

In this subsection, we provide details and proofs of technical lemmas (Lemma 14 and Lemma 15) that support the proof of Theorem 4, Theorem S7 and Theorem S8.

**Lemma 14.** $\forall \kappa, \nu, \tau > 0$, a function $\bar{\mathcal{J}}(\cdot; \kappa, \nu) : (0, 1] \to \mathbb{R}$ is defined as:

$$\bar{\mathcal{J}}(w_c; \kappa, \nu) = \tilde{\mathcal{J}}(f_1(w_c), \cos(\theta_i^c), f_2(w_c); \kappa, \tilde{\nu}), \tag{366}$$

where $f_1(\cdot) : (0, 1] \to \mathbb{R}_0^+$ and $f_2(\cdot) : [0, 1] \to \mathbb{R}_0^+$ are defined as:

$$f_1(w_c) := \cos^2(\theta_i^c) w_c + \sin^2(\theta_i^c),$$
$$f_2(w_c) := \sqrt{\cos^2(\theta_i^c) w_c^2 + \sin^2(\theta_i^c)}. \tag{367}$$

and $\tilde{\mathcal{J}}(\cdot, \cdot, \cdot; \kappa, \nu) : [-1, 1] \times [-1, 1] \times [0, 1] \to \mathbb{R}$ is defined as:

$$\tilde{\mathcal{J}}(w_1, w_2, t; \kappa, \nu) = -\frac{w_1}{\tau} + \log\left(\frac{I_\nu\left(\tilde{M}_\kappa(w_2, t)\right)}{\tilde{M}_\kappa(w_2, t)^\nu}\right) - \log\left(\frac{I_\nu(\kappa)}{\kappa^\nu}\right), \tag{368}$$

and $\tilde{M}_\kappa(\cdot, \cdot) : [-1, 1] \times [0, 1] \to \mathbb{R}_0^+$ is defined as:

$$\tilde{M}_\kappa(w, t) = \sqrt{\kappa^2 + \frac{2\kappa w}{\tau} + \frac{t^2}{\tau^2}}, \tag{369}$$

and $I_\nu$ is the modified Bessel function of the first kind of order $\nu$, which is defined as:

$$I_\nu(m) = \sum_{k=0}^{\infty} \frac{1}{k!\Gamma(\nu + k + 1)} \left(\frac{m}{2}\right)^{2k+\nu}, \tag{370}$$

It holds that, for any fixed $\theta_i^c \in [0, \frac{\pi}{2}]$, $\bar{\mathcal{J}}(\cdot)$ is a strictly decreasing function on $(0, 1]$.

*Proof.* Let us first decompose the function $\mathcal{J}$. Denote a constant and a function $C_1$ and $G_2(t)$ as:

$$
\begin{aligned}
G_1(w_c) &= -\frac{\cos^2(\theta_i^c) \, w_c}{\tau}, \\
G_3(m) &= \log(I_\nu(m)) - \nu \log(m), \\
G_2(w_c) &= G_3\left(\tilde{M}_\kappa(\cos(\theta_i^c), f_2(w_c))\right) \\
&= \log\left(I_\nu\left(\tilde{M}_\kappa(\cos(\theta_i^c), f_2(w_c))\right)\right) - \nu \log\left(\tilde{M}_\kappa(\cos(\theta_i^c), f_2(w_c))\right).
\end{aligned}
\tag{371}
$$

Denote the function $G(w_c)$ and the constant $C$ as:

$$
\begin{aligned}
G(w_c) &= G_2(w_c) + G_2(w_c), \\
C &= -\log\left(\frac{I_\nu(\kappa)}{\kappa^\nu}\right).
\end{aligned}
\tag{372}
$$

Then the function $\bar{\mathcal{J}}$ can be written as:

$$
\begin{aligned}
\bar{\mathcal{J}}(w_c; \kappa, \nu) &= -\frac{\cos^2(\theta_i^c) \, w_c}{\tau} + \log\left(\frac{I_\nu\left(\tilde{M}_\kappa(\cos(\theta_i^c), f_2(w_c))\right)}{\tilde{M}_\kappa(\cos(\theta_i^c), f_2(w_c))^\nu}\right) - \log\left(\frac{I_\nu(\kappa)}{\kappa^\nu}\right) \\
&= G(w_c) + C.
\end{aligned}
\tag{373}
$$

Now, we investigate derivatives of $G(w_c)$.

The first derivative of $G_1$ is:

$$G_1'(w_c) = -\frac{\cos^2(\theta_i^c)}{\tau} < 0. \tag{374}$$

According to Lemma 7, the first derivative of $G_3(m)$ is:

$$G_3'(m) = \frac{I_{\nu+1}(m)}{I_\nu(m)} \in (0, 1). \tag{375}$$

The derivative of $\tilde{M}_\kappa$ with respect to is $f_2^2(w_c)$:

$$
\begin{aligned}
\tilde{M}_\kappa'(\cos(\theta_i^c), f_2(w_c)) &= \frac{\partial}{\partial f_2^2(w_c)} \tilde{M}_\kappa(\cos(\theta_i^c), f_2(w_c)) \\
&= \frac{\partial}{\partial f_2^2(w_c)} \left(\kappa^2 + \frac{2\kappa \cos(\theta_i^c)}{\tau} + \frac{f_2^2(w_c)}{\tau^2}\right)^{1/2} \\
&= \frac{1}{2}\left(\kappa^2 + \frac{2\kappa \cos(\theta_i^c)}{\tau} + \frac{f_2^2(w_c)}{\tau^2}\right)^{-1/2} \cdot \frac{1}{\tau^2} \\
&= \frac{1}{2\tau^2} \frac{1}{\tilde{M}_\kappa(\cos(\theta_i^c), f_2(w_c))} \\
&> 0.
\end{aligned}
\tag{376}
$$

The derivative of $f_2^2$ is:

$$
\begin{aligned}
f_2^{2\prime}(w_c) &= \frac{d}{dw_c}\left(\cos^2\left(\theta_i^c\right)w_c^2 + \sin^2\left(\theta_i^c\right)\right) \\
&= 2\cos^2\left(\theta_i^c\right)w_c \\
&\geq 0.
\end{aligned}
\tag{377}
$$

Let $m = \tilde{M}_\kappa\left(\cos\left(\theta_i^c\right), f_2\left(w_c\right)\right)$. Then, the first derivative of $G_2$ is:

$$
\begin{aligned}
G_2'\left(w_c\right) &= G_3'\left(m\right)\tilde{M}_\kappa'\left(\cos\left(\theta_i^c\right), f_2\left(w_c\right)\right)f_2^{2\prime}\left(w_c\right) \\
&= \frac{I_{\nu+1}\left(m\right)}{I_\nu\left(m\right)}\frac{1}{2\tau^2 m}2\cos^2\left(\theta_i^c\right)w_c \\
&= \frac{\cos^2\left(\theta_i^c\right)w_c}{\tau^2}\frac{1}{m}\frac{I_{\nu+1}\left(m\right)}{I_\nu\left(m\right)} \\
&> 0.
\end{aligned}
\tag{378}
$$

Combining Eq. (374) and Eq. (378), we have:

$$
\begin{aligned}
\bar{\mathcal{J}}'\left(w_c; \kappa, \nu\right) = G'\left(w_c\right) &= G_1'\left(t\right) + G_2'\left(t\right) \\
&= \frac{\cos^2\left(\theta_i^c\right)}{\tau}\left(-1 + \frac{w_c}{\tau}\frac{1}{m}\frac{I_{\nu+1}\left(m\right)}{I_\nu\left(m\right)}\right).
\end{aligned}
\tag{379}
$$

Since $0 < w_c < 1$, then:

$$
\begin{aligned}
0 \leq w_c^2 \leq 1 &\Leftrightarrow \sin^2\left(\theta_i^c\right) \geq \sin^2\left(\theta_i^c\right)w_c^2 \\
&\Leftrightarrow \sin^2\left(\theta_i^c\right) \geq w_c^2 - \cos^2\left(\theta_i^c\right)w_c^2 \\
&\Leftrightarrow \cos^2\left(\theta_i^c\right)w_c^2 + \sin^2\left(\theta_i^c\right) \geq w_c^2 \\
&\Leftrightarrow f_2^2(w_c) \geq w_c^2.
\end{aligned}
\tag{380}
$$

Therefore, consider $\theta_i^c \in [0, \frac{\pi}{2}]$, we have:

$$
\begin{aligned}
m^2 &= \tilde{M}_\kappa^2\left(\cos\left(\theta_i^c\right), f_2\left(w_c\right)\right) \\
&= \kappa^2 + \frac{2\kappa\cos\left(\theta_i^c\right)}{\tau} + \frac{f_2^2(w_c)}{\tau^2} \\
&\geq \kappa^2 + \frac{2\kappa\cos\left(\theta_i^c\right)}{\tau} + \frac{w_c^2}{\tau^2} \\
&\geq \frac{w_c^2}{\tau^2} \\
&\geq 0,
\end{aligned}
\tag{381}
$$

which implies:

$$
m \geq \frac{w_c}{\tau} \Leftrightarrow \frac{w_c}{\tau}\frac{1}{m} \leq 1.
\tag{382}
$$

Plugging Eq. (375) and Eq. (382) into Eq. (379), we have:

$$
\begin{aligned}
\bar{\mathcal{J}}\left(w_c; \kappa, \nu\right) &= \frac{\cos^2\left(\theta_i^c\right)}{\tau}\left(-1 + \frac{w_c}{\tau}\frac{1}{m}\frac{I_{\nu+1}\left(m\right)}{I_\nu\left(m\right)}\right) \\
&< 0.
\end{aligned}
\tag{383}
$$

So we can conclude that, for any fixed $\theta_i^c \in [0, \frac{\pi}{2}]$, $\bar{\mathcal{J}}\left(\cdot\right)$ is a strictly decreasing function on $(0, 1]$. $\qquad\square$

**Lemma 15.** *Let $X$ be an $N$-point configuration, where $X = (x_1, \ldots, x_N) \in (\mathbb{S}^{h-1})^N$ are iid samples from $\mu = \mathrm{vMF}(c, \kappa)$. When $\kappa$ is sufficiently large, $\forall i, j \in [K], i \neq j$, it holds that:*

$$P(x_i \cdot x_j \geq 0) \approx 1. \tag{384}$$

*Proof.* Let $X \sim \mathrm{vMF}(c, \kappa)$ on $\mathbb{S}^{h-1}$ and set $U = c^\top X = \cos \Theta \in [-1, 1]$. Then:

$$P(X \cdot c \geq 0) = \frac{\int_0^1 e^{\kappa u} \left(1 - u^2\right)^{\frac{p-3}{2}} du}{\int_{-1}^1 e^{\kappa u} \left(1 - u^2\right)^{\frac{p-3}{2}} du}. \tag{385}$$

Using standard integral representations of the modified Bessel and modified Struve functions,

$$I_\nu(z) = \frac{(z/2)^\nu}{\sqrt{\pi}\Gamma\left(\nu + \frac{1}{2}\right)} \int_{-1}^1 e^{zt} \left(1 - t^2\right)^{\nu - \frac{1}{2}} dt,$$

$$\nu(z) = \frac{(z/2)^\nu}{\sqrt{\pi}\Gamma\left(\nu + \frac{1}{2}\right)} \int_0^1 2\sinh(zt) \left(1 - t^2\right)^{\nu - \frac{1}{2}} dt, \tag{386}$$

with $\nu = h/2 - 1$, the ratio simplifies to the neat closed form

$$P(X \cdot c \geq 0) = \frac{1}{2}\left(1 + \frac{L_\nu(\kappa)}{I_\nu}\right) \tag{387}$$

where $L_\nu$ the modified Struve function. And we list numerical values of this probability:

- $h = 128$:

| $\kappa$ | 1 | 5 | 10 | 20 | 30 | 50 | 100 | 200 |
|---|---|---|---|---|---|---|---|---|
| P | 0.5353 | 0.6710 | 0.8117 | 0.9609 | 0.9956 | 1.0000 | 1.0000 | 1.0000 |

- $h = 512$:

| $\kappa$ | 1 | 5 | 10 | 20 | 30 | 50 | 100 | 200 |
|---|---|---|---|---|---|---|---|---|
| P | 0.5176 | 0.5875 | 0.6708 | 0.8116 | 0.9075 | 0.9863 | 1.0000 | 1.0000 |

- $h = 1024$:

| $\kappa$ | 1 | 5 | 10 | 20 | 30 | 50 | 100 | 200 |
|---|---|---|---|---|---|---|---|---|
| P | 0.5125 | 0.5621 | 0.6227 | 0.7340 | 0.8258 | 0.9409 | 0.9991 | 1.0000 |

$\square$

