# OpenReview forum: "Decrypt Modality Gap in Multimodal Contrastive Learning: From Convergent Representation  to Pair Alignment"
_ICLR.cc/2026/Conference — ICLR 2026 Conference Withdrawn Submission_

### Official Review · Reviewer_9Gva · 2025-10-15

**Soundness:** 2
**Presentation:** 2
**Contribution:** 3
**Rating:** 4
**Confidence:** 4

**Summary:**

This paper introduces a thorough analysis of the modality gap from the perspective of von Mises-Fisher distributions, which is consistently different wrt previous works. The authors derive several theorems and corollary to prove some novel insights, among which the crucial one is that the modality gap is not caused by the contrastive loss itself nor by the cone initialization/constraint, but rather from the dimension collapse.
The perspective is interesting and novel.

**Strengths:**

The paper analyzes the modality gap under a different and novel perspective, proving findings that are consistenly novel wrt previous works.

The paper derives rigorous proofs for the claims.

The paper is certianly relevant to the multimodal learning community dealing with the modality gap and may open new discussions on this phenomenon.

**Weaknesses:**

W1) The paper contains several assumptions that are not proven, so some theorems and claims may not hold under different assumptions.
For example, in Section 3.1, the authors assume that (X,Y) are iid, but is it true? There may be dependencies on the two distributions, and it would be better to prove this assumption, at least empirically. Another example that should be clarified, the authors say that "Since the modality gap depends solely on the angle between the two center vectors", but this is an assumption. Indeed, this is a matter on how we want to define the modality gap. For example, this is the way Liang et al. 2022 defined the modality gap, but different definitions may exist such as the relative modality gap in Schrodi et al., 2025. Furthermore, the modality gap affects all the pairs, although it is reasonable to consider a single modality gap measure considering the centroids of the two distributions. However, this is an approximation that should be taken into account, especially if rigorous theorems are defined on top of it.

W2) Which is the derivation or the design choice of equation 9? Why do we consider this function? Further explanations can help understanding the choice.

W3) (minor wrt previous ones) Definition 8 is straightforward and redundant.

W4) The authors spent mainly the whole paper explaining what and why does not affect the modality gap, which is relevant and interesting, and reserve only the one page and a half for the method they propose and the experimental validation. Although I do not believe that tons of experiments are crucial to accept papers, the results and the design of the experiments proposed in this paper are not convincing.
For example:
W4.1) which is the difference between rotating the hyperplanes and shifting the two embedding distributions as in LIang et al? the post hoc shift can be done by decreasing the cosine similarity between the two centers, thus is somewhat rotating the two distributions in the hypersphere.
W4.2) Theoretically, the post hoc shift can achieve zero gap, so why it is not achieved in the experiments?
W4.3) The results of the proposed method are very similar to the posthoc shift, which has been however surpassed by novel methods to mitigate the gap, so the results are not convincing. The improvement is marginal and the method is not so different from the original one by Liang et al.

W5) Yaras et al., 2025 derives a thorough theoretical explanation for the modality gap and the link with the temperature. The authors should discuss more on the differences between their derivation and the one from Yaras et al. and provide some comparisons, as the light discussion present in Appendix is not sufficient. A more detailed discussion would strengthen authors' contributions.

W6) Although the authors put their results in bold, the standard CLIP achieves better retrieval performance. So, it is not clear why we should close the gap with the proposed method if it brings no improvements.

minor typos:
- caption of fig 2d, valuse instead of values.
- table 4 in appendix, I guess the time was running out and the deadline approached :) just a minor typo.

**Questions:**

See weaknesses.

---

### Official Review · Reviewer_U3BL · 2025-10-27

**Soundness:** 2
**Presentation:** 3
**Contribution:** 2
**Rating:** 2
**Confidence:** 4

**Summary:**

This paper presents a theoretical analysis of the **modality gap** in multimodal contrastive learning (MCL). It formalizes the conditions under which this gap disappears (in unconstrained or cone-restricted cases) and shows that under **subspace constraints**, the gap converges to the **angle between modality subspaces**, identifying **dimensional collapse** as its underlying cause. The authors further demonstrate that perfect pairwise alignment becomes impossible under such collapse and propose a **Shared Subspace Projection (SSP)** as a *post-hoc* remedy. Experiments on CLIP embeddings indicate that SSP effectively reduces the gap without degrading zero-shot performance.

**Strengths:**

1. The paper is well structured and clearly written, despite the number of theoretical and practical contributions.
2. The theoretical analysis appears correct and internally consistent.
3. It provides a simple and practical *post-hoc* method (SSP) that yields measurable improvements.

**Weaknesses:**

## Major
1. The analysis relies on several strong assumptions:
   - (i) embeddings are modeled as vMF i.i.d., whereas in practice their distributions are often far more complex (e.g., multimodal);
   - (ii) the infinite-batch limit is assumed — an analysis of how finite batch size affects the theoretical results would be valuable;
   - (iii) the intra-modal similarity assumption in **Theorem 4 (IMS)** rarely holds in practice, as similarity measures like **CKA** or **SVCCA** are seldom close to 1.

2. The experimental validation is very limited: only CLIP ViT-B/32 and zero-shot classification are evaluated.

3. The paper is not fully self-contained. For example, **SSP**, the main practical contribution, is described only in the Appendix, and some corollaries or theorems rely on results that also appear exclusively there (e.g., *Corollary 2*).

4. The “hyperplane rotation” result, while theoretically interesting, is not implemented or empirically demonstrated.

## Minor
1. **Line 123:** as defined, $f_I(i_1) \in \mathbb{R}^h$ rather than $\mathbb{S}^{h-1}$. Although embeddings are later normalized in contrastive learning, the encoder definition should reflect this explicitly.
2. **Line 179:** it would be clearer to state $k = 0$, since the text later refers directly to $k$ and not its inverse.
3. **Theorem 1:** $\mu_x$ and $\mu_y$ are not defined. Presumably these are distributions, but they should be explicitly introduced here (and ideally use a different letter, e.g., $f$, to avoid confusion with means).
4. **Definition 8:** alignment metrics such as **CKA** or **SVCCA** typically measure *similarities of similarities*, not direct similarities as defined here. The concept is reasonable, but the term *alignment* may cause confusion with prior literature.
5. Several typos and small issues:
   - “changeS” (line 166)
   - missing space before “a” (line 167)
   - $(-1)^k$ in Eq. (6)
   - “Valuse” → “Values” in Fig. 6 caption
   - $y_i$ instead of $y_x$ in Definition 8
   - “post-hoc” (line 446)
   - double “and” (line 456)

**Questions:**

1. Could you evaluate SSP on additional models?
2. You mention that “rotating a high-dimensional hyperplane can be complicated in practice.” Could this be implemented via an orthogonal mapping?
3. Does reducing the modality gap consistently correlate with downstream improvements when the IMS assumption does not hold?

---

### Official Review · Reviewer_evtg · 2025-10-31

**Soundness:** 3
**Presentation:** 3
**Contribution:** 3
**Rating:** 4
**Confidence:** 4

**Summary:**

This paper investigates the modality gap in multimodal contrastive learning (MCL), the phenomenon where embeddings from different modalities occupy distinct regions in the representation space. Despite extensive empirical observations, the causes and effects of this gap remain unclear.

The authors propose the first theoretical framework to analyze optimal representations in MCL and their alignment at convergence. They show that without any constraints or under a cone constraint the modality gap disappears, but persists under a subspace constraint, where embeddings from each modality collapse into distinct hyperplanes. This theoretical result identifies dimension collapse as the fundamental cause of the modality gap. The authors further demonstrate that perfect alignment of paired samples is impossible under the subspace constraint, but can still be recovered through hyperplane rotation or shared space projection. Overall, the paper provides a principled explanation for the emergence of modality gaps and their impact on downstream task performance. Although the theoretical framework and results are rigorous and well presented, the experimental validation is weak and does not allow one to fully assess the potential of the proposed theory.

**Strengths:**

1. The paper presents the first rigorous theoretical study on the modality gap and the representational properties of cross-modal contrastive learning, providing valuable formal insights into a phenomenon that has so far been primarily studied empirically. This contribution lays important groundwork for understanding the limitations and behavior of contrastive alignment objectives.

2. The paper is overall clear and well-organized, with precise mathematical reasoning and well-structured proofs that make complex theoretical results accessible. The implications of each theorem are clearly articulated, helping the reader grasp both the technical and conceptual contributions.

**Weaknesses:**

1. **Limited experimental validation.**
Although the theoretical framework is rigorous and elegant, the experimental results are rather weak. The empirical analysis is too limited in scope (only zer-shot classification) to convincingly support or challenge the theoretical findings. As a result, it remains unclear how the modality gap impacts performance in practical settings. Results on retrieval tasks are missing (appendix).

2. **Restricted to cross-modal contrastive learning.**
The analysis is focused solely on standard cross-modal contrastive learning setups (i.e. CLIP-like), while several alternative paradigms, such as Factorized Contrastive Learning (FactorCL) [Liang et al., NeurIPS 2023], CoMM [Dufumier et al., ICLR 2025],  SLIP [Mu et al., ECCV 2022], propose different alignment mechanisms or objectives, all based on contrastive learning for multimodal data. A discussion or comparison with these frameworks would have strengthened the paper and clarified the generality of the proposed theory.

3. **Narrow modality scope (image–text only).**
The work appears restricted to image–text contrastive learning, which limits its generality across other modality pairs (e.g., audio–video, text–graph, or vision–language–audio). The title should be revised to reflect this limitation more accurately.

4. **Ambiguity in the definition of alignment.**
The notion of alignment is central to the paper’s arguments, yet it is not clearly defined until later in the text (l. 354). This can cause confusion early on, especially for readers trying to connect the theoretical framework with prior definitions of alignment in contrastive learning literature.

5. **Questionable practical validity of assumptions.**
Some of the mathematical assumptions, e.g. the subspace constraint or the simplified geometry or distributions of embeddings, may be too idealized compared to real neural network representations. It would help to discuss the extent to which these assumptions are met in practice and whether the conclusions remain valid when they are relaxed.

6. **Unconvincing experimental results.**
Beyond being limited, the reported experiments do not provide strong evidence that the theoretical findings translate into measurable benefits or predictive power. The gap between theory and practice remains large, preventing a full assessment of the practical potential of the proposed approach.

**Questions:**

- Could the authors expand the experimental section to include more diverse datasets, modalities, and tasks?

- How would the proposed theoretical framework extend to other contrastive paradigms such as SLIP, FactorCL or CoMM?

- How does their notion of alignment differ from pairwise similarity as defined in existing works?

- How realistic are the subspace and cone constraints in the context of deep embeddings?

- Can the authors provide additional experiments demonstrating how their theoretical insights can guide practical model design or improve performance?

- In lines 382–384, the authors state that “Theorem 4 implies that MCL aims to maximize the mutual information between modalities, while preserving modality-specific information in the complementary space.”
However, this interpretation is not immediately evident from the theorem’s formulation. Could the authors expand on how this conclusion follows from Theorem 4, and clarify the derivation or underlying intuition? In particular, how does this result relate to the findings of Dufumier et al. (ICLR 2025), who showed that cross-modal contrastive learning methods are limited to capture only redundant information across modalities?

- Finally, is alignment truly a critical issue in multimodal contrastive learning, or does it have limited practical impact on downstream tasks?

---

### Official Review · Reviewer_61wz · 2025-11-03

**Soundness:** 2
**Presentation:** 4
**Contribution:** 2
**Rating:** 2
**Confidence:** 4

**Summary:**

The paper tries to understand how the representations converge in multimodal settings.

**Strengths:**

1. The paper addresses an important question of how representations align and converge in multimodal models optimized with contrastive objectives.

2. The presentation is clear and the paper is well structured.

**Weaknesses:**

1. The paper relies on multiple definitions and assumptions that lack sufficient justification or empirical grounding (see questions for details).

2. The validity of the paper’s conclusions is uncertain given the current analysis (see questions for clarification).

**Questions:**

1. The definition of modality gap appears problematic. It essentially measures the closeness of the centers of two sets of representations, but this is a very weak notion of alignment. One can easily construct counterexamples, for instance, taking a set X and rotating it would yield representations that are structurally different yet have identical centers, resulting in zero gap. This suggests that the definition may not capture the intended concept of alignment or convergence.

2. It is unclear why the “conic” constraint is theoretically valid. It seems to arise primarily due to ReLU activations, which restrict representations to the positive orthant. However, representations taken directly after a linear layer need not reside in a cone, and switching to an activation function such as Tanh would violate this assumption entirely. This raises concerns about the generality and necessity of the conic formulation.

3. The paper assumes that data from the two modalities are independently and identically distributed (i.i.d.) from two distinct distributions. In a multimodal setting, one would typically assume that the data are sampled jointly from a single multimodal distribution. Why is it appropriate to assume independent distributions across modalities? This assumption seems to contradict the notion of aligned multimodal pairs.

4. The argument in lines 347–352 appears inconsistent with [A]. In [A], it is explicitly stated that maximizing alignment does not guarantee improved downstream performance. Therefore, it is unclear what is meant by the claim that maximizing alignment necessarily improves or correlates with downstream accuracy.

5. The motivation for Table –1 is unclear. The performance of CLIP alone is already strong, but incorporating SSP, Removal Transaction, etc., appears to degrade performance. What insight or justification does this table provide? If these methods consistently reduce performance, what is their intended contribution?

---

**References**

[A] Tjandrasuwita, M., Ekbote, C., Ziyin, L., & Liang, P. P. (2025). Understanding the emergence of multimodal representation alignment. arXiv preprint arXiv:2502.16282.

---

### Note · Authors · 2025-12-04

**Comment:**

We are grateful for the reviewers’ careful analysis and valuable comments, which have greatly helped us understand the strengths and limitations of our work. With appreciation for their efforts, we wish to withdraw our submission from further consideration.

**Withdrawal Confirmation:**

I have read and agree with the venue's withdrawal policy on behalf of myself and my co-authors.